# The molecular evolution of spermatogenesis across mammals

Florent Murat[1,2,19 ✉], Noe Mbengue[1,19 ✉], Sofia Boeg Winge[3,4,5], Timo Trefzer[6], Evgeny Leushkin[1], Mari Sepp[1], Margarida Cardoso-Moreira[7], Julia Schmidt[1], Celine Schneider[1], Katharina Mößinger[1], Thoomke Brüning[1], Francesco Lamanna[1], Meritxell Riera Belles[5], Christian Conrad[6], Ivanela Kondova[8], Ronald Bontrop[8], Rüdiger Behr[9,10], Philipp Khaitovich[11], Svante Pääbo[12], Tomas Marques-Bonet[13,14,15,16], Frank Grützner[17], Kristian Almstrup[3,4,18], Mikkel Heide Schierup[5] & Henrik Kaessmann[1 ✉]

The testis produces gametes through spermatogenesis and evolves rapidly at both the morphological and molecular level in mammals[1–6], probably owing to the evolutionary pressure on males to be reproductively successful[7]. However, the molecular evolution of individual spermatogenic cell types across mammals remains largely uncharacterized. Here we report evolutionary analyses of single-nucleus transcriptome data for testes from 11 species that cover the three main mammalian lineages (eutherians, marsupials and monotremes) and birds (the evolutionary outgroup), and include seven primates. We find that the rapid evolution of the testis was driven by accelerated fixation rates of gene expression changes, amino acid substitutions and new genes in late spermatogenic stages, probably facilitated by reduced pleiotropic constraints, haploid selection and transcriptionally permissive chromatin. We identify temporal expression changes of individual genes across species and conserved expression programs controlling ancestral spermatogenic processes. Genes predominantly expressed in spermatogonia (germ cells fuelling spermatogenesis) and Sertoli (somatic support) cells accumulated on X chromosomes during evolution, presumably owing to male-beneficial selective forces. Further work identified transcriptomal differences between X- and Y-bearing spermatids and uncovered that meiotic sex-chromosome inactivation (MSCI) also occurs in monotremes and hence is common to mammalian sex-chromosome systems. Thus, the mechanism of meiotic silencing of unsynapsed chromatin, which underlies MSCI, is an ancestral mammalian feature. Our study illuminates the molecular evolution of spermatogenesis and associated selective forces, and provides a resource for investigating the biology of the testis across mammals.

The rapid evolution of the testis across mammals is probably mainly explained by positive selection associated with sperm competition, which reflects the evolutionary pressure on males to achieve reproductive success[7]. Consequently, testis sizes, sperm production rates, sperm morphologies and other cellular traits substantially vary across mammals, even between closely related species such as the great apes, due to great mating system differences, especially regarding the extent of female promiscuity[7]. The rapid evolution of the testis is reflected at the molecular level. Previous gene expression comparisons for various organs across mammals revealed that rates of evolutionary expression change are highest in the testis, probably due to frequent adaptive changes but potentially also widespread relaxation of purifying selection[1–6]. Consistently, genes with testis-specific expression tend to be enriched with genes whose coding sequences have been shaped by positive selection[8]. In addition, new genes that emerge during evolution tend to be predominantly expressed in the testis and thus probably also contribute to its rapid phenotypic evolution[3,9].

The testis also shows several other unique molecular features. First, chromatin in spermatogenic cells is massively remodelled during spermatogenesis, a process that culminates in the tight packaging of DNA around protamines in the compact sperm head[10]. This remodelling leads to widespread leaky transcription in the genome[11], which in turn probably facilitates the initial transcription and, hence, the frequent emergence of new testis-expressed genes and alternative exons during evolution[3,9,11,12]. Second, the differentiation of sex chromosomes from ancestral autosomes triggered the emergence of MSCI in eutherians and marsupials[13] (therians), which led to the establishment of backup gene copies that substitute for parental genes on the X during meiosis[3,14]. In spite of MSCI, the X chromosome has become enriched with testis-expressed genes during evolution[3,15–20], presumably due

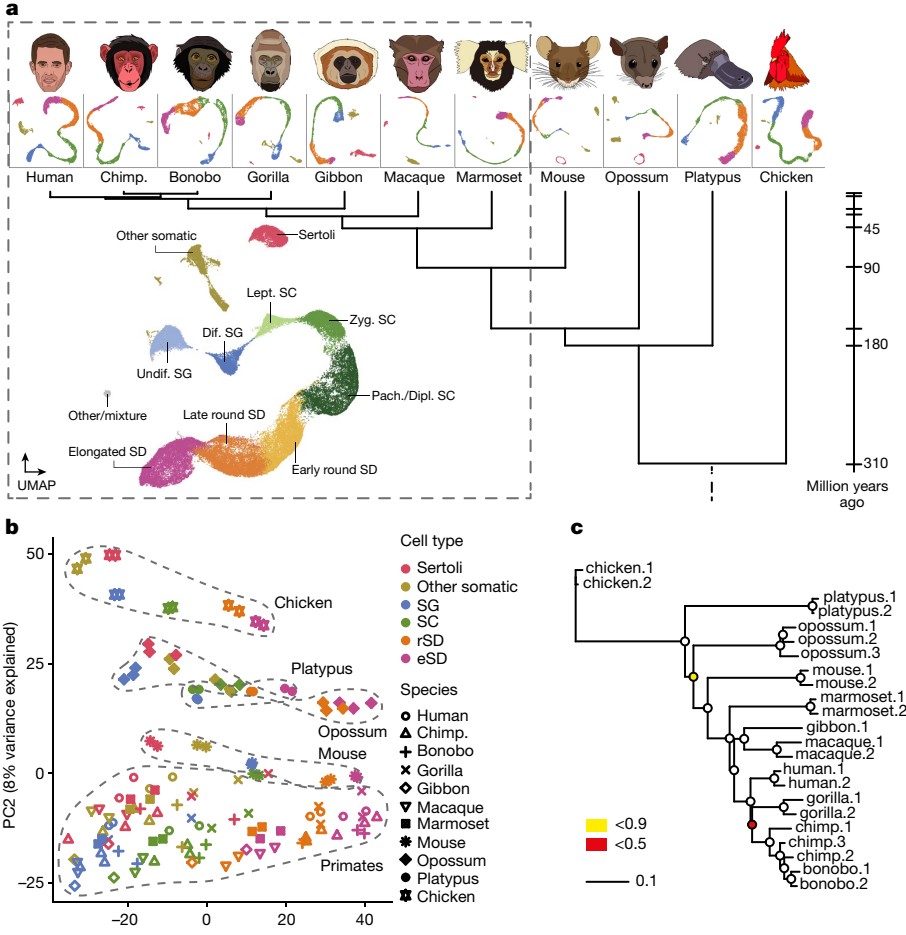

**Fig. 1 | snRNA profiling across ten mammals and a bird. a**, Species sampled and uniform manifold approximation and projection (UMAP) of snRNA-seq datasets. UMAP of the integrated primate dataset (dashed box), showing undifferentiated and differentiated SG (undif. SG and dif. SG, respectively), leptotene, zygotene, pachytene and diplotene SCs (lept. SC, zyg. SC, pach. SC and dipl. SC, respectively), spermatids (SD) and somatic cell types.

Chimp., chimpanzee. **b**, Principal component (PC) analysis of cell-type pseudo-bulks. Species and lineages are encircled by a dashed line. Each symbol represents an individual. **c**, Gene expression phylogeny based on pseudo-bulk transcriptomes for whole testes. Bootstrap values (4,498 1:1 orthologous amniote genes were randomly sampled with replacement 1,000 times) are indicated by circles, ≥0.9 (white fill).

to sexually antagonistic selective forces favouring the fixation of male-beneficial mutations on this chromosome[21]. Finally, translational regulation of transcriptomes is widespread across spermatogenesis[6].

Previous large-scale transcriptomic investigations of testis evolution were largely limited to bulk-organ samples[1–6,19]. Recent high-throughput single-cell (sc) or single-nucleus (sn) RNA-sequencing (RNA-seq) technologies enable detailed investigations of the cellular and molecular evolution of the testis, as exemplified by two scRNA-seq comparisons between human, macaque and mouse[22,23], but a comprehensive investigation of the evolution of spermatogenesis across all main mammalian lineages is lacking.

Here we provide an extensive snRNA-seq resource covering testes from ten representative mammals and a bird, enabling detailed comparisons of spermatogenic cells and underlying gene expression programs within and across mammals (https://apps.kaessmannlab. org/SpermEvol/). Our evolutionary analyses of these data unveiled ancestral as well as species- and lineage-specific cellular and molecular characteristics of mammalian spermatogenesis.

## Spermatogenesis across 11 species

We generated snRNA-seq data for testes from ten species that cover the three main mammalian lineages and include key primate species,

representing all simian (anthropoid) lineages (Fig. 1a): eutherian mammals (representatives for five of the six extant ape lineages, including humans; rhesus macaque, an Old World monkey; common marmoset, a New World monkey; and mouse), marsupials (grey short-tailed opossum) and egg-laying monotremes (platypus). Corresponding data were generated for a bird (red jungle fowl, the progenitor of domestic chicken; hereafter referred to as 'chicken'), to be used as an evolutionary outgroup. The dataset consists of 27 libraries, with one to three biological replicates per species and a median of roughly 275 million snRNA-seq reads per library (Supplementary Table 1). We refined and extended existing genome annotations across all species on the basis of bulk-testis RNA-seq data (seven libraries) (Supplementary Tables 1 and 2 and Methods), to ensure optimal read-mapping and prevent biases in cross-species analyses. After quality controls and filtering steps (Methods), we obtained transcriptomes for a total of 97,521 high-quality nuclei for the 11 species, with a mean of 8,866 cells per species, a median of 1,856 RNA molecules (unique molecular identifiers (UMIs)) detected per cell and low percentages of mitochondrial UMIs (Supplementary Fig. 1 and Supplementary Table 1).

We identified the main germ cell types[11] along the continuous cellular proliferation and differentiation path of spermatogenesis across all species: spermatogonia (SG), the mitotic cells fuelling spermatogenesis, including spermatogonial stem cells; spermatocytes (SC), where

meiosis takes place; and the haploid round spermatids (rSD) and elongated spermatids (eSD), which together reflect spermiogenesis (Fig. 1a, Extended Data Figs. 1a–f, Supplementary Table 3 and Methods). We also identified separate clusters corresponding to somatic testicular cells, in particular Sertoli cells (except for platypus, in which Sertoli cells could not be unambiguously distinguished), the main spermatogenesis support cells, but also other cell types, such as Leydig cells, peritubular cells, endothelial cells and macrophages (Extended Data Figs. 1a–e). The close evolutionary relationship of the seven primates in our study enabled the direct integration of datasets across these species and thus the identification of sub-cell types that correspond to intermediate events during spermatogenesis (Fig. 1a). For all species, we traced the dynamic gene expression programs underlying spermatogenic differentiation and key molecular events, thus also identifying a host of new marker genes (Extended Data Fig. 1g and Supplementary Table 4).

To obtain an overview of cell-type relationships across species, we performed a principal component analysis (PCA) based on pseudo-bulk cell-type transcriptomes (Fig. 1b). The first principal component (PC1) orders the spermatogenic cell types according to the progression of spermatogenesis for all species (Fig. 1b, from left to right). This observation suggests that our data capture ancestral aspects of spermatogenic gene expression programs that are shared across mammals or amniotes, despite the rapid evolution of the testis[1–6] and the long divergence times of 310 million years (Fig. 1a). PC2 separates the data by species or lineages, reflecting diverged aspects of spermatogenesis, whereas PC1 and PC2 together separate somatic and spermatogenic cell types for each species. The close clustering of biological replicates is a further indicator of the high data quality.

## Rates of evolution along spermatogenesis

To investigate rates of gene expression evolution across cell types, we reconstructed gene expression trees (Methods). A tree based on pseudo-bulk transcriptomes for the whole testis (Fig. 1c) recapitulates the known mammalian phylogeny (Fig. 1a, except for the gibbon–macaque grouping), akin to previous trees based on bulk-tissue RNA-seq data across mammalian organs[1,2]. This observation is consistent with the view that regulatory changes steadily accumulated over evolutionary time[1], with present-day RNA abundances reflecting the evolution of mammalian lineages and species.

To trace the cellular source of the rapid evolution of the testis, we built expression trees for the different cell types, which also recapitulate the known species relationships (Extended Data Fig. 2a). Notably, the total branch lengths of the trees, which reflect the amount of evolutionary expression change, vary substantially between cell types (Fig. 2a). Whereas the rate of expression evolution is similar in Sertoli cells and diploid spermatogenic cells (and lower than that in other somatic cell types), it is substantially higher in the postmeiotic haploid cell types (rSD and eSD), consistent with a recent inference based on data for three eutherians[23]. The higher resolution afforded by a primate-specific analysis provides further details (Fig. 2a). Starting in late meiosis (pachytene SC), evolutionary rates progressively increase until the end of spermiogenesis (late eSD). Thus, late spermatogenic stages drive the previously observed rapid evolution of the testis[1–6].

Pairwise species comparisons, including downsampling analyses (Extended Data Fig. 2b), confirm the rapid expression evolution of postmeiotic cell types across amniotes and that gene expression divergence increases with evolutionary time (Fig. 2b), in accord with the expression phylogeny results (Extended Data Fig. 2a). It is noteworthy, however, that expression divergence levels are roughly as similar between human and chicken as they are between human and platypus, although the bird lineage diverged 110 million years before the separation of monotremes and therians (that is, eutherians and marsupials). This observation, previously made for whole organs[1], supports at the cellular level the notion that the conservation of core spermatogenic functions restricts transcriptome divergence.

## Evolutionary forces

We sought to trace the evolutionary forces underlying the rapid evolution of late spermatogenesis. Two non-mutually exclusive patterns of natural selection may account for this observation. First, later stages of spermatogenesis might evolve under weaker purifying selection (that is, reduced functional constraints) and hence be less refractory to change. Second, the greater divergence in later stages might result from stronger positive selection, increasing the rate of fixation of adaptive changes. To investigate patterns of functional constraint during spermatogenesis, we assessed the tolerance to functional mutations of genes[24] used in different spermatogenic stages in humans, which showed a progressive increase of mutational tolerance starting during meiosis and culminating in early spermiogenesis (Fig. 2c and Extended Data Fig. 3a). Consistently, on the basis of a set of neutrally ascertained mouse knockouts[25], we found that the percentage of expressed genes associated with lethality decreases during spermatogenesis (Fig. 2d). Also, in agreement with a progressive reduction in functional constraints towards later spermatogenic stages, we find that the normalized rate of amino acid altering substitutions in coding sequences across primates is higher in late spermatogenesis (Fig. 2e and Extended Data Fig. 3b), although this increase might additionally reflect a higher proportion of genes under positive selection. Indeed, an examination of the temporal expression pattern of genes whose encoded protein sequences have been shaped by positive selection revealed a notable increase in percentages of positively selected genes used during spermatogenesis, with a peak in rSD (Fig. 2f and Extended Data Fig. 3c).

Because new genes also contribute to evolutionary innovations, we investigated the temporal contribution of recently emerged genes to gene expression programs in germ cells, using an index that combines the phylogenetic age of genes with their expression[2] (Methods). This analysis revealed that transcriptomes become younger during spermatogenesis (Fig. 2g and Extended Data Fig. 3d), indicating that new genes have increasingly more prominent roles in later stages, particularly in rSD, consistent with previous observations[22,23]. Previous work based on bulk cell-type analyses in mouse[11] uncovered a transcriptionally permissive chromatin environment during spermatogenesis, in particular in rSD, which was suggested to have facilitated the emergence of new genes during evolution[3,9,11]. Consistently, we detect in all species considerably increased contributions of intergenic transcripts after meiosis and a concomitant decrease in the contributions of protein-coding genes (Fig. 2h and Extended Data Fig. 3e). Notably, an analysis of translatome data[6] revealed a decline of translational efficiencies of transcripts during spermatogenesis (reaching a minimum in rSD) in all species (Fig. 2i and Extended Data Fig. 3f). This decline is consistent with observations from mouse bulk data for a restricted number of cell types[6] and probably mitigates the functional consequences of the concurrent increase in transcriptional promiscuity of the genome.

We next explored the reasons underlying the dynamic changes of selective forces and patterns of innovation during spermatogenesis. The breadth of expression across tissues and developmental processes (here referred to as expression pleiotropy) was proposed to represent a key determinant of the types of mutation that are permissible under selection[26]. We, therefore, assessed patterns of expression pleiotropy across spermatogenesis using spatiotemporal transcriptome data for several mammalian organs[2], which revealed that genes used later in spermatogenesis, in particular those in rSD, have substantially more specific spatiotemporal profiles than genes used earlier in spermatogenesis and in somatic cells (Fig. 2j and Extended Data Fig. 3g,h). Given that a decrease in expression pleiotropy can explain both a decrease in functional constraints and an increase in adaptation[2,26], we suggest

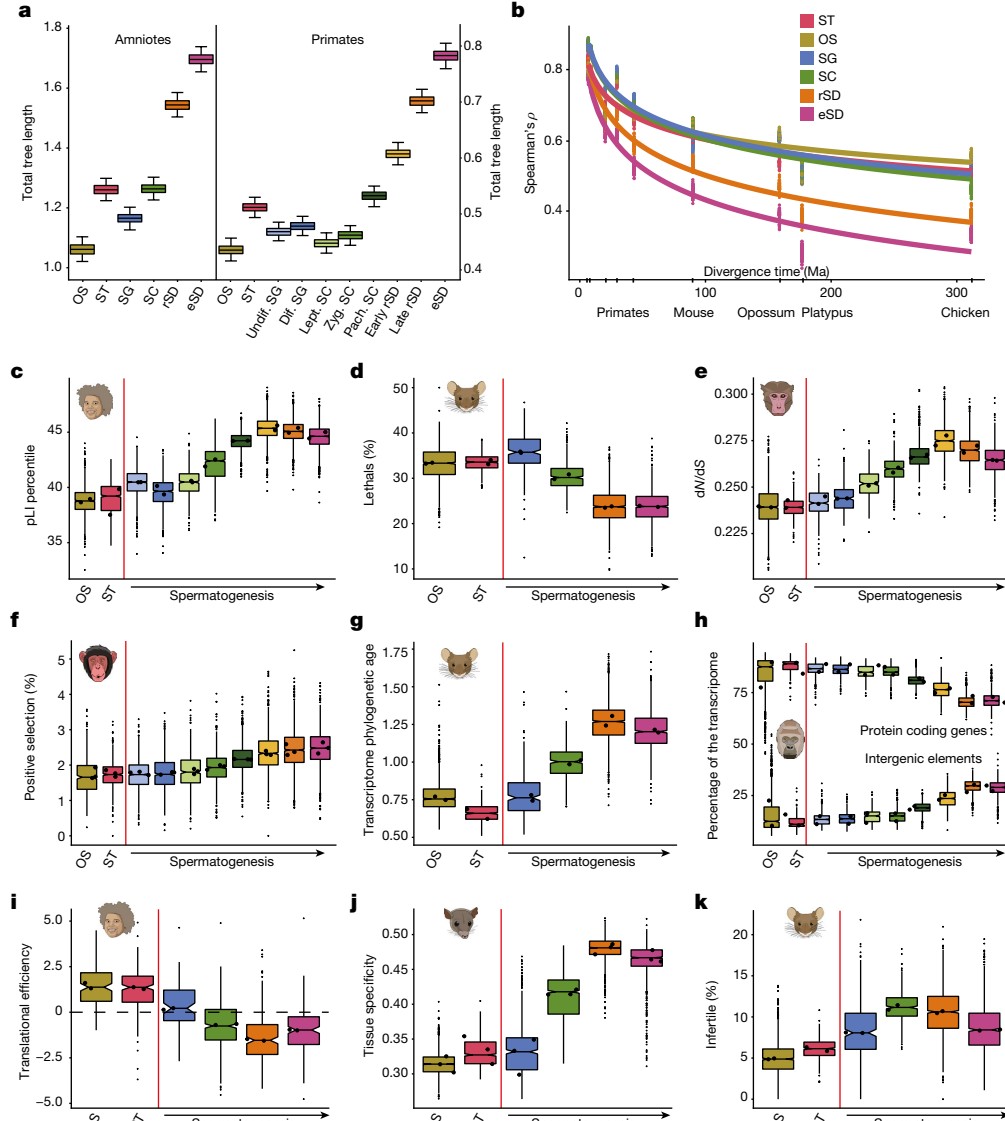

**Fig. 2 | Gene expression divergence and evolutionary forces. a**, Total branch lengths of expression trees among testicular cell types for amniotes and primates. Box plots show the median (central value); upper and lower quartile (box limits) and 95% confidence intervals (whiskers) for 1,000 bootstrap replicates. **b**, Spearman's correlations between humans and other species from 100 bootstrap replicates (dots). Lines correspond to linear regression trends (after log transformation of the time axis. Regression $R^2$ values range from 0.86 to 0.97. Ma, millions of years ago. **c**, Mean pLI value of expressed genes in human (the pLI score reflects the tolerance of a gene to a loss-of-function mutation; lower values mean less tolerance). **d**, Percentage of expressed genes leading to a lethal phenotype when knocked out in mouse (out of 4,742 knockouts[25]). **e**, Mean normalized ratio of nonsynonymous (d$N$) over synonymous (d$S$) nucleotide substitutions of expressed genes in macaque.

**f**, Percentage of expressed genes under positive selection (out of 11,170 genes tested for positive selection) in chimpanzee. **g**, Mean phylogenetic age of expressed genes in mouse. **h**, Percentage of UMIs mapping to protein-coding genes (top) or intergenic elements (bottom) in gorilla. **i**, Translational efficiency values (data from ref. [6]) are plotted for all genes with predominant expression in a given cell type in human. **j**, Mean of tissue-specificity values (data from ref. [2]) in opossum. **k**, Percentage of expressed genes associated with infertility (out of 3,552 knockouts) in mouse. **c**–**h**,**j**,**k**, Plotted is the mean value per cell. **c**–**k**, Superimposed thick black dots indicate medians from biological replicates. Box plots depict the median (centre value); upper and lower quartile (box limits) with whiskers at 1.5 times the interquartile range. Red lines separate somatic (OS, other somatic; ST, Sertoli cells) and germ cells. Data for other studied species are shown in Extended Data Fig. 3.

that it is probably a main contributor to the accelerated molecular evolution in late spermatogenesis. In addition, the specific type of selection acting on haploid cells[27] (haploid selection), in which expressed alleles are directly exposed to selection, may have contributed to the exceptionally rapid evolution of rSD.

Whereas the tissue- and time-specific late spermatogenic genes, in general, are not essential for viability (Fig. 2d,j and Extended Data Fig. 3g,h, above), we proposed that the specific aforementioned evolutionary forces indicate that many of these genes evolved crucial roles in spermatogenesis. Indeed, we find that the proportion of genes

associated with infertility[25] is relatively high in SC and spermatids (especially rSD): higher than in SG and somatic cells (Fig. 2k).

## Gene expression conservation and innovation

We next sought to trace the individual genes underlying conserved (ancestral) and diverged aspects of germ cells by comparing expression trajectories along spermatogenesis of one-to-one (1:1) orthologous genes across species[2] (Supplementary Fig. 2 and Methods). For the primates, this analysis revealed roughly 1,700–2,900 genes with conserved

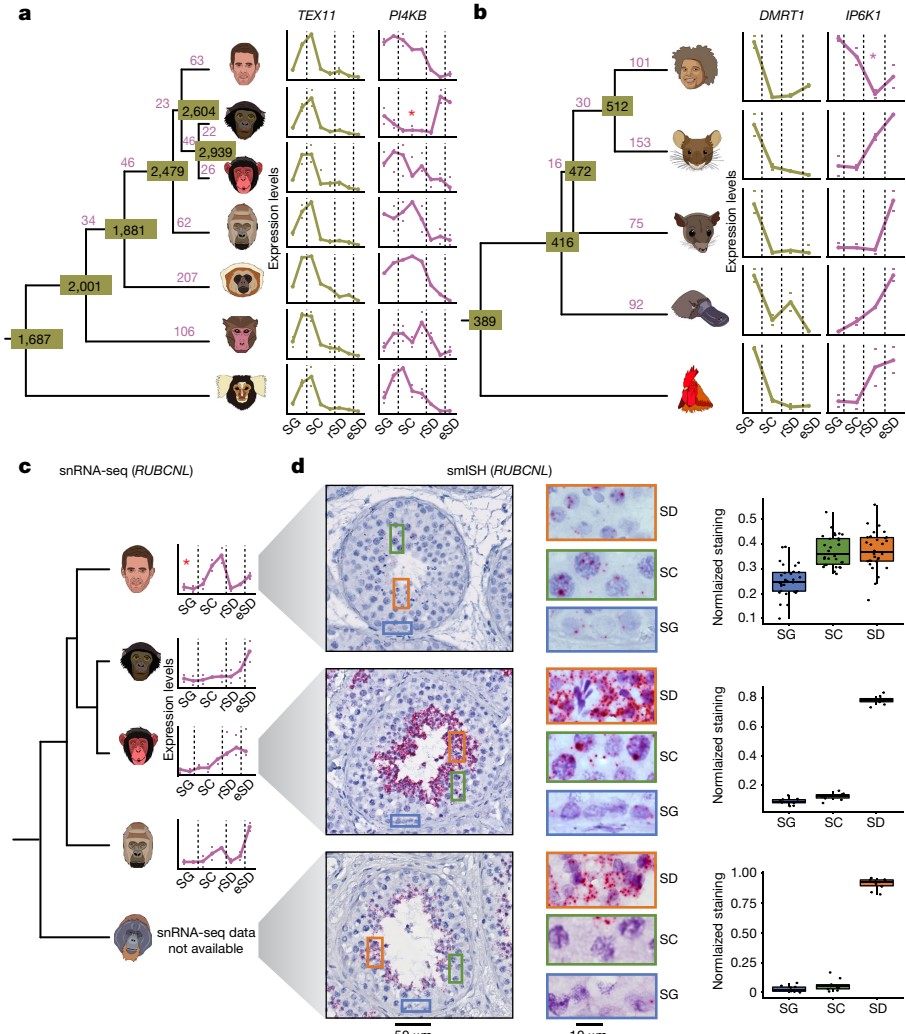

**Fig. 3 | Evolution of gene expression trajectories along spermatogenesis.**
**a**,**b**, The numbers of changed trajectories (in purple) and conserved trajectories (in olive) are indicated. In **a**–**c**, red asterisks indicate the branch for which a trajectory change has been called. For each replicate, mean expression levels across cells of a given cell type were calculated. Marks indicate values for the replicates with the highest and lowest mean expression levels, dots indicate the median of mean expression values of the replicates (three replicates for chimpanzee and opossum; two for human, bonobo, gorilla, macaque, marmoset, mouse, platypus and chicken; one for gibbon). In **a**–**c**, dashed vertical lines separate SG, SCs, rSD and eSD. **a**, Primate trajectories (human, bonobo, chimpanzee, gorilla, gibbon, macaque and marmoset; from top to bottom) based on 4,459 1:1 orthologues. *TEX11* and *PI4KB* are examples of conserved and changed trajectories, respectively. **b**, Amniote trajectories (human, mouse, opossum, platypus and chicken; from top to bottom) based on

2,927 1:1 orthologues. *DMRT1* and *IP6K1* are examples of conserved and changed trajectories, respectively. **c**, *RUBCNL* expression trajectory from snRNA-seq data along spermatogenesis for human, bonobo, chimpanzee and gorilla (from top to bottom). Available orangutan samples did not allow for the generation of high-quality snRNA-seq data. **d**, Detection of *RUBCNL* (each red dot reflects a single transcript) expression in human, chimpanzee and orangutan testis by smISH using RNAScope. Left, seminiferous tubule cross sections counterstained with haematoxylin, and closeups on areas containing SG, SC and spermatids (SD). Right, quantification of *RUBCNL* expression levels in ten tubules per section ($n = 3$ for human, $n = 1$ for chimpanzee and orangutan). Box plots represent the mean (central value) distribution of staining (dots); upper and lower quartile (box limits) with whiskers at 1.5 times the interquartile range.

expression trajectories across different lineages or species (Fig. 3a and Supplementary Table 5). For example, the temporal expression of 1,687 genes is conserved across the seven primates and probably reflects the core ancestral gene expression program of the simian testis.

By contrast, we detected 635 trajectory changes during the evolution of apes and Old World monkeys (Fig. 3a and Supplementary Table 5). For example, 63 and 94 trajectory changes occurred in the human and chimpanzee–bonobo lineages, respectively, since their divergence roughly 7 million years ago. We spatially validated three of these changes using single-molecule RNA in situ hybridization (smISH) for three great apes (human, chimpanzee and orangutan as the outgroup) (Fig. 3c, Extended Data Fig. 4, Supplementary Fig. 3, Supplementary Table 6 and Methods). Our smISH experiments confirm that the

expression of the gene *RUBCNL*, encoding a regulator of autophagy[28], changed in the human lineage, with a relative reduction of expression levels in spermatids (Fig. 3c,d). Thus, the role of *RUBCNL* in autophagy during spermatogenesis[29] is potentially different in humans compared to other Old World anthropoids. We also confirmed the inversion of the expression trajectory of the myosin-encoding gene *MYO3B* in SG and rSD on the chimpanzee–bonobo lineage (Extended Data Fig. 4a), as well as the expression increase of *ADAMTS17*, a family-member of proteases with key spermatogenic functions[30], in rSD relative to SG in humans (Extended Data Fig. 4b). We note that observed quantitative and partly qualitative expression pattern differences between the two complementary data types are expected because of technical differences.

We also assessed the conservation of expression trajectories across the three main mammalian lineages and amniotes (Fig. 3b and Supplementary Table 7). The 416 genes with conserved expression across all mammals probably trace back to ancestral gene expression programs of the ancestral mammalian testis, whereas 389 genes may represent the core ancestral spermatogenic program of amniotes. A notable example is *DMRT1*, which is highly expressed specifically in SG across amniotes and is required for mouse spermatogonial stem cell maintenance and replenishment[31].

In agreement with these highly conserved sets of genes having key roles in spermatogenesis, our analyses of fertility phenotypes[25] unveiled that genes involved in fertility are significantly more conserved in their expression trajectories than genes not associated with fertility (Extended Data Fig. 5b). Consistently, a Gene Ontology[32] enrichment analysis indicates an involvement of conserved genes in fundamental spermatogenic processes that are typical of the cell type in which they show peak expression (Extended Data Fig. 6a,b). Thus, genes with conserved trajectories for which spermatogenesis functions remain uncharacterized represent promising candidates for the exploration of fertility phenotypes (Supplementary Tables 5, 7 and 8). Notably, genes with lineage-specific trajectory changes are enriched with broader, typically metabolic, processes (Extended Data Fig. 6c,d). They are also significantly more tissue- and time-specific than genes with conserved expression (Extended Data Fig. 6e), which may have facilitated their expression change during evolution because of reduced pleiotropic constraints. However, genes with changed trajectories nevertheless include many genes for which key fertility functions have been described (Supplementary Tables 7 and 8). For example, *IP6K1*, which elicits infertility when knocked out in the mouse[33], shows strongly increasing expression towards the end of spermiogenesis in all amniotes except the primates, for which expression is high in SG and then declines (Fig. 3b and Extended Data Fig. 5c). Thus, the primary function of *IP6K1* probably shifted from late to early spermatogenesis during primate evolution.

In conjunction with mouse fertility data, we used our data to explore the contribution of new genes (mostly arising from gene duplications) to the evolution of new spermatogenic functions, focusing on the rodent lineage leading to mouse. This analysis revealed the emergence of key spermatogenic genes at different time points during evolution (Extended Data Fig. 7a and Supplementary Table 8), such as two rodent-specific retrogenes—*D1Pas1* and *H2al2a*—that originated through RNA-based duplication from parental genes on the X chromosome (Extended Data Fig. 7c). *D1Pas1* is essential for meiosis[34], whereas *H2al2a* is essential for genome compaction in late spermatogenesis[35]. Both genes show strongly increasing expression levels in late spermatogenesis (Extended Data Fig. 7b) and are thus in agreement with new genes contributing to functional roles predominantly during late spermatogenesis (above, Fig. 2g and Extended Data Fig. 3d).

Finally, we used our data to investigate ligand–receptor interactions underlying the communication between Sertoli cells, which have a central role in supporting and controlling spermatogenesis[11], and germ cells across species (Methods). Our cross-species comparisons revealed various conserved known and new ligand–receptor interactions (Extended Data Fig. 8a,b and Supplementary Table 9). For example, our data provide evidence that communication of Sertoli cells with SG, SC and rSD occurs in all amniotes through interactions of the cell adhesion molecule CADM1 (ref. [36]) or between CADM1 (Sertoli cells) and NECTIN3 (SC and rSD) (Supplementary Table 9). We note that CADM1 was previously thought to not to be expressed in Sertoli cells[37], but our data show high expression in this cell type across amniotes, in agreement with human protein atlas data (proteinatlas.org/ENSG00000182985-CADM1/tissue/testis#). Our work also supports the notion that the NECTIN2–NECTIN3 complex mediates the communication of Sertoli cells with spermatids not only in mouse[38] but also in humans[39] (Supplementary Table 9).

## Sex chromosomes

Sex chromosomes emerged twice in parallel in mammals from different sets of ancestral autosomes. The therian XY chromosome system originated just before the split of eutherians and marsupials (Fig. 4a) and hence has evolved largely independently in these two lineages. Around the same time, the monotreme sex-chromosome system arose from a different pair of autosomes and subsequently expanded to five XY pairs[40]. These sex-chromosome formation events entailed substantial remodelling of gene contents and expression patterns due to structural changes and sex-related selective forces[3]. We used our data to systematically assess testicular expression patterns of sex chromosomal genes and their evolution across mammals.

We first contrasted cell-type specificities of X-linked and autosomal genes, which showed a notable excess (12–60%) of X-linked genes with predominant expression in SG across all eutherians, in agreement with a previous mouse study[17] and includes conserved genes with key spermatogenic functions such as *TEX11* (Figs. 3a and 4b, Extended Data Fig. 9a and Supplementary Table 10). The higher-resolution primate data showed that the X chromosome is also enriched for genes expressed in leptotene SC (Fig. 4b), presumably reflecting transcript carry-over from SG, given the global transcriptional silencing of the genome during the leptotene stage[41] (Supplementary Fig. 1b and Supplementary Table 3) and the observation that most X-linked genes expressed in leptotene SC are also expressed in differentiated SG (Supplementary Fig. 4). Notably, we detected enrichments of genes with SG-specific expression also on the opossum and platypus X chromosomes, thus revealing this to be a shared pattern across mammals (Fig. 4b and Extended Data Fig. 9a,b).

We also uncovered an enrichment of genes with predominant expression in Sertoli cells across X chromosomes (except for platypus, for which Sertoli cells could not be unambiguously distinguished) (Fig. 4b). Altogether, our observations suggest that sex-related forces favoured the independent accumulation of SG- and Sertoli-specific genes on the X during the evolution of the different sex-chromosome systems. Consistently, autosomes in outgroup species corresponding to the different mammalian X chromosomes (for example, platypus chromosome 6, which is homologous to the therian X) do not show any excess of SG- or Sertoli-expressed genes (Extended Data Fig. 9a,c), which means that the ancestral autosomes that gave rise to present-day sex chromosomes were not enriched for such genes. Overall, we propose that the accumulation of SG- and Sertoli-specific genes was facilitated by the specific selective environment on the X chromosome, in which male-beneficial mutations are always visible to selection because of the single-copy (hemizygous) status of the X chromosome in males[3,17,21].

We next sought to separate X- and Y-bearing spermatids to investigate their distinct transcriptomal properties during spermiogenesis. Such an analysis is probably not possible using single-whole-cell transcriptomic data because X and Y spermatids remain connected by cytoplasmic bridges and hence are thought to contain similar cytoplasmic transcript pools[42]. However, our single-nucleus data should afford the separation of X and Y spermatids. Indeed, on the basis of differential X and Y transcript contents (Methods), we were able to separate spermatids into distinct X and Y lineages across mammals (Fig. 4c and Supplementary Table 3). As expected, our approach failed to separate X and Y spermatids in available human[23] and mouse[43] scRNA-seq datasets (Extended Data Fig. 10), supporting the notion of substantial transcript exchange across X- and Y-spermatid cells through cytoplasmic bridges[42], although this equilibration may not be complete[44].

A differential expression analysis between X and Y spermatids identified, as expected, most sex-chromosome genes, including gametologues (that is, genes with homologous counterparts on X and Y chromosomes), such as the translational regulatory genes

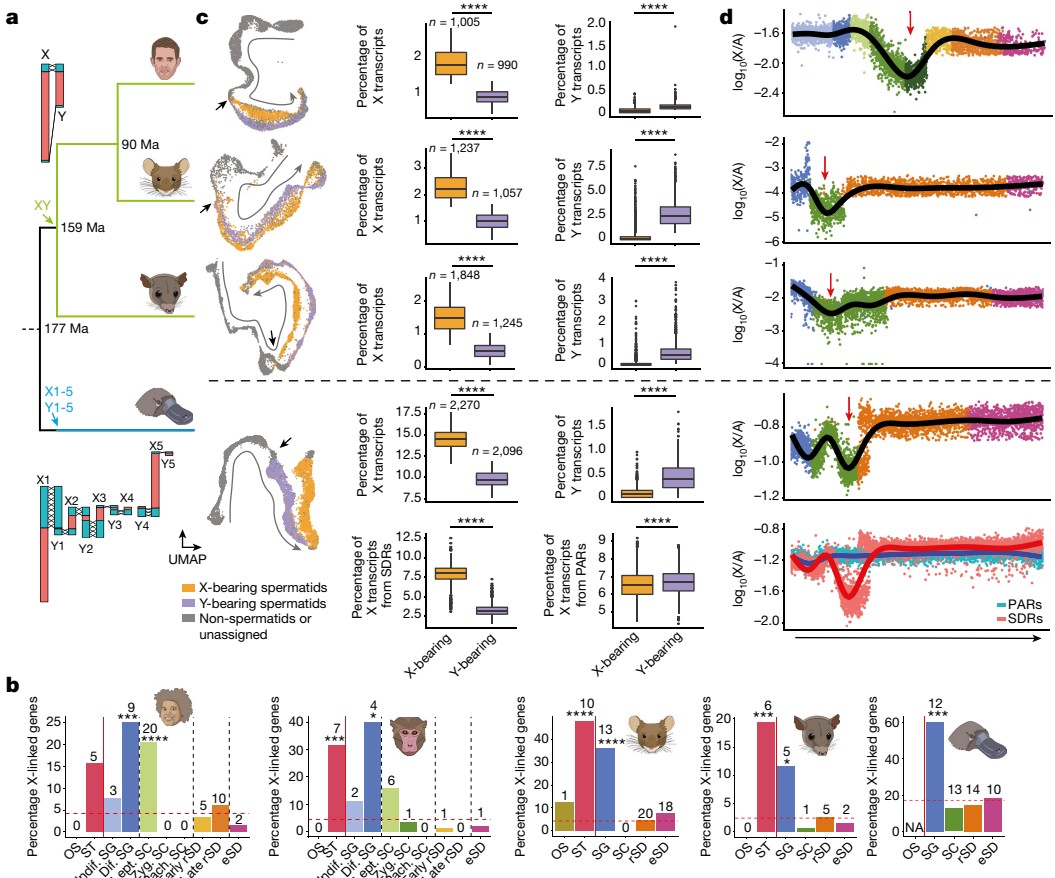

**Fig. 4 | Mammalian sex-chromosome evolution. a**, Phylogenetic tree for human, mouse, opossum and platypus (from top to bottom). Arrows show sex systems origination[40]. Illustration of therian (human, top) and monotreme (platypus, bottom) sex chromosomes. Recombining (crosses) PARs in turquoise and SDRs in red. **b**, Percentages of X-linked genes among testis-specific genes with predominant expression in a given cell type for human, macaque, mouse, opossum and platypus (from left to right; Supplementary Table 10). The red dashed line represents the expected percentages of X-linked genes, if testis-specific genes with predominant expression in the different cell types were randomly distributed across the genome. Asterisks indicate significance after two-sided exact binomial testing (Benjamini–Hochberg corrected *P* values from left to right: 0.000353, $1.18 \times 10^{-7}$, 0.000687, 0.0142, $1.08 \times 10^{-7}$, $3.45 \times 10^{-8}$, 0.000792, 0.0355 and 0.00043). The number of testis-specific X-linked genes

enriched in each cell type is indicated above each bar. **c**, UMAP representation of germ cells (left). Progression of spermatogenesis (grey arrow) and meiotic divisions (black arrows) are indicated. Spermatids identified as X- and Y-bearing are coloured in orange and purple, respectively. Box plots show the median (central value) percentages of X and Y transcripts in X- and Y-bearing spermatids; upper and lower quartile (box limits) with whiskers at 1.5 times the interquartile range. Two-sided Wilcoxon rank-sum tests were performed for statistical comparisons (Benjamini–Hochberg corrected ****$P < 2.2 \times 10^{-16}$). **d**, X-to-autosome transcript ratios in individual germ cells across spermatogenesis (from left to right). For spermatids, only X-bearing cells are considered. Lines depict generative additive model trend. Red arrows indicate MSCI. **c**,**d**, For platypus, X transcripts are dissected according to their location on the X chromosomes (Methods).

*DDX3X/DDX3Y* (Extended Data Fig. 11 and Supplementary Table 11). However, we also found autosomal genes, especially in some species (for example, human), which might reflect trans-regulatory effects associated with the X and Y chromosomes (Extended Data Fig. 11, Supplementary Table 11 and Methods).

We then assessed gene expression across spermatogenesis separately for X- and Y-linked genes. We traced a substantial dip in X transcript abundances around the pachytene stage of meiosis across therians (Fig. 4d), which reflects the process of MSCI[13], a sex-chromosome-specific instance of the general epigenetic phenomenon of meiotic silencing of unsynapsed chromatin[13] (MSUC). An analysis of Y transcripts provides consistent results but at lower resolution owing to the very small number of Y-linked genes (Extended Data Fig. 12a). Previous work did not find evidence for MSCI in monotremes[45], suggesting that MSCI originated in the therian ancestor after the separation from the monotreme lineage[45]. We revisited this question in platypus using a new assembly and annotation[46] that includes a detailed definition of sexually

differentiated regions (SDRs) versus pseudoautosomal regions (PARs), which are large in monotremes (Fig. 4a). We proposed that expression signals from the large PARs, which are expected to synapse and hence not to be affected by MSUC, might have prevented the detection of MSCI in previous studies.

Indeed, whereas the joint analysis of all platypus X-linked genes only shows a small expression dip around the pachytene stage (Fig. 4d, upper platypus graph), an analysis only of SDR genes reveals a strong reduction of X transcript levels. By contrast, PAR genes show stable expression levels across spermatogenesis (Fig. 4d, lower graph). Moreover, the difference in transcript abundances between SDRs and PARs owing to MSCI is visible for all five platypus X chromosomes (Extended Data Fig. 12b). Notably, our assessment of the completeness of MSCI across species reveals that platypus MSCI is as complete as that in other species for which there is little or no MSCI escape[23,47,48] (Extended Data Fig. 12c). The presence of MSCI at the SDRs in monotremes is consistent with the partial association of platypus sex

chromosomes with perinucleolar repressive histone modifications at the pachytene stage[45]. Altogether, our data reveal that efficient MSCI is common to all mammalian sex-chromosome systems, which indicates that the general mechanism of MSUC is an ancestral mammalian feature.

## Discussion

Our analyses uncovered that the previously observed rapid evolution of the testis[1–7] is driven by an accelerated rate of molecular innovation in late spermatogenesis, in particular in rSD. Our findings suggest a scenario in which the accelerated fixation of regulatory changes, amino acid altering substitutions and new genes during evolution in late spermatogenesis (presumably driven by sperm competition) was facilitated by reduced pleiotropic constraints, a transcriptionally permissive chromatin environment and potentially haploid selection for spermatid genes that are not rendered effectively diploid through transcript exchange through cytoplasmic bridges. In agreement with late spermatogenic stages explaining the rapid evolution of the testis, early spermatogenic cell types and somatic cells show patterns of constraint and innovation similar to those of cell types in the brain (Extended Data Fig. 3a,b,d), an organ that evolves slowly at the molecular level[1–3]. We note that differences in cell-type abundances across mammals, which are pronounced for the testis[7] (Extended Data Fig. 1f), presumably also contributed to the rapid gene expression divergence of this organ[6].

Our cross-species comparisons of individual genes revealed temporal expression differences across species, which were probably facilitated by reduced pleiotropic constraints. Our results thus provide an extensive list of candidates whose contributions to the evolution of species-specific spermatogenesis phenotypes can be experimentally scrutinized. We also uncovered conserved expression programs underlying spermatogenic processes ancestral to individual mammalian lineages and mammals as a whole.

Further analyses illuminated the role of sex chromosomal genes in spermatogenesis. We found that genes predominantly expressed in SG and Sertoli cells independently accumulated on X chromosomes across mammals and their two sex-chromosome systems during evolution. This suggests that X chromosomes have been shaped by strong male-related selective forces[21], leading to the emergence of X-linked genes with functional roles in testis cell types in which active transcription is possible. Indeed, in addition to the SG and Sertoli-cell enrichment found here, previous work showed that these selective pressures led to the repeated duplication of genes on the X that facilitated expression after meiosis[15].

Our ability to separate X- and Y-bearing spermatids and the availability of a new platypus genome assembly[46] unveiled that MSCI is a general feature of mammalian sex-chromosome systems, indicating that MSUC was already present in the common mammalian ancestor. Previous work did not find evidence for MSCI in birds[49], which raises the question whether MSUC arose in the common mammalian ancestor after the split from the reptile lineage roughly 310 million years ago or whether this mechanism was lost in the avian lineage and arose earlier in evolution. The latter scenario would be consistent with the observation of MSCI in invertebrate species[13].

Our data and results, together with the accompanying online resource we developed (https://apps.kaessmannlab.org/SpermEvol/) to facilitate the exploration of our data, provide an extensive resource for investigating the biology of the testis and associated fertility disorders across mammals. Future studies should seek to complement our snRNA-seq data to overcome its limitations. scRNA-seq data will be valuable for inferring transcriptome patterns unique to the cytoplasm, single-cell full-length transcript data are needed to assess the pronounced isoform diversity of the testis[11], and single-cell translatome data[50] are required to understand the contribution of post-transcriptional changes[6] to the evolution of spermatogenesis.

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

[1]Center for Molecular Biology (ZMBH), DKFZ-ZMBH Alliance, Heidelberg University, Heidelberg, Germany. [2]INRAE, LPGP, Rennes, France. [3]Department of Growth and Reproduction, Rigshospitalet, Copenhagen University Hospital, Copenhagen, Denmark. [4]International Center for Research and Research Training in Endocrine Disruption of Male Reproduction and Child Health, Rigshospitalet, Copenhagen University Hospital, Copenhagen, Denmark. [5]Bioinformatics Research Centre, Aarhus University, Aarhus, Denmark. [6]Berlin Institute of Health at Charité, University of Medicine Berlin, Corporate Member of the Free University of Berlin, Humboldt-University of Berlin, Berlin, Germany. [7]Evolutionary Developmental Biology Laboratory, Francis Crick Institute, London, UK. [8]Biomedical Primate Research Center (BPRC), Rijswijk, the Netherlands. [9]German Primate Center (DPZ), Platform Degenerative Diseases, Göttingen, Germany. [10]German Center for Cardiovascular Research (DZHK), Partner Site Göttingen, Göttingen, Germany. [11]Center for Neurobiology and Brain Restoration, Skolkovo Institute of Science and Technology, Moscow, Russia. [12]Max Planck Institute for Evolutionary Anthropology, Leipzig, Germany. [13]Institute of Evolutionary Biology (UPF-CSIC), Barcelona, Spain. [14]Catalan Institution of Research and Advanced Studies (ICREA), Barcelona, Spain. [15]CNAG-CRG, Centre for Genomic Regulation (CRG), Barcelona Institute of Science and Technology (BIST), Barcelona, Spain. [16]Miquel Crusafont Catalan Institute of Paleontology, Autonomous University of Barcelona, Barcelona, Spain. [17]The Robinson Research Institute, School of Biological Science, University of Adelaide, Adelaide, South Australia, Australia. [18]Department of Cellular and Molecular Medicine, Faculty of Health and Medical Sciences, University of Copenhagen, Copenhagen, Denmark. [19]These authors contributed equally: Florent Murat, Noe Mbengue. [✉]e-mail: florent.murat@inrae.fr; n.mbengue@zmbh.uni-heidelberg.de; h.kaessmann@zmbh.uni-heidelberg.de

# Methods

## Data reporting

No statistical methods were used to predetermine sample size. The experiments were not randomized and investigators were not blinded to allocation during experiments and outcome assessment.

## Biological samples and ethics statement

We generated snRNA-seq data for adult testis samples from human (*Homo sapiens*), chimpanzee (*Pan troglodytes*), bonobo (*Pan paniscus*), gorilla (*Gorilla gorilla*), lar gibbon (*Hylobates lar*), rhesus macaque (*Macaca mulatta*), common marmoset (*Callithrix jacchus*), mouse (*Mus musculus*, strain: RjOrl:SWISS; Janvier Laboratories), grey short-tailed opossum (*Monodelphis domestica*), platypus (*Ornithorhynchus anatinus*) and chicken (red jungle fowl, *Gallus gallus*) (Supplementary Table 2). In addition, we produced bulk RNA-seq data for chimpanzee, gorilla, gibbon and marmoset from the same individuals. Adult human testis samples used for in situ hybridization experiments were obtained from orchiectomy specimens from three individuals with testicular cancer. Tissue adjacent to the tumour that was devoid of cancer cells and germ cell neoplasia in situ, and tubules with normal spermatogenesis were used. Other adult primate testis tissue was obtained from a western chimpanzee and a Bornean orangutan (*Pongo pygmaeus*).

Our study complies with all relevant ethical regulations with respect to both human and other species' samples. Human samples underlying the snRNA-seq data were obtained from scientific tissue banks (http://medschool.umaryland.edu/btbank/) or dedicated companies (https://www.tissue-solutions.com/); informed consent was obtained by these sources from donors before death or from next of kin. The samples used for the RNA in situ hybridization experiments were obtained from the tissue biobank at the Department of Growth and Reproduction (Rigshospitalet, Copenhagen, Denmark) containing orchiectomy specimens from individuals with testicular cancer (The Danish Data Protection Agency, permit number J. no. 2001-54-0906). All patients have given informed consent for donating the residual tissues for research. The use of all human samples for the type of work described in this study was approved by an ethics screening panel from the European Research Council (ERC) and local ethics committees: from the Cantonal Ethics Commission in Lausanne (authorization 504/12); from the Ethics Commission of the Medical Faculty of Heidelberg University (authorization S-220/2017) and from the regional medical research ethics committee of the capital region of Copenhagen (H-16019637). All primates used in this study suffered sudden deaths for reasons other than their participation in this study and without any relation to the organ sampled. The use of all other mammalian samples for the type of work described in this study was approved by ERC ethics screening panels.

## Nuclei isolation

For the samples from therian species we developed a nuclei preparation method that includes fixation with dithio-*bis*(succinimidyl propionate) (DSP; or Lomant's Reagent), a reversible cross-linker that stabilizes the isolated nuclei. The method was adapted from protocols used for the fixation of single-cell suspensions[51] and for the isolation of single nuclei from archived frozen brain samples[52]. Tissue pieces weighing roughly 5 mg were homogenized in 100–150 μl mg$^{-1}$ of prechilled lysis buffer (250 mM sucrose, 25 mM KCl, 5 mM MgCl$_2$, 10 mM HEPES pH 8, 1% BSA, 0.1% IGEPAL and freshly added 1 μM DTT, 0.4 U μl$^{-1}$ RNase Inhibitor (New England BioLabs), 0.2 U μl$^{-1}$ SUPERasIn (ThermoFischer Scientific)) and lysed for 5 min on ice. The lysate was centrifuged at 100*g* for 1 min at 4 °C. The supernatant was transferred to a new reaction tube and centrifuged at 500*g* for 5 min at 4 °C. The supernatant was removed and the pellet resuspended in 0.67 vol. (of volume lysis buffer used) of freshly made fixation solution (1 mg ml$^{-1}$ DSP in PBS) and incubated for 30 min at room temperature. The fixation was quenched by addition of Tris-HCl to a final concentration of 20 mM. The fixed nuclei were pelleted at 500*g* for 5 min at 4 °C. The supernatant was removed and the pellet resuspended in 0.67 vol. of Wash Buffer (250 mM sucrose, 25 mM KCl, 5 mM MgCl$_2$, 10 mM Tris-HCl pH 8, 1% BSA and freshly added 1 μM DTT, 0.4 U μl$^{-1}$ RNase Inhibitor, 0.2 U μl$^{-1}$ SUPERasIn). This was centrifuged at 500*g* for 5 min at 4 °C. The supernatant was removed and the pellet resuspended in 0.5 vol. of PBS. Then nuclei were strained using 40 μm Flowmi strainers (Sigma). For estimation of nuclei concentration, Hoechst DNA dye was added and the nuclei were counted using Countess II FL Automated Cell Counter (ThermoFischer Scientific).

For platypus and chicken, a similar preparation method was used, but the nuclei were not fixed, given that this protocol gave optimal results for these species (the fixation protocol failed to yield data of adequate quality). In brief, tissue pieces weighing roughly 5 mg were homogenized in 100–150 μl mg$^{-1}$ of prechilled lysis buffer (250 mM sucrose, 25 mM KCl, 5 mM MgCl$_2$, 10 mM Tris-HCl pH 8, 1% BSA, 0.1% IGEPAL and freshly added 1 μM DTT, 0.4 U μl$^{-1}$ RNase Inhibitor, 0.2 U μl$^{-1}$ SUPERasin) and lysed for 5 min on ice. The lysate was centrifuged at 100*g* for 1 min at 4 °C. The supernatant was transferred to a new reaction tube and centrifuged at 500*g* for 5 min at 4 °C. The supernatant was removed and the pellet resuspended in 0.67 vol. of Wash Buffer. This was centrifuged at 500*g* for 5 min at 4 °C. The supernatant was removed and the pellet resuspended in 0.5 vol. of PBS. Then nuclei were strained using 40-μm Flowmi strainers (Sigma). For estimation of nuclei concentration, Hoechst DNA dye was added and the nuclei were counted using Countess II FL Automated Cell Counter (ThermoFischer Scientific).

## Preparation of sequencing libraries

For the construction of snRNA-seq libraries, Chromium Single Cell 3′ Reagent Kits (10X Genomics; v.2 chemistry for human, chimpanzee, bonobo, gorilla, gibbon, macaque, marmoset, mouse and opossum; v.3 chemistry for platypus and chicken) were used according to the manufacturer's instructions. Then 15,000 to 20,000 nuclei were loaded per lane in the Chromium microfluidic chips and complementary DNA was amplified in 12 PCR cycles. Sequencing was performed with NextSeq550 (Illumina) according to the manufacturer's instructions using the NextSeq 500/550 High Output Kit v.2.5 (75 cycles) with paired-end sequencing (read lengths of read1 26 bp, read2 57 bp; index1 8 bp, roughly 170 to 380 million reads per library for v.2 chemistry; read lengths of read1 28 bp, read2 56 bp Index1 8 bp, roughly 247 to 306 million reads per library for v.3 chemistry) (Supplementary Table 2).

For bulk RNA-seq data generation, RNA was extracted using the RNeasy Micro kit (QIAGEN). The tissues were homogenized in RLT buffer supplemented with 40 mM DTT. The RNA-seq libraries were constructed using the TruSeq Stranded messenger RNA LT Sample Prep Kit (Illumina) as described in ref. [2]. Libraries were sequenced on Illumina NextSeq550 using a single-end run (read1 159 bp; index1 7 bp) with roughly 24–60 million reads per library (Supplementary Table 2).

## Genome and transcript isoform annotation

Given that the quality of genome annotation differs substantially between the studied species and given the specific and widespread transcription of the genome in the testis[11], we refined and extended previous annotations from Ensembl[53] on the basis of testis RNA-seq data. Specifically, akin to the procedure described in refs. [2,6], we used previous extensive stranded poly(A)-selected RNA-seq datasets[2,54] (100 nt, single-end) for human, macaque, mouse, opossum, platypus and chicken, and generated and used stranded poly(A)-selected RNA-seq datasets (159 nt, single-end) for chimpanzee, gorilla, gibbon and marmoset. For each species, we downloaded the reference genome from Ensembl release 87 (ref. [53]): hg38 (human), CHIMP2.1.4 (chimpanzee), rheMac8 (rhesus macaque), C_jacchus3.2.1 (marmoset), mm10 (mouse), monDom5 (opossum), and galGal5 (chicken); from Ensembl release 96 (ref. [55]): gorGor4 (gorilla) and Nleu_3.0 (gibbon); and from Ensembl release 100 (ref. [56]): mOrnAna1.p.v1 (platypus).

Raw reads were first trimmed with cutadapt v.1.8.3 (ref. [57]) to remove adapter sequences and low-quality (Phred score <20) nucleotides, then reads shorter than 50 nt were filtered out (parameters: --adapter= AGATCGGAAGAGCACACGTCTGAACTCCAGTCAC --match-read-wildcards --minimum-length=50 -q 20). Processed reads were then mapped to the reference transcriptome and genome using Tophat2 v.2.1.1 (ref. [58]) (parameters: --bowtie1 --read-mismatches 6 --read-gap-length 6 --read-edit-dist 6 --read-realign-edit-dist 0 --segment-length 50 --min-intron-length 50 --library-type fr-firststrand --max-insertion-length 6 --max-deletion-length 6). Next, we assembled models of transcripts expressed using StringTie v.1.3.3 (ref. [59]) (parameters: -f 0.1 -m 200 -a 10 -j 3 -c 0.1 -v -g 10 -M 0.5). Stringent requirements on the number of reads supporting a junction (-j 3), minimum gap between alignments to be considered as a new transcript (-g 10) and fraction covered by multi-hit reads (-M 0.5) were used to avoid merging independent transcripts and to reduce the noise caused by unspliced or incompletely spliced transcripts. We compared the assembled transcript models to the corresponding reference Ensembl annotations using the cuffcompare program v.2.2.1 from the cufflinks package[60]. Finally, we combined the newly identified transcripts with the respective Ensembl gene annotation into a single gtf file. We extended the original Ensembl annotations by 2–61 Mbp with new transcripts and by 23–49 Mbp with new splice isoforms (Supplementary Table 1).

## Raw reads processing

CellRanger v.3.0.2 was used for platypus and chicken, and CellRanger v.2.1.1 for the other species in line with the used Chromium chemistry. The CellRanger mkref function was used with default settings to build each species reference from genomic sequences and customized extended annotation files (Supplementary Table 1). Given that pre-mRNA transcripts are abundant in nuclei[61], exons and introns features were concatenated as described in the CellRanger v.2.1.1 documentation for considering intronic and exonic reads for gene expression quantification. The CellRanger count function was used with default settings to correct droplet barcodes for sequencing errors, align reads to the genome and count the number of UMIs for every gene and barcode combination.

## Identification of usable nuclei

We used a combined approach for detection of usable nuclei. This was done to optimally account for the lower RNA content of nuclei compared to the whole cells. Specifically, to identify usable nuclei, we used a knee point-based approach combined with the fraction of intronic reads as a marker of pre-mRNA transcripts (abundant in the nucleus) (Supplementary Fig. 1f) and *MALAT1* (nuclear-enriched long non-coding (lnc)RNA) expression as a marker of nuclei (when present in the genome of a given species).

## Quality control of filtered cells

For each sample independently, high-quality nuclei were selected removing outliers on the basis of the number of UMIs and the percentage of mitochondrial RNA (Supplementary Fig. 1b,c,e). We created a Seurat[62] object using the Seurat R package v.3.1.4 from the subset raw UMI count table generated by CellRanger corresponding to the usable droplets identified upstream, normalized the data using the Normalize-Data function, identified the top 10,000 most variables genes using the FindVariableFeatures function, scaled the data using the ScaleData function, performed the PCA using the RunPCA function and calculated the Louvain clusters using the FindNeighbors (parameters dims = 1:20) and FindClusters (dims = 1:20, resolution = 0.5) functions. To optimally account for the fact that testis cell types have diverse transcriptome characteristics[11], we filtered out outlier droplets for each cluster independently with values lower than the first quartile (Q1) − 1.5 × IQR (interquartile range) and higher than the third quartile (Q3) + 1.5 × IQR for both the UMI content and the fraction of mitochondrial RNA. Then,

we removed potential doublets using doubletFinder_v3 function of DoubletFinder[63] v.2.0.1 (parameters PCs = 1:20, pN = 0.25, nExp = 5% of the total number of cells, identifying pk using paramSweep_v3, summarizeSweep and find.pK functions).

## Integration of datasets

From the previously filtered UMI count tables, we created Seurat objects for every sample independently, normalized the data and identified the top 10,000 most variable genes. Next, for each species independently, we applied the Seurat[62] anchoring approach using FindIntegrationAnchors and IntegrateData functions with 20 principal components to integrate all datasets together into a single Seurat object correcting for the batch effect. For each integrated species-specific Seurat object, we normalized (NormalizeData function) and scaled the data (ScaleData function) and performed a PCA (RunPCA function). Louvain clusters were calculated using Find-Neighbors and FindClusters functions (parameters dims = 1:20, 1:20, 1:20, 1:20, 1:20, 1:20, 1:20, 1:17, 1:8, 1:10 and 1:10, and resolution = 0.5, 0.5, 0.2, 0.5, 0.5, 0.5, 0.5, 0.5, 0.5, 0.3 and 0.5, respectively, for human, chimpanzee, bonobo, gorilla, gibbon, macaque, marmoset, mouse, opossum, platypus and chicken). The uniform manifold approximation and projection (UMAP) embedding coordinates were calculated using the RunUMAP function (parameters dims = 1:20, 1:20, 1:20, 1:20, 1:20, 1:20, 1:20, 1:17, 1:10, 1:10 and 1:10, and min_dist = 0.3, 0.3, 0.1, 0.1, 0.3, 0.3, 0.3, 0.1, 0.2, 0.3 and 0.6, respectively, for human, chimpanzee, bonobo, gorilla, gibbon, macaque, marmoset, mouse, opossum, platypus and chicken). We note that—consistent with the high correlation between biological replicates (Supplementary Fig. 5a,b)—the data already integrate well before the batch correction (Supplementary Fig. 18c,d). We also note that key marker genes are expressed in the same integrated areas across replicates when assessing their expression in the different replicates using the integrated object coordinates (Supplementary Fig. 18e), which supports that the integration is correct.

All primate datasets were merged using the LIGER[64] (v.0.5.0) integration tool. A LIGER object was created using the createLiger function based on primate 1:1 orthologues from Ensembl release 87, and normalized with normalize, selectGenes and scaleNotCenter functions with default settings. Then, the joint matrix was factorized using the optimizeALS function ($k = 20$) and the quantile normalization was performed with the quantile_norm and default settings. The Louvain clusters were calculated with the louvainCluster function and default settings as well as UMAP coordinates with the runUMAP function (n_neighbors = 100, min_dist = 0.2).

## Estimation of expression levels and normalization

The gene UMI counts per cell were normalized using the Seurat R package and its NormalizeData function. Therefore, the UMI counts of each gene in each cell are divided by the total UMI counts of each cell, multiplied by 10,000 and log transformed.

## Cell-type assignment

We identified the main cell-type populations from the primate integrated, mouse, opossum, platypus and chicken objects independently using known marker genes[65,66] mostly from human and mouse and their respective 1:1 orthologues in the other species. *CLU* marks Sertoli cells; *TAGLN* and *ACTA2* peritubular and smooth muscle cells; *CD34* and *TM4SF1* endothelial cells; *APOE* and *CD74* macrophages; *STAR* and *CYP11A1* Leydig cells; *GFRA1, PIWIL4* (undifferentiated), *DMRT1* (differentiated) and *STRA8* SG; *SYCE1* (leptotene), *SYCP1* (zygotene), *PIWIL1* (pachytene), *SYCP2, TANK* and *AURKA* SC; *LRRIQ1* (early), *ACRV1* and *SPACA1* (late) rSD; and *SPATA3, NRBP1, PRM1* and *GABBR2* eSD. Cell-type assignment was robustly reinforced by complementary analyses and metrics such as UMAP coordinates, pseudotime trajectories, transcriptional activities (UMI counts) and previous knowledge.

## Pseudotime

Pseudotime trajectories were calculated using the slingshot v.1.2.0 R package[67]. We applied the getLineages function with the upstream calculated clusters and UMAP embedding coordinates of the germ cells to obtain connections between adjacent clusters using a minimum spanning tree. We provided the starting and ending clusters on the basis of the previous cell-type assignment with known marker gene expression. Then we applied the getCurves function to the obtained lineages to construct smooth curves and order the cells along a pseudotime trajectory. Pseudotime values are highly variable depending on used tools, thus we ordered the cells one by one on the basis of their pseudotime values and divided their rank by the total number of cells, to obtain evenly distributed values between 0 and 1. Finally, we validated the obtained pseudotime trajectories on the basis of previous cell-type assignments, expression patterns of marker genes and UMAP embedding coordinates.

## Marker gene identification

To precisely identify marker genes along spermatogenesis, we grouped the germ cells into 20 evenly distributed bins along the pseudotime trajectory for each species. Then, we applied the FindAllMarkers function from the Seurat[62] R package (parameter only.pos = T, min.pct = 0.25, logfc.threshold = 0.25, return.thresh = 0.05) of the Seurat v.3.1.4 R package to the 22 groups (20 germline groups, the Sertoli and other somatic cell groups) (Extended Data Fig. 1 and Supplementary Table 4).

## Phylogenetic trees

Phylogenetic trees and indicated divergence times (Figs. 1a and 4a) are based on TimeTree[68] (v.5) (http://www.timetree.org/).

## Orthologous gene sets

We used four different sets of orthologous genes in our study: (1) comparative analyses involving all 11 amniote species were performed using 4,498 1:1 orthologue genes that are expressed (that is, one UMI in at least three cells of any cell type) across all species (among a total of 8,045 1:1 orthologues). (2) Comparative analyses involving the seven primate species were performed using 8,451 1:1 orthologue genes expressed across all primate species (among 11,948 1:1 orthologues). (3) The comparative Sertoli-germ cell communication analysis was based on mapping 35,186 human testis-expressed genes to 1:1 orthologous genes in the other species (macaque 13,090; mouse 14,302; opossum 10,865 and chicken 10,515). (4) Species-specific analyses were performed using all genes expressed in a given species (roughly 15,000 genes per cell type; Supplementary Fig. 1d). Orthologous gene sets were extracted from Ensembl[53] using the biomaRt R package v.2.40.5.

## Global patterns of gene expression differences across mammals

Pseudo-bulk samples were generated using the AverageExpression function of the Seurat R package with various groups of cells depending on the pseudo-bulk samples produced in the study. For the analyses presented in Fig. 1b, we performed the PCA of normalized expression in amniote testicular cell types (pseudo-bulks) for each individual based on 4,498 1:1 amniote orthologues. PCA was performed using the prcomp function of the stats R package. For Fig. 1c, we constructed gene expression trees (as described in ref. [1]) using the neighbour-joining approach, on the basis of pairwise expression distance matrices between corresponding pseudo-bulk samples for the different cell types across species. The distance between samples was computed as $1 - \rho$, where $\rho$ is Spearman's correlation coefficient and was computed using the cor function of the stats R package. The neighbour-joining trees were constructed using the ape R package v.5.3. The reliability of branching patterns was assessed with bootstrap analyses (the 4,498 1:1 amniote orthologues were randomly sampled with replacement 1,000 times). The bootstrap values are the proportions of replicate trees that share the branching pattern of the majority-rule consensus tree shown in the figures (Fig. 1c and Extended Data Fig. 2a). The total tree length was calculated by removing the intra-species variability between individuals (Fig. 2a).

## Evolutionary forces

In Fig. 2c, we plotted the median pLI score[69] across expressed genes ($\geq$ 1 UMI) in each nucleus. We obtained the pLI scores from ref. [70]. For Fig. 2d, we used a set of neutrally ascertained knockouts consisting in 4,742 protein-coding genes, 1,139 of which are classified as lethal. For each cell, the denominator is the number of genes expressed that were tested for lethality and the numerator the genes among those that resulted in a lethal phenotype. Tested genes for viability and associated phenotype information were downloaded from the International Mouse Phenotyping Consortium[25]. For Fig. 2e (and Extended Data Fig. 3a), we used the average d$N$/d$S$ values across 1:1 orthologues in primates. For each cell, the mean d$N$/d$S$ value is plotted. Conserved 1:1 orthologues across six primates (human, chimpanzee, gorilla, gibbon, macaque and marmoset) as well as their coding and protein sequences were extracted from Ensembl[53], providing a set of 11,791 protein-coding genes. For each species and orthologue the longest transcript was extracted. Orthologous protein sequences were aligned using clustalo v.1.2.4; then pal2nal v14 was used (with protein sequences alignments and coding sequences as input) to produce codon-based alignments. The codeml software from the PAML package[71] v.4.9 was used to estimate d$N$/d$S$ ratios. The M0 site model was applied to the orthologue alignments to estimate one average dN/dS ratio per orthologous gene set across species (parameter NSites = 0, model = 0). In Fig. 2f (and Extended Data Fig. 3b), we plotted the percentage of positively selected genes expressed across nuclei. For each nucleus, the denominator is the number of expressed genes that were tested for signatures of positive selection, and the numerator is the number of genes among those with evidence for positive selection. We used sets of genes previously identified as carrying evidence for coding-sequence adaptation in primates[72] (331 positively selected genes out of 11,170 genes tested) and mammals[73] (544 positively selected genes out of 16,419 genes tested).

In Fig. 2g (and Extended Data Fig. 3c), we plotted the average phylogenetic age of expressed genes across somatic and germ cells. The phylogenetic age of genes is an index that gives greater weight to young new genes (as described in ref. [2,74]). The range of the score differs between species depending on the number of outgroup lineages available (more lineages allowed for more details in the phylogeny) and therefore this index cannot be compared across species, only within species (that is, across cells and cell types). The phylogenetic age of genes was obtained from GenTree (http://gentree.ioz.ac.cn/) with Ensembl release 69 (ref. [74]). In Fig. 2h (and Extended Data Fig. 3d), we plotted the percentage of the cell transcripts originating from protein-coding genes and intergenic elements. Gene biotypes were obtained from Ensembl. Intergenic elements are all elements that are not protein-coding genes (lncRNAs, pseudogenes, pseudogenes and other sequences). In Fig. 2i (and Extended Data Fig. 3e), normalized log$_2$-transformed median expression values across replicates at the transcriptome ($e^{tr}$) and translatome ($e^{tl}$) layers were used to calculate translation efficiency (TE = log$_2$($e^{tr}$) − log$_2$($e^{tl}$)) in testis (as described in ref. [6]) from RNA-seq and Ribo-seq data[6], respectively. Translation efficiency values were calculated at the whole testis level, thus only cell-type-specific genes (for which 60% of their transcripts at the whole testis level are concentrated in a single cell type) were used. Higher values show a more efficient translation of transcripts, whereas lower values indicate relative translational repression. For Fig. 2j (and Extended Data Fig. 3f,g), we used time- and tissue-specificity indexes of expressed genes across somatic and germ cells in testis. As described in ref. [2], tissue and time specificity indexes are calculated from RNA-seq data across organs and developmental stages. Both indexes range

from 0 (broad expression) to 1 (restricted expression). The indexes were obtained from ref. [2]. For each nucleus, we plotted the median index across expressed genes. In Fig. 2k, we plotted the percentage of genes causing infertility when knocked out (out of 3,252 knockouts, 173 of which caused infertility). Tested genes for infertility and associated phenotype information were downloaded from the International Mouse Phenotyping Consortium database[25]. For each nucleus, the denominator corresponds to the number of genes expressed that were tested for infertility and the numerator to the genes among those that resulted in an infertility phenotype.

## Gene expression trajectories along spermatogenesis

We compared gene expression trajectories along spermatogenesis across primates using human, chimpanzee, bonobo, gorilla, gibbon, macaque and marmoset (based on Ensembl 87 orthologues), and across amniotes using human (as a representative of primates), mouse, opossum, platypus and chicken (based on Ensembl 100 orthologues). To compare robustly expressed genes, we used genes that are expressed in at least 5% of the cells in at least one cluster in all considered species. We used the mfuzz package[75] (v.2.44.0), an unsupervised soft clustering method, to cluster gene expression trajectories along spermatogenesis (eight cell types in primates; four cell types in amniotes) across species using Dmin and mestimate functions to estimate the number of clusters and the fuzzification parameter (Supplementary Fig. 2). As described in ref. [2], we inferred within a phylogenetic framework the probability that there were changes in trajectories along spermatogenesis, that is, that genes changed their cluster assignment in specific branches, using a 5% probability cut-off.

## Trajectory conservation score

We calculated a global trajectory conservation score across species for each 1:1 orthologous gene set. For a given orthologous gene set, this score corresponds to the log-transformed probability that all members fall into the same mfuzz trajectory cluster as:

$$\text{Conservation\_score}_g = \log_2\left(\sum_{i \in c} \prod_{j \in s} P_{g,i,j}\right)$$

where $g$ corresponds to a given orthologous gene set, $c$ to all mfuzz trajectory clusters (1–9 for primates, 1–12 for amniotes), $s$ to all species (human, chimpanzee, bonobo, gorilla, gibbon, macaque and marmoset for primates; human, mouse, opossum, platypus and chicken for amniotes) and $P_{g,i,j}$ to the probability that the gene $g$ of the species $j$ falls into the cluster $i$. A higher conservation score indicates a greater global trajectory conservation. As a proof of concept, we plotted the trajectory conservation score for conserved and changed trajectories that revealed a significant higher conservation score for conserved trajectories (Extended Data Fig. 5a).

## RNA in situ hybridization and expression quantification

Fresh testicular tissue was fixed in GR-fixative (7.4% formaldehyde, 4% acetic acid, 2% methanol, 0.57% sodium phosphate, dibasic and 0.11% potassium phosphate, monobasic) overnight (for at least 16 h) at 4 °C, dehydrated and embedded in paraffin. The in situ hybridization experiments were carried out on 4 μm sections mounted on SuperFrost Plus Slides (ThermoFisher Scientific) using the RNAScope 2.5 HD Detection Reagent RED kit according to the manufacturer's recommendations (Advanced Cell Diagnostics). Briefly, testicular tissue sections were dewaxed in xylene and washed in 100% ethanol followed by treatment with hydrogen peroxide for 10 min. Target retrieval was performed for 15 or 30 min (see Supplementary Table 6 for specifications for each probe and species) using a steamer, followed by treatment with protease plus for 30 min at 40 °C. The slides were hybridized with the target probe (Supplementary Table 6) for 2 h at 40 °C followed by a series of signal amplifications (with amplification around 5 for 30 or 60 min).

The sections were counterstained with Mayer's haematoxylin and mounted with Vectamount Permanent Mounting Medium (Vector Laboratories). The negative control probe *DapB* (a bacterial RNA) was run in parallel with the target probes and showed ≤5% positive cells in each section.

For *ADAMTS17* and *MYO3B*, positive (that is, with red dots) rSD and SG were counted. For each section (human $n = 3$, chimpanzee $n = 1$, orangutan $n = 1$), ten tubules were counted using the NDP.viewPlus software (Hamamatsu Photonics). Two independent observers (S.B.W. and K.A.) counted positive and negative rSD and SG. No discrimination in intensity of the dots or the number of dots per cell was performed. Cell-type identification was performed on the basis of nucleus morphology and localization in the tubule. Only SG lining the edge of the tubules were counted (Supplementary Fig. 3a). The inter-observer variance was found to be 7 and 9% for rSD and SG, respectively. For *RUBCNL*, quantification of staining intensity was performed with R v.3.6.1 using countcolors[76] (v.0.9.1) and colordistance[77] (v.1.1.1) packages. For each section (human $n = 3$, chimpanzee $n = 1$, orangutan $n = 1$), ten tubules were divided into three parts: area dominated by SG, area dominated by SCs (also containing Sertoli cells) and area dominated by spermatids (no distinction between the different types of spermatid) (Supplementary Fig. 3b). In each tubular area, the number of cells was counted manually using the NDP.view2Plus software (v.2.8.24). Then, the pixels occupied by red staining were quantified and the expression level for each cell type was calculated by dividing the stained pixels by the number of cells. For each picture, the stained pixels for each cell type were normalized by the total amount of stained pixels.

## Gene Ontology analysis

Enriched terms in the Gene Ontology[32] analyses of genes with conserved and diverged expression trajectories were identified using the goana function of the limma R package, v.3.40.6 (default parameters).

## Sertoli-germ cell communication analysis

We identified ligand–receptor interactions underlying Sertoli-germ cell communications for human, macaque, mouse, opossum and chicken, respectively, using the CellPhoneDB[78] (v.2) approach and recommended parameter settings (parameters method statistical_analysis). To apply CellPhoneDB, which uses a database of human ligand–receptor interactions, to the data for the other species, we mapped human testis-expressed genes to their corresponding 1:1 orthologues in each of these species. Enriched receptor–ligand interactions between two cell types are predicted on the basis of expression of a receptor by one cell type and a ligand by another cell type[78]. Only receptors and ligands expressed in more than 10% of the cells in each cell type were considered for pairwise comparisons between all cell types in the dataset. CellPhoneDB uses empirical shuffling to calculate which ligand–receptor pairs show significant ($P < 0.05$) cell-type specificity[78]. Significant interactions across species were illustrated using the R package UpSetR (v.1.4.0). Finally, given potential false positive (and negative) predictions of CellPhoneDB and similar approaches[79], we consider interactions that are predicted for several species and probably reflect evolutionary conservation (Extended Data Fig. 8 and Supplementary Table 9), such as those reported in the main text, to be more reliable than species-specific predictions.

## Cell-type and testis-specific genes per chromosome

Testis-specific genes were obtained from previously generated RNA-seq data[2] of adult organs (RPKM ≥1 in testis and RPKM < 1 in brain, cerebellum, heart, kidney and liver). Among these, cell-type-specific genes were studied for each chromosome. Genes with predominant expression in specific somatic cell types were identified using the FindAllMarkers function (parameter only.pos = TRUE, min.pct = 0.05, logfc. threshold = 0.25, return.thresh = 0.05). Predominant expression of genes in specific germ cell types was assigned on the basis of the trajectory analyses (above); that is, predominant expression was assigned

on the basis of the cell type in which the expression level of the gene peaks in the trajectory analysis. We then first calculated the percentage of genes located on a given chromosome among all genes in the genome (x axis of plots in Extended Data Fig. 9a,c; red horizontal line for X-linked genes in Fig. 4b). We then contrasted this with the percentage of testis-specific genes with predominant expression in a given cell type (y axis of plots in Extended Data Fig. 9a,c; y axis of plots in Fig. 4b for X-linked genes). Finally, the percentage of testis-specific genes per cell type and chromosome was statistically compared to the percentage of genes per chromosome in the genome using exact binomial tests.

### Classification of X- and Y-bearing spermatids

The Y chromosome carries a low number of genes and is missing in some genome assemblies. Thus, we focused on the fraction of X transcripts in spermatids to classify them as X- or Y-bearing cells. For this, we fitted a Gaussian Mixture Model to the data with two components (bimodal distribution) independently for each replicate, using the function normalmixEM of the mixtools (v.1.2.0) R package. The two obtained normal distributions were used to classify X- (higher levels of X transcripts) and Y-bearing (lower levels of X transcripts) spermatids using 95% confidence intervals. Outlier and overlapping cells were not assigned to either category. Finally, we checked that the fraction of Y transcripts was significantly higher in Y-bearing spermatids (Fig. 4c). Bifurcating UMAPs (Fig. 4c) were obtained using X- and Y-linked genes in addition to previously identified highly variable genes to perform the PCA associated with the UMAP coordinate calculation. For platypus, X transcripts are separated according to their location on the X chromosomes (that is, PARs and SDRs, respectively, as annotated in a previous study[46]). In the platypus genome assembly used[46], X and Y PARs are both assigned to the X chromosome. Thus, reported X transcripts may stem from X SDRs, X PARs or Y PARs, whereas reported Y transcripts only stem from Y SDRs. Illustrations of human and platypus sex chromosomes (with their respective PARs/SDRs) in Fig. 4a are based on previous work[46,80].

### Transcriptomal differences between X and Y spermatids

We identified differentially expressed genes between X- and Y-spermatid populations using the FindMarkers function from Seurat[62] R package (parameters, default). A Wilcoxon rank-sum test was used to calculate P values that were adjusted using Bonferroni corrections for several tests. Only genes that were detected as expressed in at least 10% of cells from either of the two populations were tested. Genes that show, on average, at least a 0.25-fold higher expression ($\log_2$-scale) in one of the populations, that are at the same time expressed in twice the number of cells in that population and that have an adjusted P value below 0.01 were considered to be differentially expressed. We note that several of the differentially expressed genes, including the most significant cases in human (Extended Data Fig. 11a), are putatively non-coding, which is noteworthy because lncRNAs are typically nuclear[81] and hence their differential expression levels are unlikely to be offset by transcript exchange between spermatid cells through cytoplasmic bridges. The three most Y-spermatid-specific transcripts are lncRNAs emanating from homologous low copy repeats (that is, segmental duplications) on chromosomes 13 (FAM230C), 21 (XLOC-095504) and 22 (FAM230F) that cause genomic disorders by triggering non-allelic homologous recombination events. They include the FAM230F lncRNA in the q11.2 low copy repeat region on chromosome 22 (22q11.2) that is particularly susceptible to non-allelic homologous recombination-generated deletions that lead to various congenital malformation disorders, including the DiGeorge syndrome, the most frequent microdeletion disorder[82].

### MSCI completeness analysis

To identify potential MSCI escapee genes, we screened for X-linked genes with a significant increase in transcript abundance from SG to SC stages subject to MSCI, which ensures that potential escape genes are indeed actively transcribed in SC (that is, they do not merely represent genes expressed in SG with stable transcripts still detectable in SC), akin to previous work[23]. Specifically, we identified differentially expressed genes between SC and SG using the FindMarkers function from Seurat[62] R package (parameters, default). A Wilcoxon rank-sum test was used to calculate P values, which were then adjusted using a Bonferroni correction for several tests. Only genes that were detected as expressed in at least 10% of cells from either of the two cell-type populations were tested. Genes showing, on average, at least a 0.25-fold expression difference ($\log_e$-scale) between the two groups, and an adjusted P value below 0.05 were considered to be differentially expressed. X-linked genes in SDRs with significantly higher expression in SC than SG were considered to be potential escapees (Extended Data Fig. 12c).

### General statistics and plots

Unless otherwise stated, all statistical analyses and plots were done in R v.3.6.2 (ref. [83]). Plots were created using ggplot2 v.3.2.1, tidyverse v.1.3.0, dplyr v.0.8.5, cowplot v.1.0.0 and pheatmap v.1.0.12.

### Reporting summary

Further information on research design is available in the Nature Portfolio Reporting Summary linked to this article.

## Data availability

Raw and processed bulk and snRNA-seq data have been deposited in ArrayExpress with the accession codes E-MTAB-11063 (human snRNA-seq), E-MTAB-11064 (chimpanzee snRNA-seq), E-MTAB-11067 (bonobo snRNA-seq), E-MTAB-11065 (gorilla snRNA-seq), E-MTAB-11066 (gibbon snRNA-seq), E-MTAB-11068 (macaque snRNA-seq), E-MTAB-11069 (marmoset snRNA-seq), E-MTAB-11071 (mouse snRNA-seq), E-MTAB-11072 (opossum snRNA-seq), E-MTAB-11070 (platypus snRNA-seq), E-MTAB-11073 (chicken snRNA-seq) and E-MTAB-11074 (chimpanzee, gorilla, gibbon and marmoset bulk RNA-seq) (https://www.ebi.ac.uk/arrayexpress/). All other data are available as Supplementary Information or available upon request. The testis gene expression at the single-nucleus level across the 11 studied species can be visualized using the shiny app we developed: https://apps.kaessmannlab.org/SpermEvol/.

## Code availability

Custom scripts used to generate the results reported in the paper and processed data are available at https://github.com/evo-bio/Spermatogenesis.

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

**Acknowledgements** We thank all members of the Kaessmann group, S. Tirier for discussions and N. Trost for the administration of the Kaessmann laboratory server. Computations were performed on the Kaessmann laboratory server and the bwForCluster from the Heidelberg University Computational Center (supported by the state of Baden-Württemberg through bwHPC and the German Research Foundation grant no. INST 35/1134-1 FUGG). This research was supported by grants from the ERC (grant no. 615253, OntoTransEvol) and German Research Council (DFG, grant nos. SFB 873 and KA 1710/4-1) to H.K., by the CellNetworks Postdoc Fellowship and EMBO Long-Term Fellowship to F.M. (grant no. ALTF 591-2017), and by the Australian Research Council (grant no. FT160100267) to F.G. and by the Novo Nordisk Foundation (grant no. NNF21OC0069913) to K.A. and (grant no. NNF18OC0031004) to M.H.S. The use of all other mammalian samples for the type of work described in this study was approved by ERC ethics screening panels (ERC starting grant no. 242597, SexGenTransEvolution and ERC consolidator grant no. 615253, OntoTransEvol).

**Author contributions** F.M., N.M and H.K. conceived and organized the study based on H.K.'s original design. F.M., N.M. and H.K. wrote the manuscript, with input from all authors. F.M. performed all analyses, developed the shiny app and drew the species icons. K.M. and T.B. collected the samples. R. Behr, P.K, F.G, S.P., I.K., R. Bontrop, T.M.-B., K.A. and M.H.S. provided samples. F.M., N.M., T.T. and M.S. established snRNA-seq methods. N.M. performed the snRNA-seq experiments with support from J.S. C.S., K.M. and T.B. S.B.W. performed the smISH experiments, and analysed the results together with K.A. E.L. performed the testis genomic annotations. M.S., M.C.-M, F.L., M.R.B., C.C., K.A. and M.H.S. provided useful feedback and discussions.

**Competing interests** The authors declare no competing interests.

**Additional information**
**Correspondence and requests for materials** should be addressed to Florent Murat, Noe Mbengue or Henrik Kaessmann.

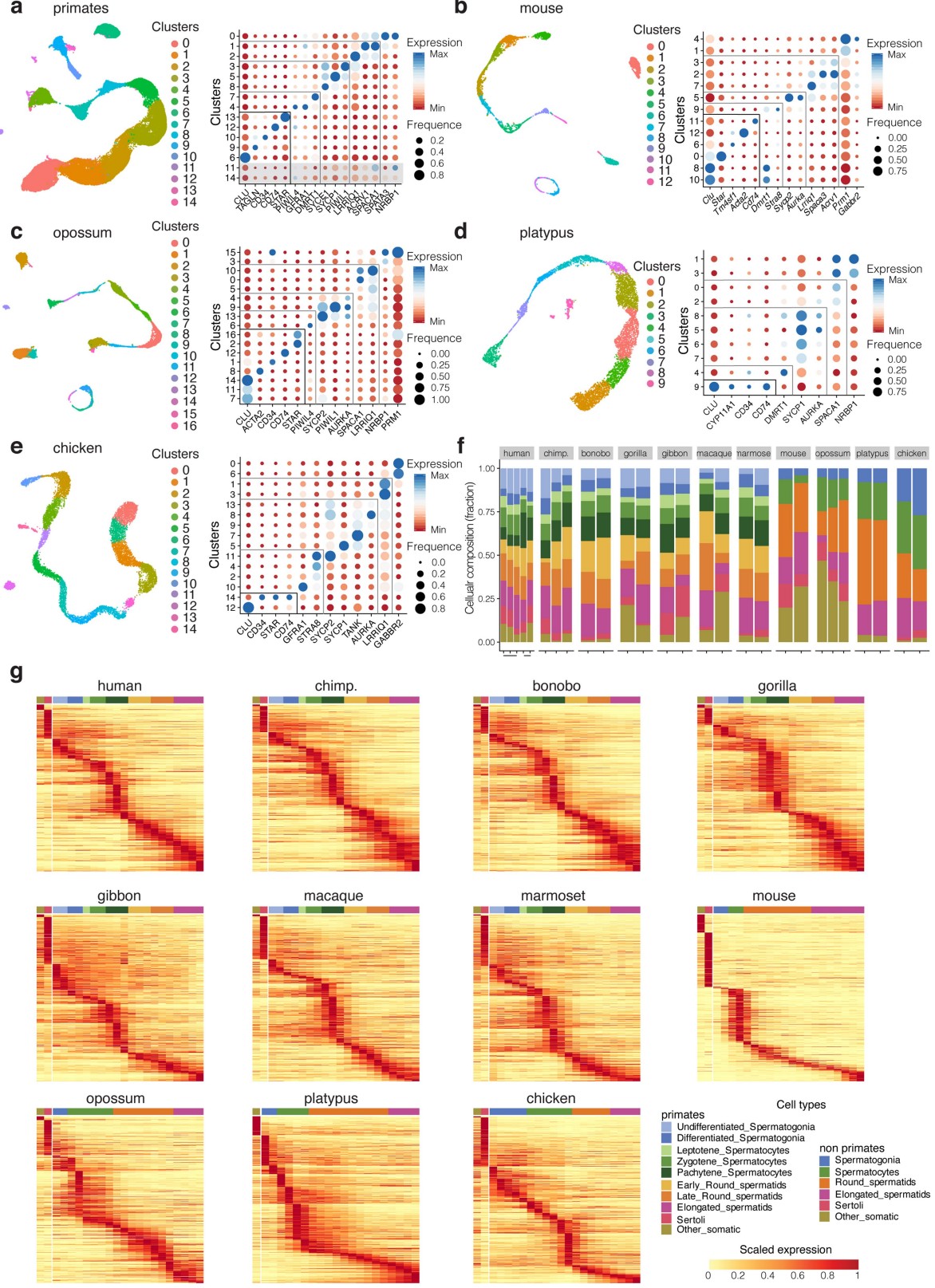

**Extended Data Fig. 1 | Cellular composition and marker gene assessment.**
**a**–**e**, Clustering and UMAP representations (left), marker gene-based cell type assignment (right), for **a**, primates, **b**, mouse, **c**, opossum, **d**, platypus and **e**, chicken. *CLU* marks Sertoli cells; *TAGLN* and *ACTA2* peritubular and smooth muscle cells; *CD34* and *TM4SF1* endothelial cells; *APOE* and *CD74* macrophages; *STAR* and *CYP11A1* Leydig cells; *GFRA1, PIWIL4* (undifferentiated), *DMRT1*

(differentiated), and *STRA8* spermatogonia; *SYCE1* (leptotene), *SYCP1* (zygotene), *PIWIL1* (pachytene), *SYCP2, TANK* and *AURKA* spermatocytes; *LRRIQ1* (early), *ACRV1* (late) and *SPACA1* (late) round spermatids; *SPATA3, NRBP1, PRM1* and *GABBR2* elongated spermatids. **f**, Cell-type composition. **g**, Scaled expression of specific genes (y-axis) for all species and cell types (x-axis).

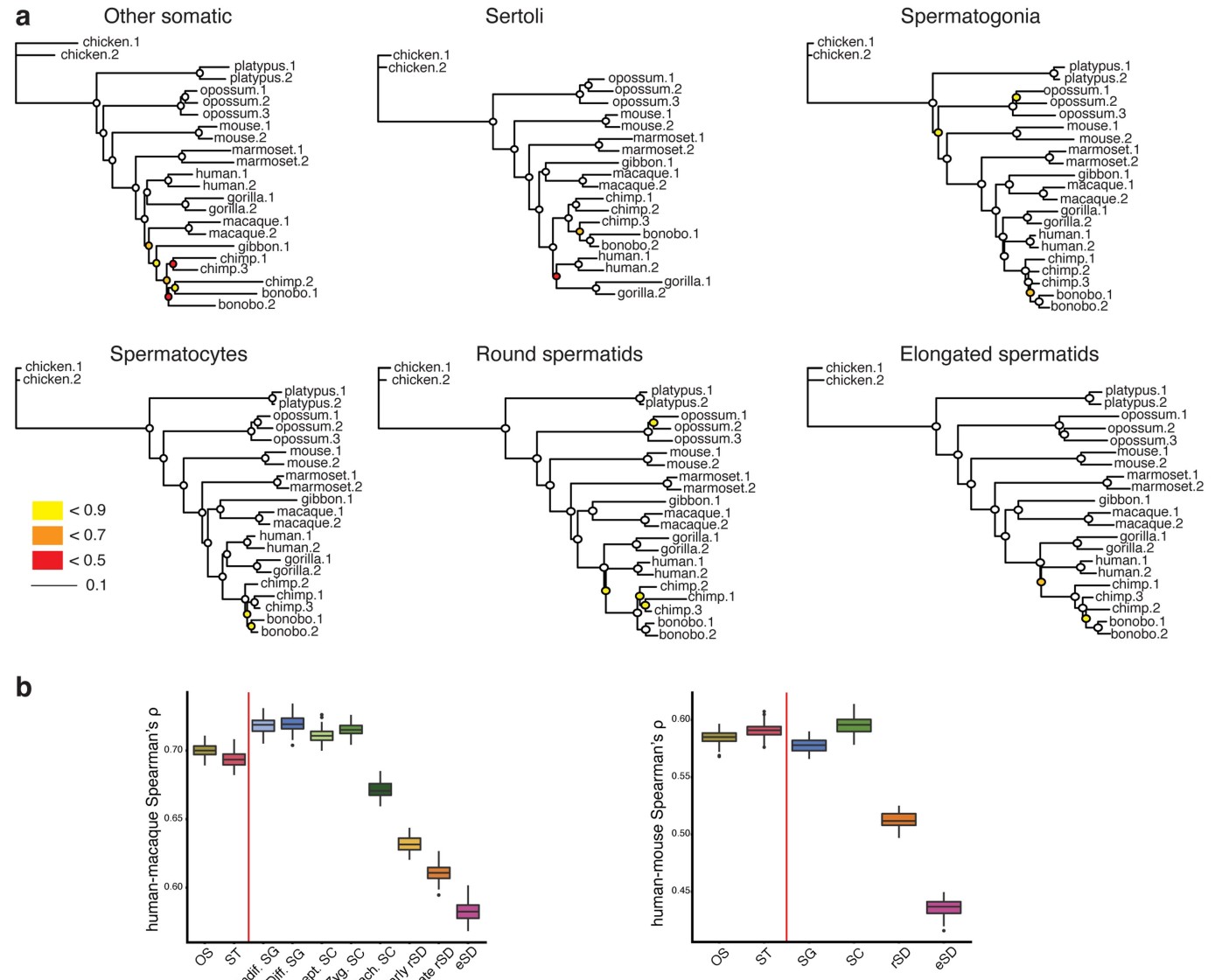

**Extended Data Fig. 2 | Mammalian testicular gene expression divergence.**
**a**, Gene expression phylogenies. Neighbor-joining trees based on pairwise expression level distances (1–rho, Spearman's correlation coefficient) for the main testicular cell types, respectively ("Other somatic" refers to peritubular, smooth muscle, endothelial, macrophage, and Leydig cells as detailed in the legend of Extended Data Fig. 1 and Methods). Trees are drawn to the same scale (indicated by the scale bar). Bootstrap values (i.e., proportions of replicate trees that have the branching pattern as in the majority-rule consensus tree shown) are indicated by circles at the corresponding nodes: ≥ 0.9 (white fill). **b**, Human-macaque (left, n = 304) and human-mouse (right, n = 257) pairwise Spearman correlations after downsampling to the same number of nuclei and UMI across cell types for each species. Boxplots show the median (central value); upper and lower quartile (box limits) and 95% confidence intervals (whiskers) for 100 bootstrap replicates.

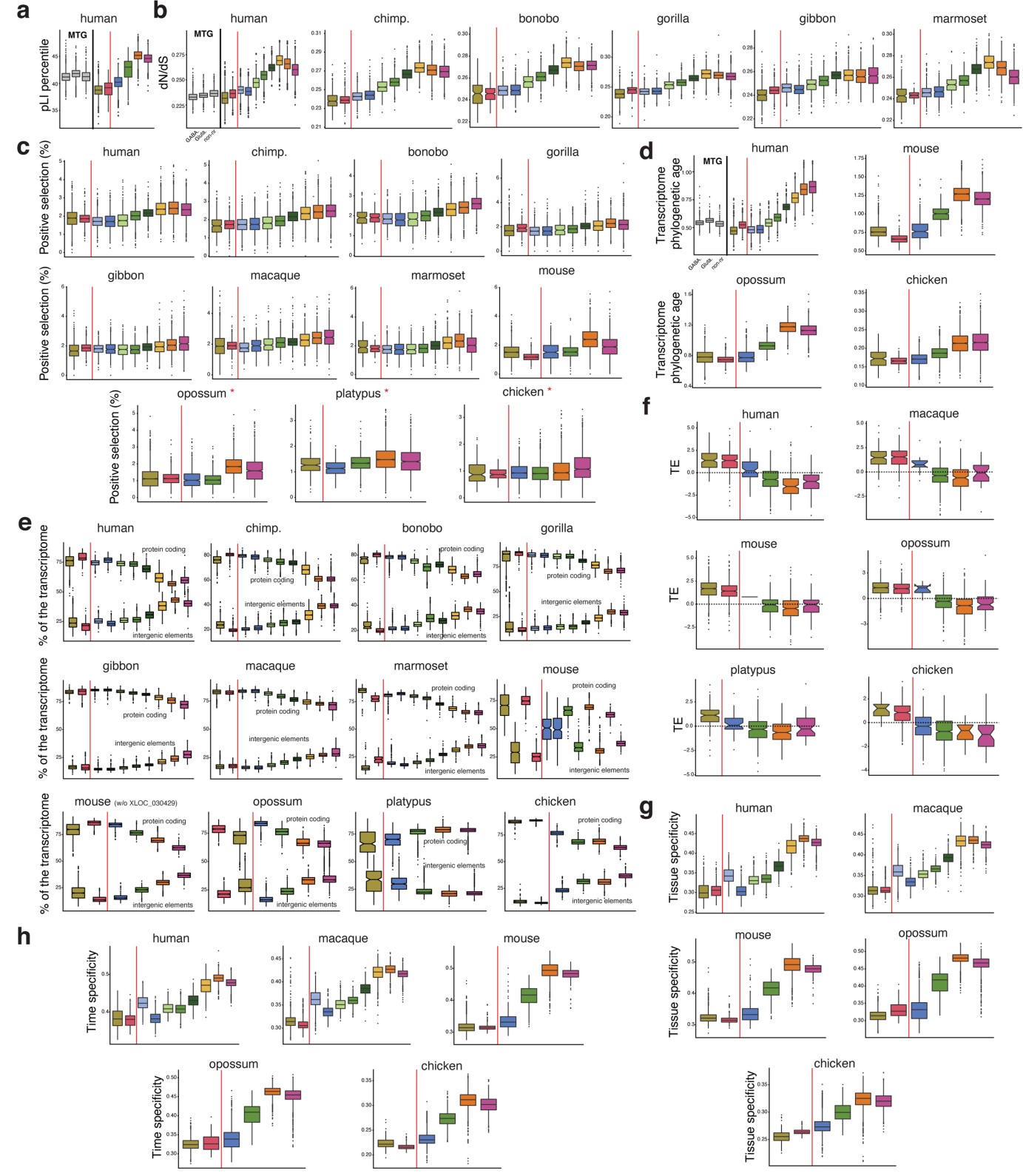

**Extended Data Fig. 3 | Evolutionary forces. a**, Tolerance to functional variants. **b**, Average normalized ratio of nonsynonymous over synonymous nucleotide substitutions (dN/dS) of expressed genes across primates. **c**, Percentage of positively selected genes along spermatogenesis. Opossum, platypus, and chicken were not used in the original studies[72,73] and so the 1:1 orthologues in these species were used. **d**, Phylogenetic age of cellular transcriptomes. Higher values reflect increased lineage-specific gene contributions (that is, younger transcriptomes). **e**, Percentages of UMIs that map to protein coding genes (top) or to intergenic elements (bottom). XLOC_030429 drives a different mouse pattern. **f**, Translational efficiency (TE; normalized log$_2$-transformed values) of expressed genes. **g**, Tissue specificity. **h**, Time specificity (development). In **a**, **b**, and **d**, human middle temporal gyrus (MTG) snRNA-seq data[84] was used for non-gonadal (brain) comparisons. See legend of main Fig. 2 for further details regarding the plotted measures. Box plots depict the median (center value); upper and lower quartile (box limits) with whiskers at 1.5 times the interquartile range.

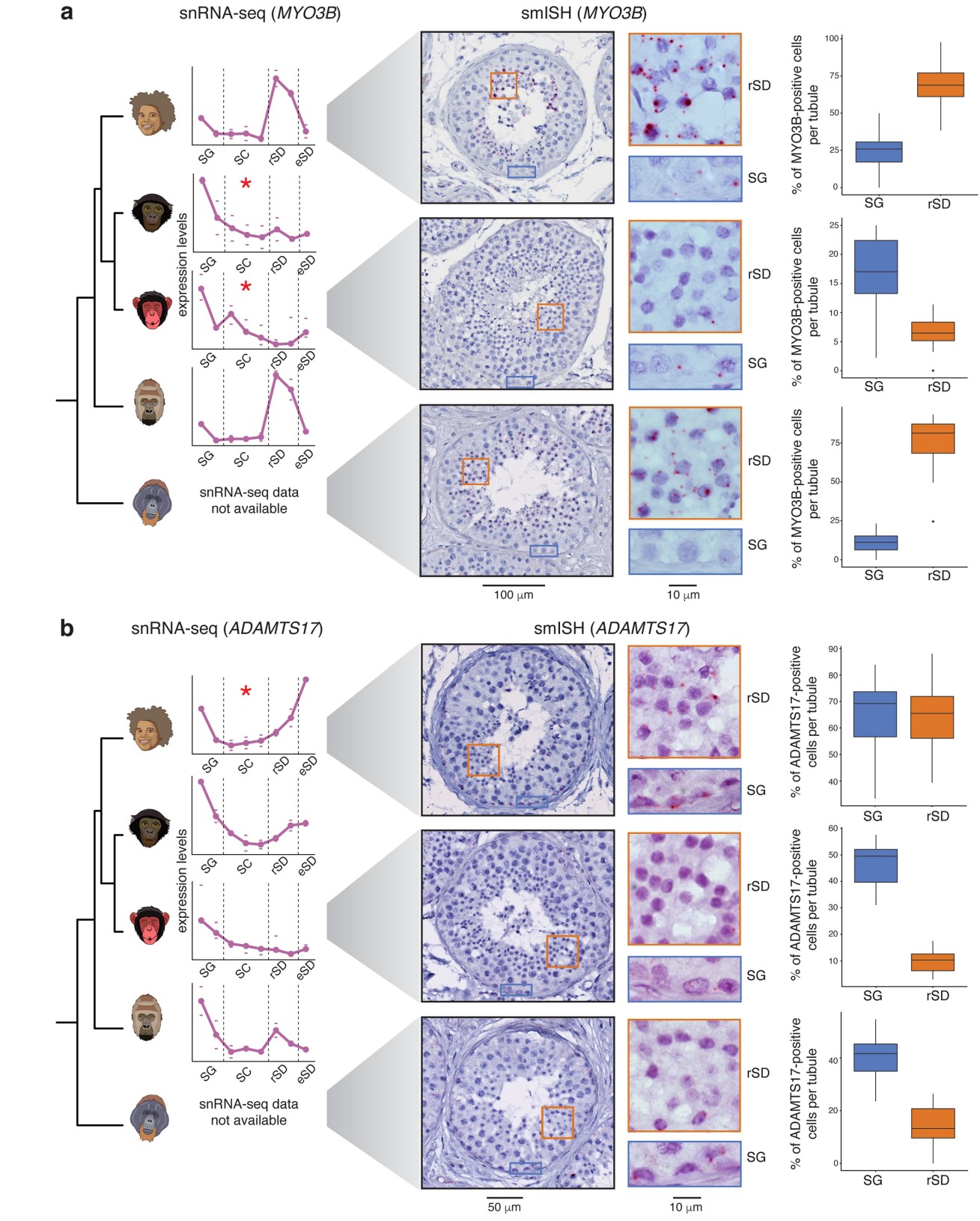

**Extended Data Fig. 4** | See next page for caption.

**Extended Data Fig. 4 | Examples of gene expression trajectory changes along primate spermatogenesis validated by smISH.** Left: Median gene expression trajectory from snRNA-seq along spermatogenesis for human, bonobo, chimp., and gorilla (from top to bottom). For each replicate, mean expression levels across cells of a given cell type were calculated. Marks indicate values for the replicates with the highest and lowest mean expression levels, dots indicate the median of mean expression values of the replicates. There are two biological replicates for human, bonobo, and gorilla and three biological replicates for chimpanzee. Center: images of seminiferous tubule cross sections from human, chimp., and orangutan (from top to bottom) stained with smISH for *MYO3B* (**a**) and *ADAMTS17* (**b**) using RNAScope. Red asterisks indicate on which branch a trajectory change was called. Examples of spermatogonia (SG) and round spermatids (rSD) closeups are provided for visualization purposes. Right: percentage of positive cells in SG and rSD from 10 tubules per section (n = 3 for human, n = 1 for chimp. and orangutan). Box plots represent the mean (central value) distribution of staining; upper and lower quartile (box limits) with whiskers at 1.5 times the interquartile range.

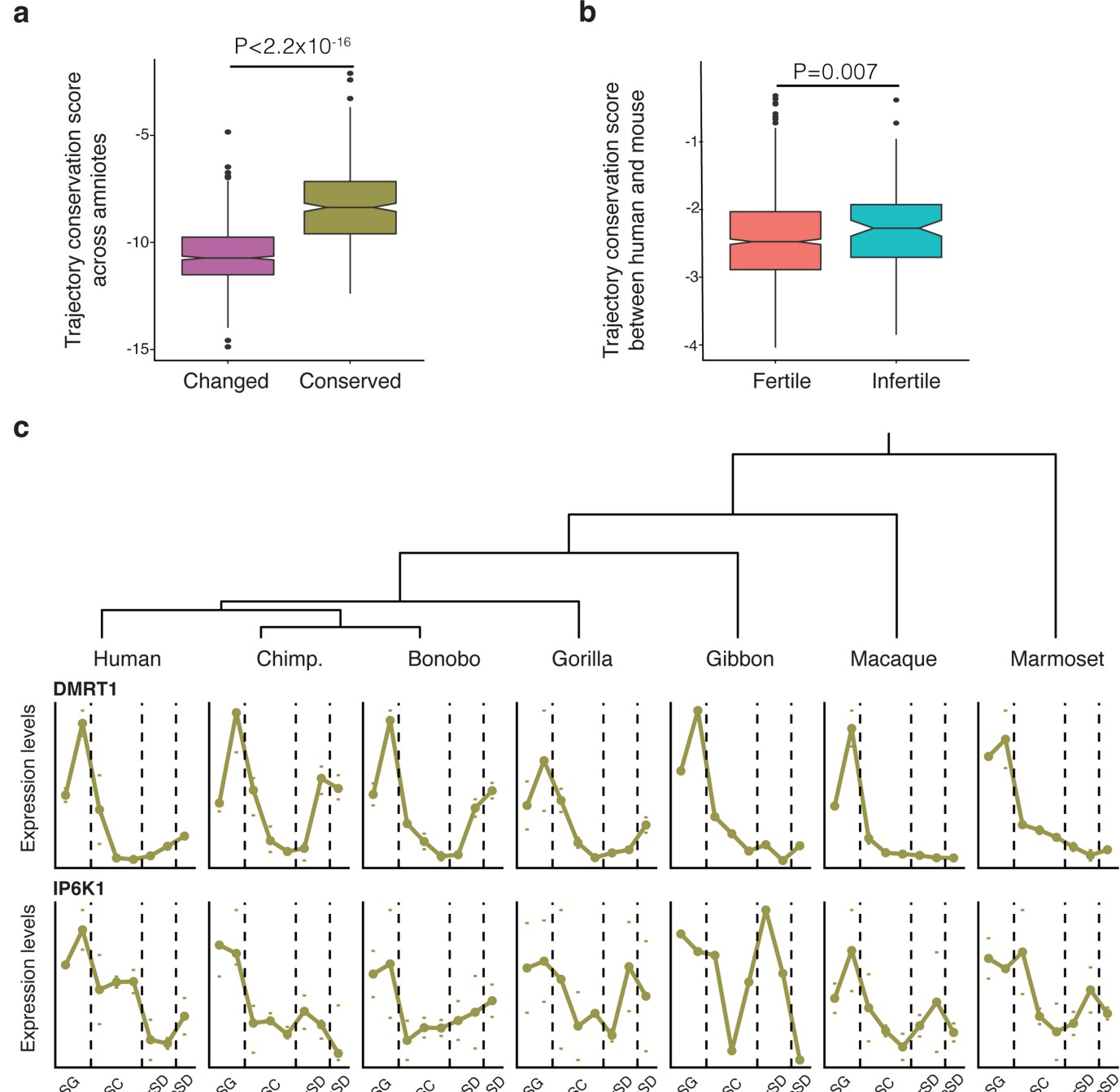

**Extended Data Fig. 5 | Conservation of gene expression trajectories along spermatogenesis. a**, Global trajectory conservation score across amniotes for changed (n = 237) and conserved (n = 389) trajectories (*P* < 2.2x10⁻¹⁶, two-sided Wilcoxon rank-sum test). **b**, Trajectory conservation score between human and mouse of genes that lead to either a fertile (n = 1077) or an infertile (n = 108) phenotype when knocked out in mouse (*P* = 0.007, two-sided Wilcoxon rank-sum test). **a,b**, Box plots depict the median (center value); upper and lower quartile (box limits) with whiskers at 1.5 times the interquartile range. **c**, Gene expression trajectories (*DMRT1*, top; *IP6K1*, bottom) along primate spermatogenesis (spermatogenesis, from left to right). For each replicate, mean expression levels across cells of a given cell type were calculated. Marks indicate values for the replicates with the highest and lowest mean expression levels, dots indicate the median of mean expression values of the replicates. There are three replicates for chimpanzee; two replicates for human, bonobo, gorilla, macaque, and marmoset; one replicate for gibbon. The dashed vertical lines separate spermatogonia (SG), spermatocytes (SC), round (rSD) and elongated spermatids (eSD).

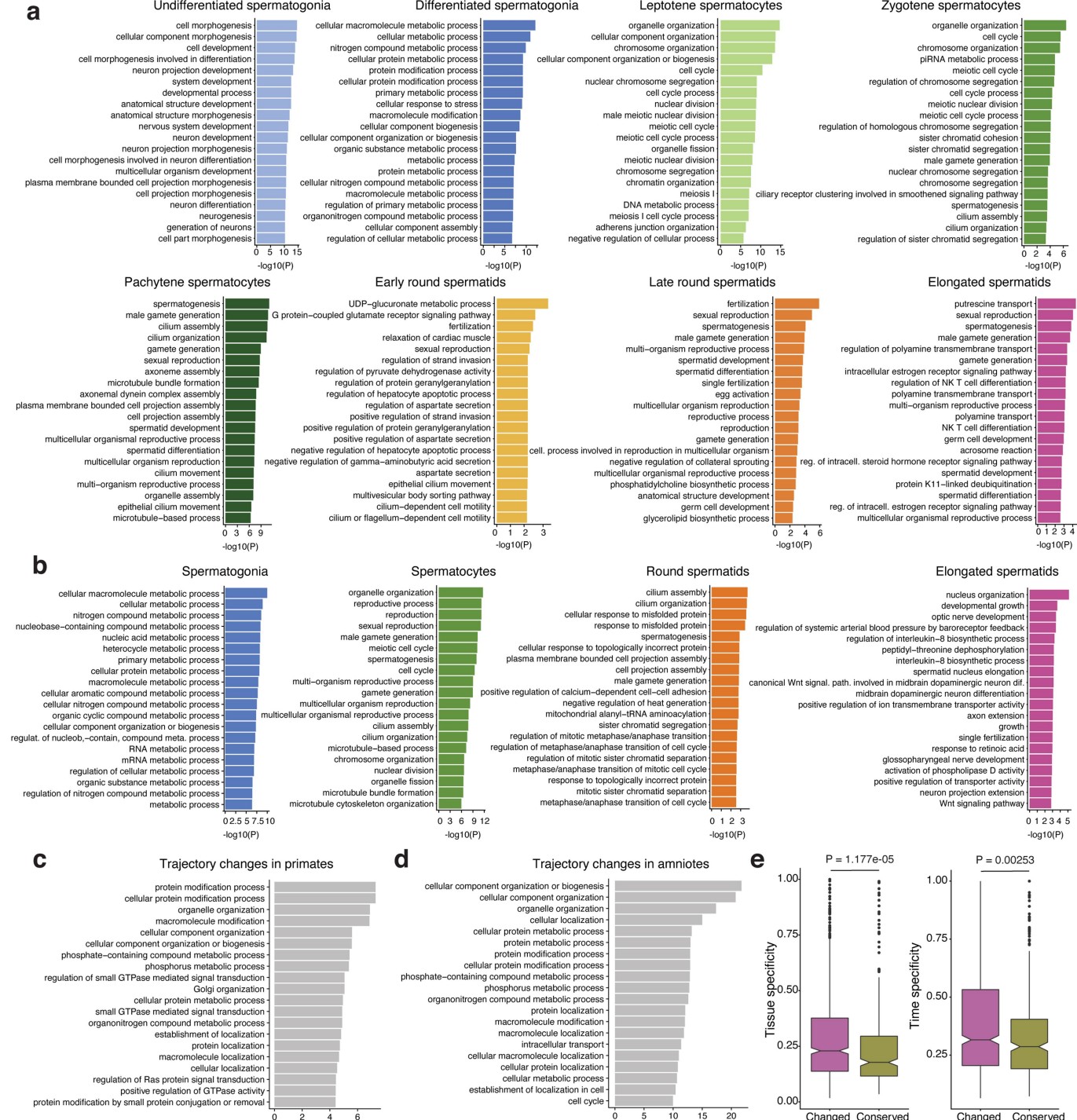

**Extended Data Fig. 6 | Functions of conserved and changed trajectories.**
**a**, Top 20 enriched biological process GO terms of genes showing conserved expression trajectories across primates and peak expression in the different cell types, respectively. **b**, Top 20 enriched biological process GO terms of genes showing conserved trajectories between human and mouse and peak expression in the different cell types, respectively. **c**, The top 20 enriched biological process GO terms of genes with trajectory changes in primates.

**d**, The top 20 enriched biological process GO terms in genes showing trajectory changes in amniotes. **e**, Human tissue (left) and time (right) specificity for changed (n = 237) and conserved (n = 389) trajectories across amniotes, respectively (two-sided Wilcoxon rank-sum tests were performed for statistical comparisons). Box plots depict the median (center value); upper and lower quartile (box limits) with whiskers at 1.5 times the interquartile range.

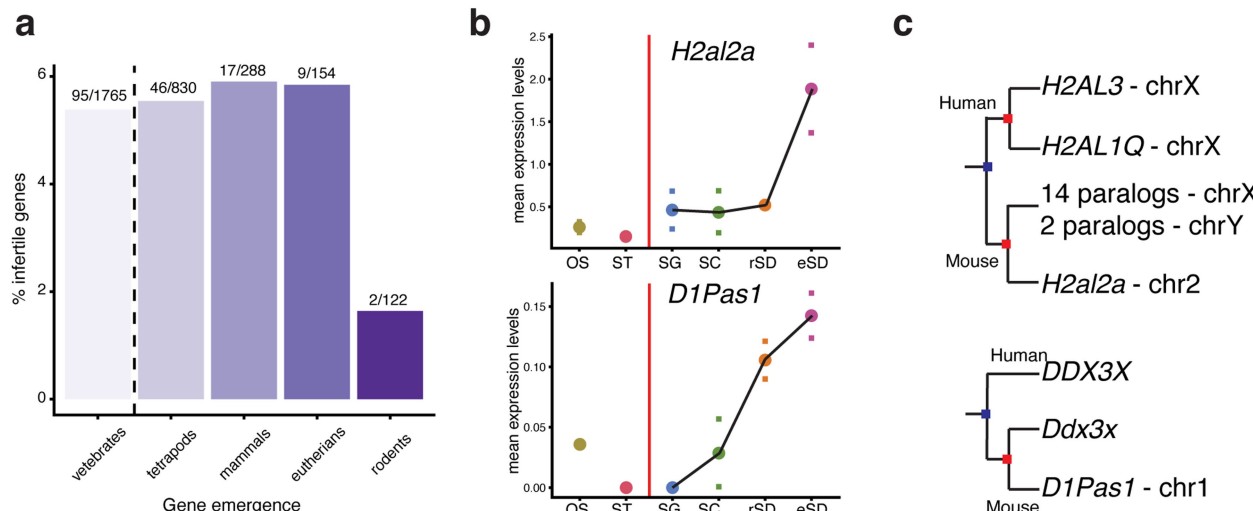

**Extended Data Fig. 7 | New genes with crucial spermatogenic functions.** **a**, percentages of genes among new genes of different ages (i.e., emergence in last common vertebrate, tetrapod, mammalian, eutherian, or rodent ancestors) that show infertility phenotypes when knocked out in the mouse (numbers of genes with infertility phenotypes and total number of genes considered for an age category are provided above each bar). **b**, Expression level trajectories of the retrogenes *H2al2a* and *D1Pas1* in spermatogenic and somatic cell types. For each of the two replicates, mean expression levels across cells of a given cell type were calculated. Marks indicate values for the replicates and dots indicate the median of the values of the replicates. **c**, Trees of *H2al2a* and *D1Pas1* and paralogs from which they originated through the mechanism of RNA-based gene duplication (i.e., *D1Pas1* stems from *Ddx3y*, whereas the precise ancestral parental gene of *H2al2a* is not easily discernable due to a complex history of a number of RNA- and DNA-based duplication events).

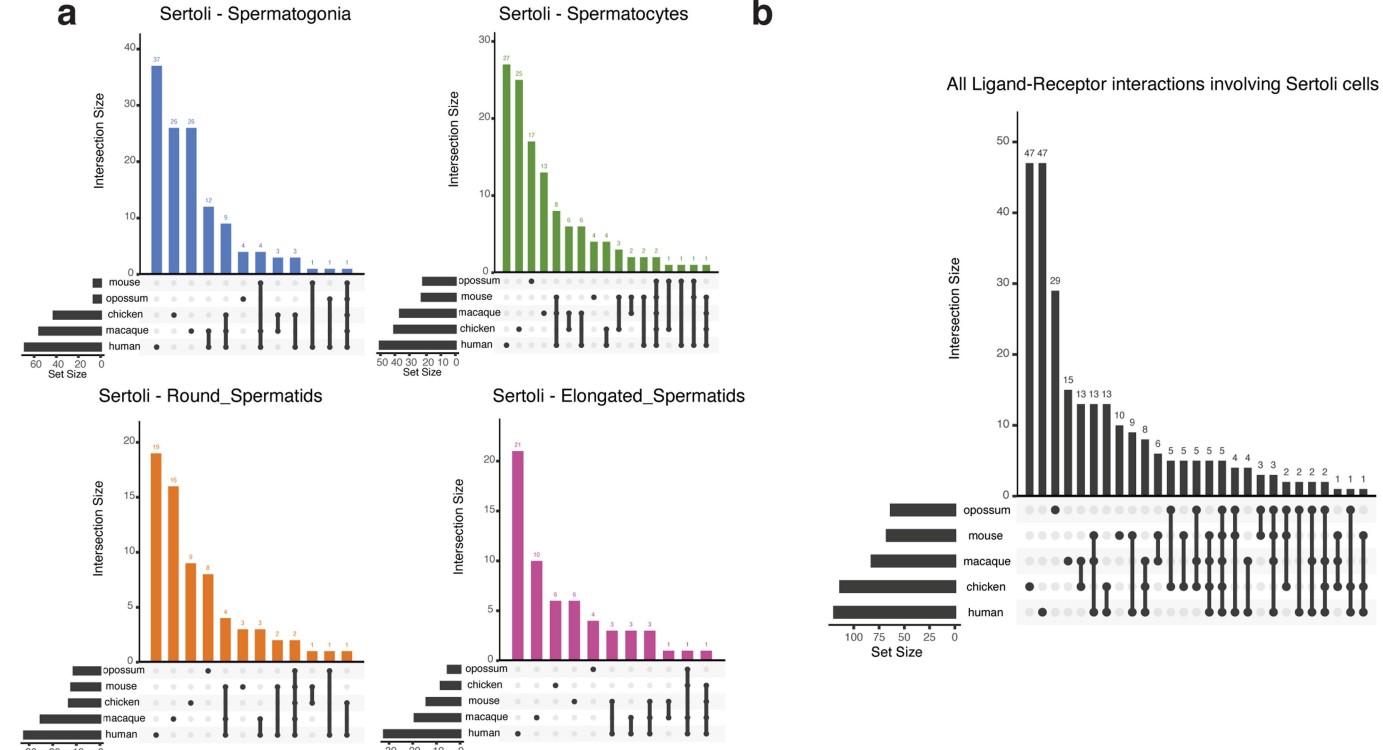

**Extended Data Fig. 8 | Sertoli-germ cell communications mediated by ligand-receptor interactions in testis across mammals. a**, Significant ligand-receptor interactions (as assessed by CellPhoneDB[78]; see Methods) across species for Sertoli-spermatogonia, Sertoli-spermatocytes, Sertoli-round spermatids and Sertoli-elongated Spermatids communications (details in Supplementary Table 9). **b**, Overview of all distinct significant ligand-receptor interactions between Sertoli and the four principal germ cell types across species. The number of detected significant interactions for each species in both panels is indicated to the bottom left of each plot (Set size).

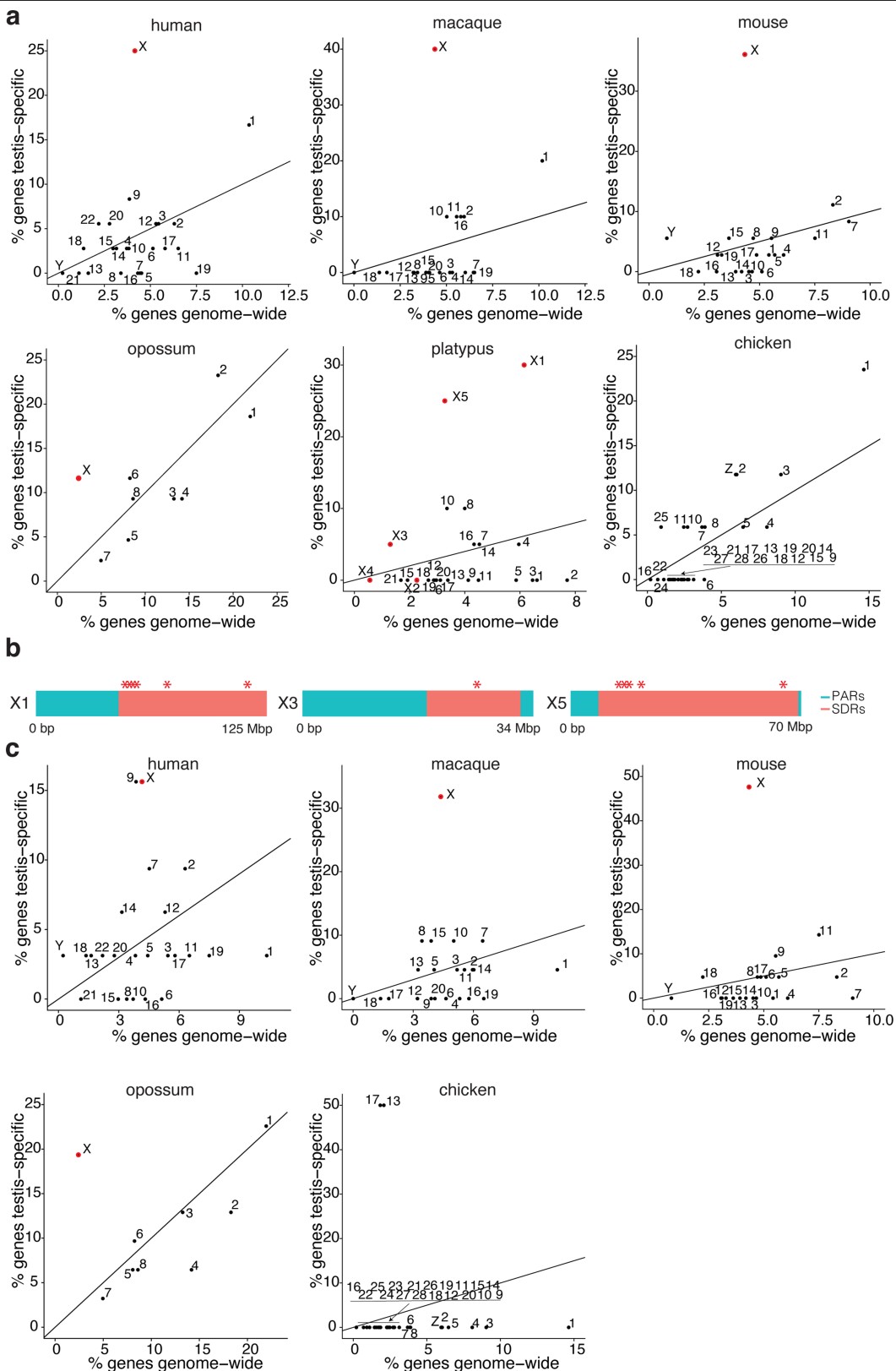

**Extended Data Fig. 9 | Per chromosome testis-specific genes enriched in spermatogonia and Sertoli cells. a,c,** Per chromosome percentage of expressed testis-specific genes (y-axis) with predominant expression in differentiated spermatogonia (human and macaque) or spermatogonia (mouse, opossum and platypus) (**a**) and Sertoli cells (**c**) versus the percentage of all genes in the genome located on a given chromosome (x-axis: "% genes genome-wide"; i.e., the number of genes on a given chromosome divided by the total number of genes in the genome). X chromosomes are colored in red. The diagonal shows the numbers expected if testis-specific genes were randomly distributed across the genome. **b,** Location of platypus testis-specific genes enriched in spermatogonia on chromosomes X1, X3 and X5.

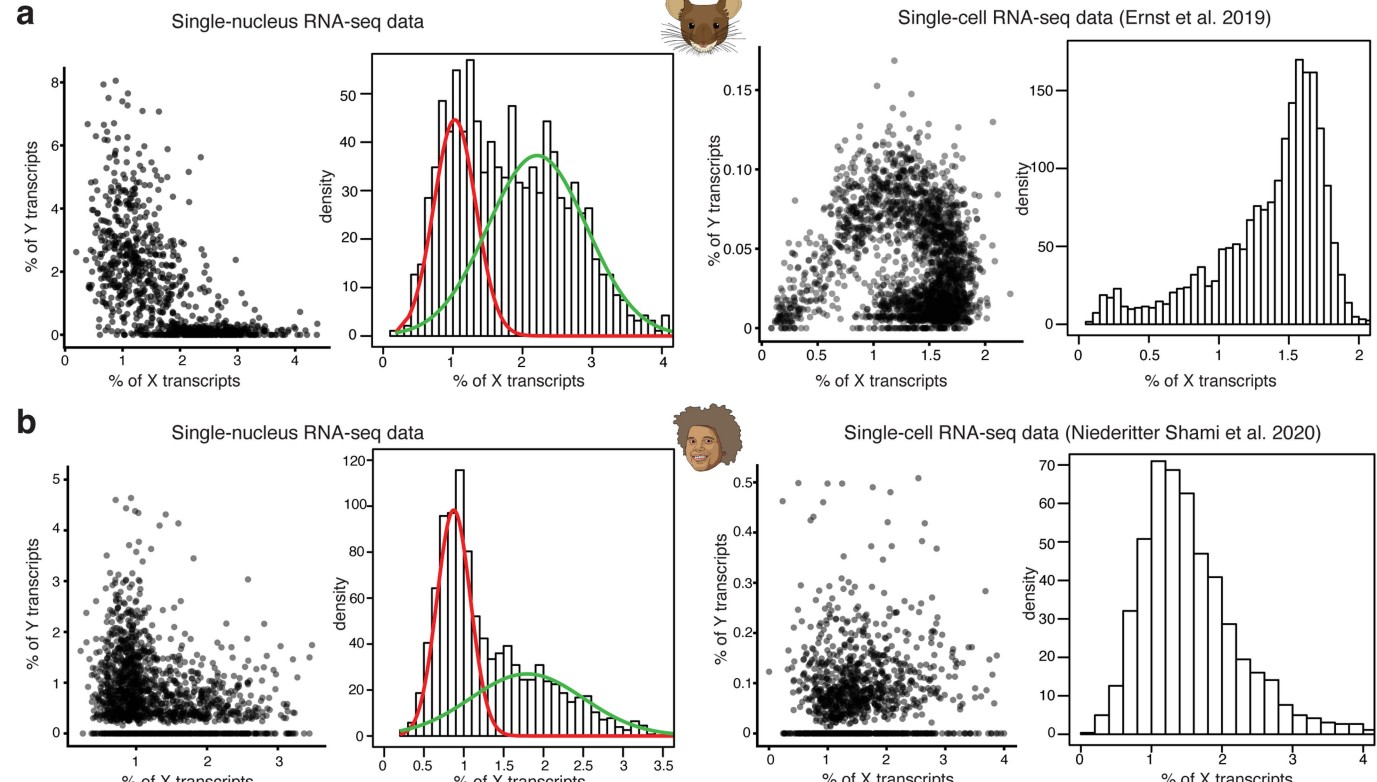

**Extended Data Fig. 10 | Classification of X and Y bearing spermatids.** The scatter and bar plots show the percentages of X and Y transcripts, and the distribution of X transcripts (%), respectively, across nuclei or cells in our single-nucleus (left) and publicly available single-cell data in mouse[43] (**a**) and human[23] (**b**). The red and green lines depict the fitted curves for the bimodal distributions.

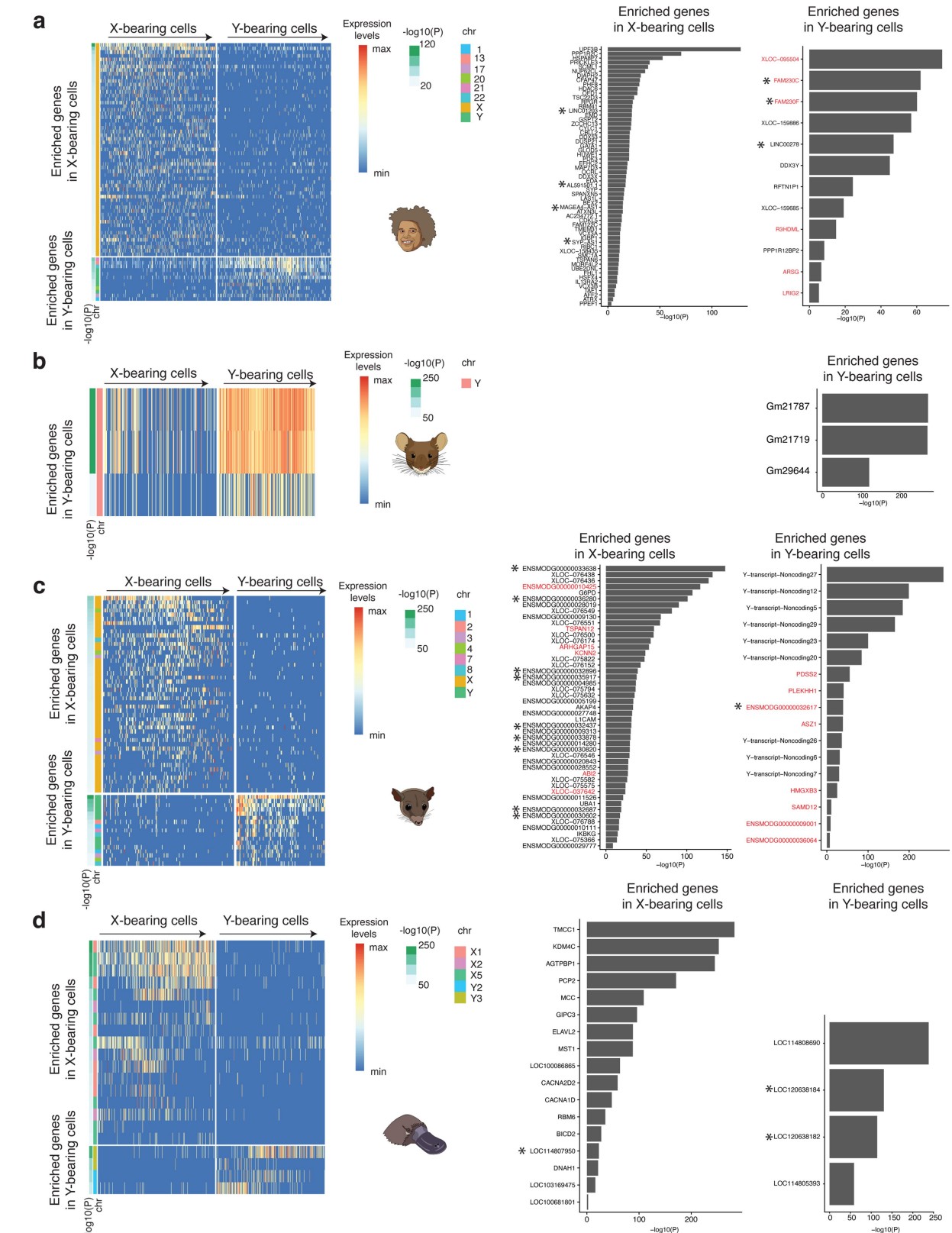

**Extended Data Fig. 11 | Transcriptome differences between X and Y bearing spermatids.** The heatmaps indicate transcript abundances of significantly enriched genes in X- and Y-bearing spermatids in the different species (a: human, b: mouse, c: opossum, d: platypus). The corresponding bar plots (genes in the same order) show the genes with significantly enriched expression in X- and Y-bearing cells, respectively using two-sided Wilcoxon rank-sum tests with Bonferroni correction (ordered according to increasing P-values) (see also lists in Supplementary Table 11). Genes marked with an asterisk are long non-coding RNAs. Genes in red are located on autosomes, those in black on sex chromosomes. "XLOC" denotes loci annotated in our previous work[6]; notably, only few XLOC loci (opossum: XLOC-076551, XLOC-076500, XLOC-076152, XLOC-076546) were found to have coding potential (as assessed by analyses based on ribosome profiling data generated in our previous work[6]), whereas all others likely represent lncRNAs.

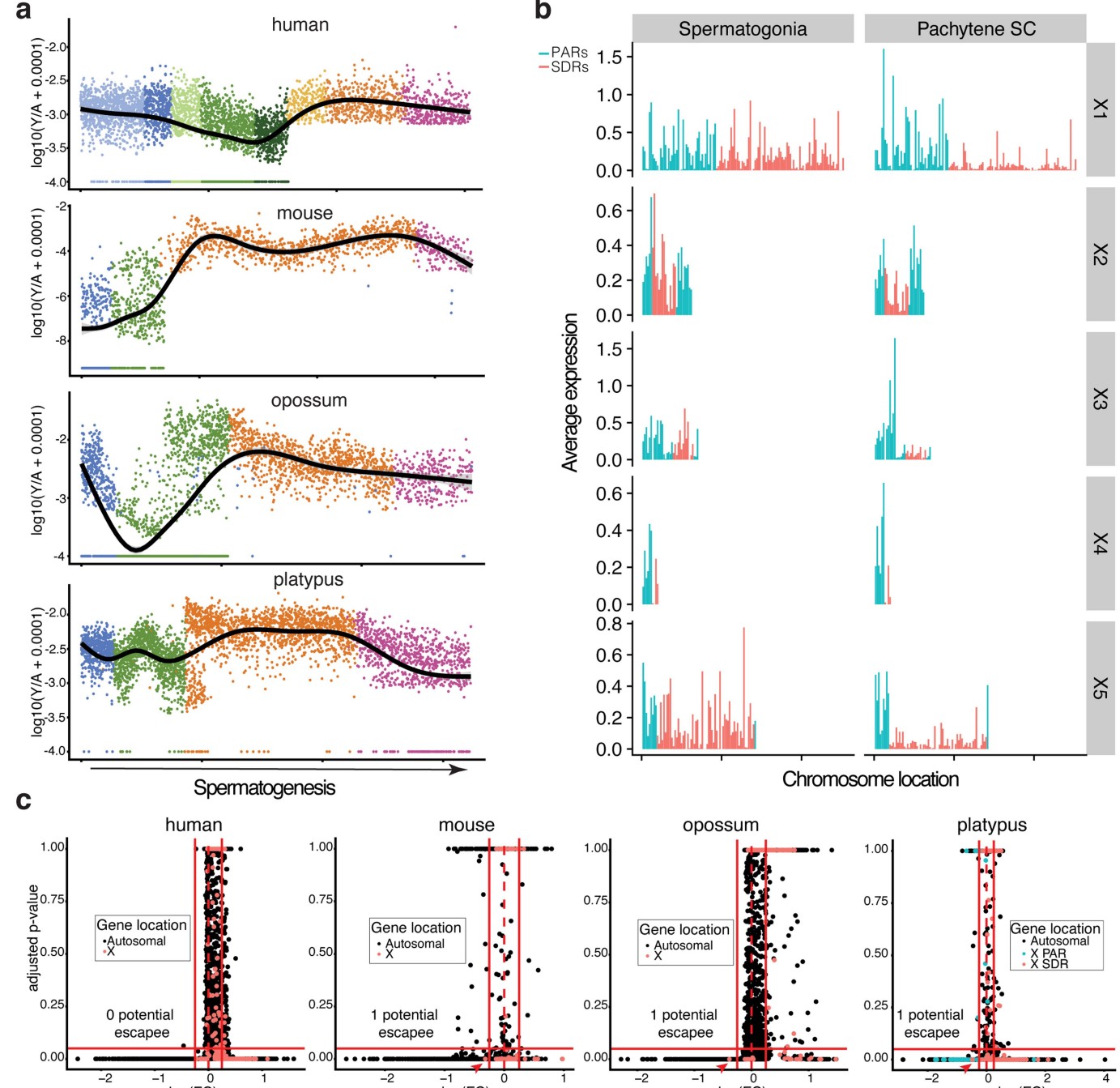

**Extended Data Fig. 12 | Mammalian sex chromosome evolution.**
**a**, Y-to-autosome transcript ratios in individual germ cells along spermatogenesis. For spermatids, only Y-bearing cells are considered. Whiskers depict the median ± 25th and 75th percentiles. **b**, Average expression along platypus X chromosomes in spermatogonia and pachytene spermatocytes. Genes are grouped into 1,000,000 bp bins along the chromosomes. **c**, To identify potential MSCI escapees, we screened for X-linked genes in SDRs with a significant increase in transcript abundance from SG to SC stages subject to MSCI, which ensures that potential escape genes are indeed actively transcribed in SC (i.e., they do not merely represent genes expressed in SG with stable transcripts still detectable in SC) (Methods). Expression differences (log$_e$-fold change, x-axis) of X and autosomal genes between SC and SG were assessed using differential expression analysis (two-sided Wilcoxon rank-sum tests; $p$-values on the Y-axis are Bonferroni-corrected). Vertical red lines indicate a 0.25-fold expression change (log$_e$-scale); horizontal red lines indicate corrected $P < 0.05$. Potential escapees (i.e., X-linked genes in SDRs with significantly higher expression in SC than SG) are indicated by red arrows.

# Reporting Summary

## Statistics

For all statistical analyses, confirm that the following items are present in the figure legend, table legend, main text, or Methods section.

| n/a | Confirmed | |
|-----|-----------|---|
| ☐ | ☒ | The exact sample size (*n*) for each experimental group/condition, given as a discrete number and unit of measurement |
| ☐ | ☒ | A statement on whether measurements were taken from distinct samples or whether the same sample was measured repeatedly |
| ☐ | ☒ | The statistical test(s) used AND whether they are one- or two-sided *Only common tests should be described solely by name; describe more complex techniques in the Methods section.* |
| ☒ | ☐ | A description of all covariates tested |
| ☐ | ☒ | A description of any assumptions or corrections, such as tests of normality and adjustment for multiple comparisons |
| ☐ | ☒ | A full description of the statistical parameters including central tendency (e.g. means) or other basic estimates (e.g. regression coefficient) AND variation (e.g. standard deviation) or associated estimates of uncertainty (e.g. confidence intervals) |
| ☐ | ☒ | For null hypothesis testing, the test statistic (e.g. *F*, *t*, *r*) with confidence intervals, effect sizes, degrees of freedom and *P* value noted *Give P values as exact values whenever suitable.* |
| ☒ | ☐ | For Bayesian analysis, information on the choice of priors and Markov chain Monte Carlo settings |
| ☒ | ☐ | For hierarchical and complex designs, identification of the appropriate level for tests and full reporting of outcomes |
| ☐ | ☒ | Estimates of effect sizes (e.g. Cohen's *d*, Pearson's *r*), indicating how they were calculated |

*Our web collection on statistics for biologists contains articles on many of the points above.*

## Software and code

Policy information about availability of computer code

| Data collection | No software was used. |
|-----------------|------------------------|
| Data analysis | Open source software including CellRanger (2.1.1) and (3.0.2), CellPhoneDB v2, cutadapt (1.8.3), Tophat2 (2.1.1), StringTie (1.3.3), cuffcompare (v2.2.1), PAML (4.9), NDP.view2Plus (2.8.24), the R (3.6.2) packages Seurat (3.1.4), biomaRt (2.40.5), LIGER (0.5.0), slingshot (1.2.0), mfuzz (2.44.0), ape (5.3), dplyr (0.8.5), countcolors (0.9.1), colordistance (1.1.1), mixtools (1.2.0), ggplot2 (3.2.1), tidyverse (1.3.0), cowplot (1.0.0), limma (3.40.6), UpSetR (v1.4.0), and pheatmap (1.0.12) were used in this study for data analyses. |

For manuscripts utilizing custom algorithms or software that are central to the research but not yet described in published literature, software must be made available to editors and reviewers. We strongly encourage code deposition in a community repository (e.g. GitHub). See the Nature Portfolio guidelines for submitting code & software for further information.

## Data

Policy information about availability of data

All manuscripts must include a data availability statement. This statement should provide the following information, where applicable:

- Accession codes, unique identifiers, or web links for publicly available datasets
- A description of any restrictions on data availability
- For clinical datasets or third party data, please ensure that the statement adheres to our policy

Raw and processed bulk and single-nucleus RNA-seq data have been deposited in ArrayExpress with the accession codes E-MTAB-11063 (human snRNA-seq), E-MTAB-11064 (chimpanzee snRNA-seq), E-MTAB-11067 (bonobo snRNA-seq), E-MTAB-11065 (gorilla snRNA-seq), E-MTAB-11066 (gibbon snRNA-seq), E-MTAB-11068 (macaque snRNA-seq), E-MTAB-11069 (marmoset snRNA-seq), E-MTAB-11071 (mouse snRNA-seq), E-MTAB-11072 (opossum snRNA-seq), E-MTAB-11070 (platypus snRNA-seq), E-MTAB-11073 (chicken snRNA-seq) and E-MTAB-11074 (chimpanzee, gorilla, gibbon and marmoset bulk RNA-seq) (https://

www.ebi.ac.uk/arrayexpress/).
IMPC database is available at: ftp://ftp.ebi.ac.uk/pub/databases/impc/
Receptor-ligand interactions: https://www.cellphonedb.org/downloads
Gene phylogenetic ages: http://gentree.ioz.ac.cn/
Translational efficiency data: E-MTAB-7247
Time and tissue specificity data: https://apps.kaessmannlab.org/evodevoapp/
pLI scores: http://genetics.bwh.harvard.edu/genescores/selection.html

# Field-specific reporting

Please select the one below that is the best fit for your research. If you are not sure, read the appropriate sections before making your selection.

☒ Life sciences  ☐ Behavioural & social sciences  ☐ Ecological, evolutionary & environmental sciences

For a reference copy of the document with all sections, see nature.com/documents/nr-reporting-summary-flat.pdf

# Life sciences study design

All studies must disclose on these points even when the disclosure is negative.

| | |
|---|---|
| Sample size | No statistical methods were used to determine sample size. Sample size was based on the number of individuals available (see Supplementary Table 1). |
| Data exclusions | Low quality nuclei were excluded as described in Methods. |
| Replication | We generated 2 biological replicates for human (5 technical replicates), bonobo, gorilla, macaque, marmoset, mouse, platypus, and chicken; 3 biological replicates for chimpanzee and opossum; and 2 technical replicates for gibbon. |
| Randomization | Not relevant, because no treatment groups. |
| Blinding | Blinding was not relevant to our study. Both data collection and analyses required an understanding of the nature of the sample being collected/analyzed. |

# Reporting for specific materials, systems and methods

We require information from authors about some types of materials, experimental systems and methods used in many studies. Here, indicate whether each material, system or method listed is relevant to your study. If you are not sure if a list item applies to your research, read the appropriate section before selecting a response.

### Materials & experimental systems

| n/a | Involved in the study |
|---|---|
| ☒ | Antibodies |
| ☒ | Eukaryotic cell lines |
| ☒ | Palaeontology and archaeology |
| ☐ | ☒ Animals and other organisms |
| ☐ | ☒ Human research participants |
| ☒ | Clinical data |
| ☒ | Dual use research of concern |

### Methods

| n/a | Involved in the study |
|---|---|
| ☒ | ChIP-seq |
| ☒ | Flow cytometry |
| ☒ | MRI-based neuroimaging |

# Animals and other organisms

Policy information about studies involving animals; ARRIVE guidelines recommended for reporting animal research

| | |
|---|---|
| Laboratory animals | All samples used are from males. The species used in this study were chimpanzee (Pan troglodytes, 14 yo, 21 yo, 45 yo), bonobo (Pan paniscus, 36 yo, 15 yo), gorilla (Gorilla gorilla 43 yo, 51 yo), gibbon (Hylobates lar, 5 yo), macaque (Macaca mulatta, 7 yo, 9 yo), marmoset (Callithrix jacchus, 10 yo), mouse (Mus musculus, CD-1, adult; 12 h day night cycle, temperature 20-24 °C, 45-65 % humidity), opossum (Monodelphis domestica, adult), platypus (Ornithorhynchus anatinus, adult) and chicken (Gallus gallus, red junglefowl, adult) (see Supplementary Table 1). |
| Wild animals | Adult male platypus (Ornithorhynchus anatinus), Animals were euthanized with an intraperitoneal injection of 0.1 mg/g pentobarbital, for tissue collection. Note that the samples were collected as part of a previous study; i.e., animals were not sacrificed for the purpose of this study. |
| Field-collected samples | Adult male platypus (Ornithorhynchus anatinus), Animals were euthanized with an intraperitoneal injection of 0.1 mg/g pentobarbital, for tissue collection. Note that the samples were collected as part of a previous study; i.e., animals were not sacrificed for the purpose of this study. |

| Ethics oversight | The use of mammalian animal samples for the type of work in this study was approved by ERC Ethics Screening panels (ERC Starting Grant 242597, SexGenTransEvolution, and ERC Consolidator Grant 615253, OntoTransEvol). |

Note that full information on the approval of the study protocol must also be provided in the manuscript.

# Human research participants

Policy information about studies involving human research participants

| Population characteristics | Human samples were obtained from official scientific tissue banks or dedicated companies; informed consent was obtained by these sources from donors prior to death or from next-of-kin. All samples are from adult caucasian males |

| Recruitment | Human samples were obtained from official scientific tissue banks or dedicated companies; informed consent was obtained by these sources from donors prior to death or from next-of-kin. |

| Ethics oversight | The use of all human samples for the type of work described in this study was approved by an Ethics Screening panel from the European Research Council (ERC) (associated with H.K.'s ERC Consolidator Grant 615253, OntoTransEvol) and local ethics committees; that is, from the Cantonal Ethics Commission Lausanne (authorization 504/12), Ethics Commission from the Medical Faculty of Heidelberg University (authorization S-220/2017), and the regional medical research Ethics committee of the capital region of Copenhagen (H-16019637) |

Note that full information on the approval of the study protocol must also be provided in the manuscript.

