## [Peer Review File. · Nature]

Manuscript Title: The molecular evolution of spermatogenesis across mammals

Reviewer Comments & Author Rebuttals

Reviewer Reports on the Initial Version:

Referee expertise:

Referee #1: comparative genomics

Referee #2: germ cells, single-cell genomics

Referee #3: germ cell transcriptomics

Referees' comments:

Referee #1 (Remarks to the Author):

In this study, Murat, Mbengue and colleagues profile gene expression in single nuclei from testis samples across 11 vertebrate species (10 mammals and one bird outgroup), to characterize the evolution of testis cell types. The authors show that overall, testis cell types and their core gene expression programs are conserved, especially in primates. However, late-spermatogenesis cell types have significantly more diverged gene expression programs than early cell types, and seem to underlie the rapid evolution of the testis transcriptome previously described in the literature. The authors confirm that this evolutionary divergence in late cell types is associated with both relaxed and positive selection on expressed genes, expression of many testis-specific genes – some of which are involved in infertility – and generally pervasive transcription. The authors then detect relatively large numbers of genes whose expression has changed during the evolution of primates, and which may be involved in recent testis evolution, although the functional implications of these genes are not investigated. Finally, the last part of the paper focuses on sex-chromosome-specific gene expression. The authors show that X- and Y-bearing spermatids can be distinguished based on their nuclear transcriptome, and testis-specific genes are enriched on the X chromosome. Additionally, male-specific sex chromosome inactivation was detected in sex-differentiated regions of the gonosomes in platypus, suggesting that the process may be ancestral to vertebrates.

The paper is well written and easy to follow, with clear messages for each of the main figures. Generally the conclusions seem well-supported by the data, although I have some caveats related to a number of methodological points. Especially, it is not clear to me throughout the paper what fraction of the transcriptomes are actually analysed, and whether the conclusions are representative of the full picture (see detailed comments below). In my opinion, the most exciting and novel aspect is the investigation of sex chromosome differences and MSCI at the end of the manuscript, which will be of interest to a wide readership but would benefit from a more focused and deeper analysis. The rest of the manuscript is a technical tour de force, but the results are more incremental and

largely confirm previous results about transcriptome divergence in testis.

Major comments

1. Quality control needs to be discussed in more detail, including in the main manuscript. The total numbers of cells per species described in Table S1 are rather low by current standards, and some samples have very low median numbers of detected genes (under 1000). The authors provide no QC per cell type, so it is unclear whether some cell types are underrepresented in the sampling, or suffer from specific quality issues, which may in turn affect the conclusions when the authors compare evolutionary dynamics between cell types. The authors also mention using Seurat to remove batch effects within species, but it is unclear how strong this batch effect was, and whether this again may affect interpretations. How correlated are technical/biological replicates on average?

2. Due to the nature of this work, the authors focus on 1-to-1 ortholog genes for transcriptomic comparisons across species. For all 11 vertebrate species, this effectively reduces transcriptomes to ~5,000 coding genes, less than 25% of human genes, as described in Fig 1. It was not clear to me whether all analyses in the manuscript use this same set of ~5,000 genes, or whether some parts (comparisons within primates, etc) use more extensive datasets. This is a potential issue because deeply conserved 1-to-1 orthologs are functionally and evolutionarily biased, and conclusions based on these genes may fail to generalize.

3. Fig 2 c-k: it is difficult to interpret these differences between cell types along spermatogenesis because the display is confusing.

(i) There are a mixture of boxplot types (e.g. panels f and g) which both represent distributions, presumably of different types, but I'm not always sure distributions of what (values by genes? averages per cell?). For example, panel d: is that the fraction of 4,742 tested lethal KO that are expressed per cell at each stage? Or is it the fraction of genes expressed at each stage that result in lethal KO (and if so – why are these distributions? is that per cell, despite low numbers of detected genes as noted in comment 1)? Similarly I don't understand how the authors obtain distributions of percentages of positively selected genes in panel f.

(ii) As I don't fully understand how some of these distributions were obtained, it's unclear to me whether the statistical differences in distributions are actually strong, or if they reflect some form of means estimation via data bootstrapping or resampling (in which case, boxplot comparisons can be misleading).

4. In Fig 3, the authors detect genes with either very conserved expression patterns along spermatogenesis during evolution, or on the contrary, which have changed trajectories at different timepoints during primate evolution. This analysis is a bit disappointing because the functional interpretation remains shallow, with just a few example genes highlighted. What are those genes, are there any enriched functions and pathways, are they also expressed elsewhere? The authors speculate in the discussion that these contributed to species-specific phenotypes (l.459) but this is not actually investigated.

5. In Fig 4b, it is unclear to me how genes “predominantly expressed” in one cell type were determined (l.365). This makes it difficult to appreciate the enrichments on the X chromosomes.

Also, are the p-values corrected for multiple testing?

6. Fig 4c-d is in my opinion the most interesting part of the paper, but the analysis of the results is again a bit frustrating because there is so little biological interpretation and the data feels underexploited. I assume that since X and Y-bearing cells can be separated based on transcriptional signatures, they differ on gene expression quite significantly: is this only due to genes on the X and Y chromosomes, or are there differently expressed genes on autosomes as well, which would reflect trans effects linked to the sex chromosome? What are these genes involved into? Are they the same across different species? And why do some X-bearing spermatids harbour Y genes transcripts in their nuclei?

Minor comments

- l. 122: Misleading phrasing: the authors refer to 34 libraries, but it actually accounts for 27 single-nuclei libraries and 7 bulk libraries only used for genome reannotation, which should be clarified.

- l.179: which "other somatic" cell types are these? This should at least be clarified in the legend of Supp Fig 2.

- for simplicity, the orange box in the legend of Fig 1c could be removed since it is not used in the plot

- l.532: names and authorization numbers of human tissue biobanks and companies should probably be provided (to be confirmed by the editor)

- for clarity, the reason for the difference in tissue processing protocols (fixed/unfixed) used for chicken and platypus should be explicitly laid out (I suspect it is due to a change in chemistry in the 10x Genomics Chromium kits used?)

Referee #2 (Remarks to the Author):

This study by Murat and colleagues analyzes 90K single nuclei isolated from the testis using the 10X genomics platform. The 90K cells are comprised of eleven species from multiple mammalian lineages including monotremes, eutherian and marsupial, with chicken being an outgroup. The data generated confirms many early observations made in the field (e.g rapid evolution of spermiogenesis genes, enrichment of spermatogenesis genes on x chromosome, enrichment of infertility genes in early stages, identification of conserved and diverged changes in transcriptome). However, these datasets have enabled a finer assessment of conserved and diverged changes in the gametogenesis program over a shorter evolutionary timescale. Moreover, these datasets revealed the existence of meiotic sex chromosome silencing in monotremes, suggesting that MSCI program predates the split of monotremes and therian mammals. Finally, this analysis allowed the distinct separation of X and Y

bearing spermatids, but this analysis fall short of revealing any interesting biological findings to cells with different chromosomal complement. Taken together, the data is of high quality and would be great utility to the community, but the novelty of findings may be rather limited in its current form.

1) Given the large number of species analyzed, a greater dive in the data is needed to demonstrate functional consequences of transcriptomic differences between species. Given the limitation with using primates– identifying a novel rodent specific program and demonstrating functionality may be important to demonstrate impact or importance of programmatic differences.

2) The snRNAseq allows the identification of Sertoli cells which have been missed by earlier scRNAseq datasets of human and macaque because of the difficulty of sorting intact cells. Here, the authors have an opportunity to glean insights about transcriptomic and signaling differences of Sertoli cells, which may reveal novel insights in the regulation of the gametogenesis program.

3) The ability to distinguish X and Y bearing spermatids is exciting. Are there greater programmatic differences between these spermatids other than the X and y transcripts. Comparing pseudo bulk transcriptomes of these separate cells will be interesting since many chromatin modifiers are detected on either X or Y – which can lead to more global differences in transcriptome.

4) In extended figure 9 – it's curious that Gibbons are the only species where spermatogonia UMI counts are comparable to pachytene cells, is there anything special about Gibbon's? Furthermore, in opossum and chicken zygotene stages appear to be expanded. Is this a color scheme or cell classification issue?

5) MSCI appears to be conserved feature of all mammalian species. Is MSCI as tightly regulated in platypus as it is in mammals? How many genes escape silencing and how does this compare to other species.

6) Large variations in Extended Figure 4 are noted for each of the clustered patterns expression patterns especially in primates as compared to non-primates. Can this be a result of Seurat integration? Has the integration of cell across species checked in an independent manner?

Referee #3 (Remarks to the Author):

Dear authors,

I read the manuscript “The molecular evolution of spermatogenesis across mammals” with great interest. The manuscript is an interesting resource for the community, even though several of the datasets (at least human, mouse, macaque) have been previously compared and recently published (Niederriter Shami 2020). In that sense, I am not sure if the current manuscript provides sufficient novelty and advance in knowledge. However, I am aware of the uniqueness and the novelty of some of the datasets and the computational analysis, although in many aspects relatively standard, is well performed/presented and some of the results are nicely validated, making the results robust.

I have some points that would need clarification:

Figure 2: on the binning, what is the logic to choosing 20 bins? Could you mention the number of cells per bin per species in a Suppl panel? How did you chose a color of the bin if it included cells from several clusters? Could you provide the results regarding the pseudotime per species, colored by cluster and provide there information on the bins there as well. This is an important aspect used through the manuscript and needs to be clarified/presented.

Figure 3: how many bins were used to generate these plots? Should you not use the previous binning for consistency (20 bins)? Perhaps you can consider using the same number of dashed lines across all the plots in all graphs depicting pseudotime-spermatogenesis for consistency?

Also for consistency: I would suggest to use one cartoon for human throughout.

Figure 4: It is not usual to present % X-linked genes, but it would be more informative to have a ratio X/A when discussing MSCI. Could the authors also provide the plots for Figure 4 with the X/A ratio in the main figure and place the % X-linked in the supplementary data?

Figure 4: the use of red arrow to mark (X-bearing and Y-bearing) spermatids and in different graphs to mark spermatocytes in pachytene is confusing. I would suggest to use arrows with different colors to mark the different populations of interest.

Line 696-697: TAGLN and ACTA2 peritubular cells and smooth muscle cells

Ext Data Fig 7c: please use the 20 bins and dashed lines, instead of 4 bins? What do you mean with 'replicates' in the Fig legends?

Ext Data Fig 8: could you explain more extensively in the MM how the 'testis-specific' genes were identified? Was the only threshold $RPKM \geq 1$? What is meant by 'genome wide' gene expression? How was that calculated: is this $RPKM \geq 1$ is each of the organ (brain/heart/kidney/liver) but $RPKM < 1$ in testis?

Ext Data Fig 10: either use the bins or use the clusters...using them interchangeably throughout the manuscript is confusing.

Some Supp Tables are not well formatted (need to have commas instead of dots): ST1, ST2

I would suggest the authors mention the limitations of the study: including number of individuals; limitations of technique and limitations of the computational analysis that the authors may be aware of.

Author Rebuttals to Initial Comments:

We would like to thank the editor and the referees for their positive appreciation of our work and the very useful and
 constructive comments, which allowed us to substantially improve our manuscript.

 To facilitate the second review round, we would like to outline the main novel findings of our original manuscript:

 1. We report an unprecedented single-cell dataset for 11 species, covering all major mammalian lineages and 7 key
 primate species. Our data and results, together with the online resource we have developed
 (<https://apps.kaessmannlab.org/SpermEvol/>) provide an extensive resource for investigating the biology of the
 testis and associated fertility disorders across mammals, in addition to its evolution.
 2. We found that the rapid evolution of the testis is driven by late spermatogenic cell types and identified the
 underlying evolutionary forces and mechanisms: accelerated fixation rates of gene expression changes, amino acid
 altering substitutions, and newly emerged genes. We further provided evidence that this rapid evolution was
 facilitated by reduced pleiotropic constraints, haploid selection, and a transcriptionally permissive chromatin
 environment.
 3. For the first time, we provide a systematic analysis of ancestral and lineage/species-specific spermatogenic gene
 expression programs across amniotes, including a focused analysis on primates (for which we generated data for all
 simian lineages) supported by *in situ* hybridization data. We have identified genes with conserved expression that
 remain uncharacterized and constitute a promising resource for the investigation of fertility phenotypes, and genes
 with changed expression trajectories, which are strong candidates for the molecular basis of species-specific
 spermatogenic phenotypes.
 4. Before our study, it was believed that meiotic sex chromosome inactivation (MSCI) evolved in the common ancestor
 of eutherians and marsupials. We show that MSCI is also present in monotremes, and therefore, across all
 mammalian lineages. This result means that the general mechanism underlying MSCI, meiotic silencing of
 unsynapsed chromatin (MSUC), represents an ancestral mammalian feature.

In our revisions, motivated by the very useful reviewer comments, we strengthened several of our previous findings
 and provide additional novel insights:

 1. We performed additional data and quality controls (e.g., nuclei/UMI downsampling analyses) that show that our
 finding of a higher evolutionary divergence in late spermatogenesis is robust to the variation of the number of cells
 and transcriptome information between cell types/samples. We also validate the batch correction and cross-primate
 integration procedures and clarify what sets of 1:1 orthologous gene were used for the different analyses.
 2. We expanded our investigations of genes with conserved and changed trajectories. Specifically, new GO enrichment
 analyses revealed that genes with conserved expression are frequently involved in fundamental spermatogenic
 processes that are typical of the cell type in which they show peak expression, which supports our original hypothesis
 that genes with highly conserved trajectories are crucial for spermatogenesis. By contrast, genes with changed
 trajectories tend to be involved in broader molecular processes. Notably, our new spatiotemporal expression
 analyses uncovered that reduced pleiotropy likely facilitated the expression changes during evolution.
 3. We now provide a complete list of genes with changed trajectories that are essential for spermatogenesis/fertility
 in mouse.
 4. We investigated the contribution of new genes to the evolution of new spermatogenic functions (on the rodent
 lineage leading to mouse) by combining fertility data with gene age information. This revealed the emergence of key
 spermatogenic genes at different time points during evolution.
 5. We expanded our cross-species analysis of Sertoli cells – the major supporting cell type of spermatogenesis – beyond
 our original X chromosome enrichment analyses. That is, we investigated ligand-receptor interactions underlying
 the communication of Sertoli cells with germ cells. This work revealed various conserved known and potentially
 novel interactions.
 6. We further leveraged our ability to separate X- and Y-bearing spermatids by assessing global transcriptomal
 differences between them. We now also demonstrate that X and Y spermatids cannot be differentiated based on
 single-cell data; this observation also supports the notion of substantial transcript exchange across X/Y spermatids
 via cytoplasmic bridges.
 7. We refined our MSCI findings by showing that MSCI is as complete in monotremes as in the other mammals (i.e., we
 detect zero to one potential transcriptional “escapees” in each of the species). Thus, MSCI emerged from MSUC as
 a highly efficient mechanism of meiotic chromosome silencing across all mammals.

-----
Please find below our detailed responses to the referees' comments; we used the following font emphases/colors to
facilitate readability:

**Comments from the reviewers**

Our response to the reviewers

**Changes to the manuscript (tables, figures, page and line numbers always refer to the revised manuscript)**

**Referee #1 (Remarks to the Author):**

**In this study, Murat, Mbengue and colleagues profile gene expression in single nuclei from testis samples across 11**
**vertebrate species (10 mammals and one bird outgroup), to characterize the evolution of testis cell types. The authors**
**show that overall, testis cell types and their core gene expression programs are conserved, especially in primates.**
**However, late-spermatogenesis cell types have significantly more diverged gene expression programs than early cell**
**types, and seem to underlie the rapid evolution of the testis transcriptome previously described in the literature. The**
**authors confirm that this evolutionary divergence in late cell types is associated with both relaxed and positive**
**selection on expressed genes, expression of many testis-specific genes – some of which are involved in infertility –**
**and generally pervasive transcription. The authors then detect relatively large numbers of genes whose expression**
**has changed during the evolution of primates, and which may be involved in recent testis evolution, although the**
**functional implications of these genes are not investigated. Finally, the last part of the paper focuses on sex-**
**chromosome-specific gene expression. The authors show that X- and Y-bearing spermatids can be distinguished**
**based on their nuclear transcriptome, and testis-specific genes are enriched on the X chromosome. Additionally,**
**male-specific sex chromosome inactivation was detected in sex-differentiated regions of the gonosomes in platypus,**
**suggesting that the process may be ancestral to vertebrates.**

**The paper is well written and easy to follow, with clear messages for each of the main figures. Generally the**
**conclusions seem well-supported by the data, although I have some caveats related to a number of methodological**
**points. Especially, it is not clear to me throughout the paper what fraction of the transcriptomes are actually analysed,**
**and whether the conclusions are representative of the full picture (see detailed comments below). In my opinion, the**
**most exciting and novel aspect is the investigation of sex chromosome differences and MSCI at the end of the**
**manuscript, which will be of interest to a wide readership but would benefit from a more focused and deeper**
**analysis. The rest of the manuscript is a technical tour de force, but the results are more incremental and largely**
**confirm previous results about transcriptome divergence in testis.**

We are very thankful to the referee for their positive feedback. In our revisions, we sought to carve out better the
novelty of our results and conclusions. We also agree that the sex chromosome/MSCI findings are very exciting, and,
therefore, we refined and extended this section.

**Major comments**

**1. Quality control needs to be discussed in more detail, including in the main manuscript. The total numbers of cells**
**per species described in Table S1 are rather low by current standards, and some samples have very low median**
**numbers of detected genes (under 1000). The authors provide no QC per cell type, so it is unclear whether some cell**
**types are underrepresented in the sampling, or suffer from specific quality issues, which may in turn affect the**
**conclusions when the authors compare evolutionary dynamics between cell types.**

This is a fair comment. We now provide more details on our various quality control measures. First, in a new figure
(**Extended Data Fig. 1**, pasted below), which complements Supplementary Table 1, we provide overviews across all
species of the numbers of cells per cell type, UMIs per cell, genes per cell, protein-coding genes per cell type, and
mitochondrial UMIs per cell. This shows, for example, that the percentage of mitochondrial UMIs is low (median of
0.15% per nucleus). We refer to this new figure when introducing the data in the main manuscript (section:
Spermatogenesis across eleven species) and add several summary statistics in the revised text:

(P03, L121-129) “The dataset consists of 27 libraries, with one to three biological replicates per species and a median of ~275 million snRNA-seq reads per library (Supplementary Table 1). We refined and extended existing Ensembl²⁸ genome annotations across all species based on bulk-testis RNA-seq data (7 libraries) (Supplementary Tables 1, 2; Methods), to ensure optimal read-mapping and prevent biases in cross-species analyses. After quality controls and filtering steps (Methods), we obtained transcriptomes for a total of 97,521 high-quality nuclei for the eleven species, with a mean of 8,866 cells per species, a median of 1,856 RNA molecules (unique molecular identifiers, UMIs) detected per cell, and low percentages of mitochondrial UMIs (Extended Data Fig. 1; Supplementary Table 1).”

**Extended Data Fig. 1 | Data overview and quality controls across cell types and species.** a, Numbers of nuclei per cell type. b,
Numbers of UMI per nucleus. c, Numbers of genes per nucleus. d, Numbers of protein-coding genes per cell type. e, Percentages of
mitochondrial UMIs per nucleus. f, Cumulative sum of reads per droplet (top) and fraction of intronic reads per droplet (bottom)
for one human individual (human.2.2); droplets are ranked by the number of reads (decreasing order).

We also improved our description of the quality control procedures in the Methods. For example, in the section
“Identification of usable nuclei”, we refined the description of our custom procedure to identify high-quality nuclei that
includes the consideration of the fraction of intronic reads as a measure of pre-mRNA transcripts (abundant in the
nucleus compared to the cytoplasm) (see also the new **Extended Data Fig. 1f**):

(P18, L687-692) “We used a combined approach for detection of usable nuclei. This was done to optimally account for
the lower RNA content of nuclei compared to the whole cells. Specifically, to identify usable nuclei, we used a knee

point-based approach combined with the fraction of intronic reads as a marker of pre-mRNA transcripts (abundant in
the nucleus) (Extended Data Fig. 1f), and *MALAT1* (nuclear-enriched long non-coding RNA) expression as a marker of
nuclei (when present in the genome of a given species).”

The subsequent section “Quality control of filtered cells”, which cites the new Extended Data Fig. 1, now clearly
describes how we filtered out outlier droplets based on UMI numbers and mitochondrial RNA fractions for the different
testicular cell types, which have very diverse biological characteristics, given substantial differences in size, ploidy, and
RNA content (Soumillon et al. *Cell Rep.* 2013, PMID: 23791531).

(P19, L695-708) “For each sample independently, high quality nuclei were selected removing outliers based on the
number of UMIs and the percentage of mitochondrial RNA (Extended Data Fig. 1b, c, e). We created a Seurat⁸² object
using the Seurat R package v3.1.4 from the subset raw UMI count table generated by Cell Ranger corresponding to the
usable droplets identified upstream, normalized the data using the *NormalizeData* function, identified the top 10,000
most variables genes using the *FindVariableFeatures* function, scaled the data using the *ScaleData* function, performed
the PCA using the *RunPCA* function and calculated the Louvain clusters using the *FindNeighbors* (parameters: dims =
1:20) and *FindClusters* (dims = 1:20, resolution = 0.5) functions. To optimally account for the fact that testis cell types
have diverse transcriptome characteristics¹⁴, we filtered out outlier droplets for each cluster independently with values
lower than the first quartile (Q1) - 1.5 x IQR (InterQuartile Range) and higher than the third quartile (Q3) + 1.5 x IQR for
both the UMI content and the fraction of mitochondrial RNA. Then, we removed potential doublets using
*doubletFinder_v3* function of DoubletFinder⁸³ v2.0.1 (parameters: PCs = 1:20, pN = 0.25, nExp = 5% of the total number
of cells, identifying pk using *paramSweep_v3*, *summarizeSweep* and *find.pk* functions).”

We would like to note that the testis (spermatogenesis) is less complex in terms of cell types than other organs, such as
the brain, and thus requires fewer cells to adequately cover its constituent cell types. Indeed, our data allowed us to
identify all major cell types, even those (Sertoli cells, except for platypus, Leydig cells) that were missed in other studies
(e.g., Shami et al. 2020 for human and macaque).

The numbers of expressed genes detected per cell (median: 1,599) in our study is lower than that typically seen in single-
cell RNA-seq experiments, but that is typical of single-nucleus experiments. However, to this point we note that our
work is based on pseudo-bulk analyses of cell types, and the number of expressed genes detected per cell type (mean:
~15K) is similar across cell types and species (Extended Data Fig. 1d). Importantly, cell type pseudo-bulks – including
those with lower median numbers of detected genes – show biologically/phylogenetically consistent clustering in our
PCA and gene expression tree analyses (Figs. 1b, c and Extended Data Fig. 3), which demonstrates that the data are
comparable and of high quality (P05, L163-164).

Nevertheless, we sought to further show that the variation in the numbers of cells and transcriptome information
between cell types/samples does not affect our conclusions regarding the differences in evolutionary dynamics between
cell types. To do so, we did subsampling analyses. We downsampled the data to the same number of cells and UMIs
across all cell types for human, macaque, and mouse. We chose these three species to ensure we had a sufficient
number of expressed 1:1 orthologous genes. We then carried out cross-species correlation analyses akin to those
presented in the original manuscript (Fig. 2a, b), to assess the evolutionary divergence of cell types. The analyses with
the downsampled data show the same pattern of increasing divergence in late spermatogenesis (see new Extended
Data Fig. 3b) as observed on the whole data set, thus confirming our original conclusion. We mention this result and
the new Extended Data Fig. 3b in the revised manuscript in the section “Rates of evolution along spermatogenesis”:

(P05, L186-189) “Pairwise species comparisons, including downsampling analyses (Extended Data Fig. 3b), confirm the
rapid expression evolution of post-meiotic cell types across amniotes and that gene expression divergence increases
with evolutionary time (Fig. 2b), in accord with the expression phylogeny results (Extended Data Fig. 3a).”

**Extended Data Fig. 3: Mammalian testicular gene expression divergence.** **a**, Gene expression phylogenies. Neighbor-joining trees
 based on pairwise expression level distances (1-rho, Spearman's correlation coefficient) for the main testicular cell types,
 respectively ("Other somatic" refers to peritubular, smooth muscle, endothelial, macrophage, and Leydig cells as detailed in the
 legend of Extended Data Fig. 2 and Methods). Trees are drawn to the same scale (indicated by the scale bar). Bootstrap values (i.e.,
 proportions of replicate trees that have the branching pattern as in the majority-rule consensus tree shown) are indicated by circles
 at the corresponding nodes: ≥ 0.9 (white fill). **b**, Human-macaque (left) and human-mouse (right) pairwise Spearman correlations
 after downsampling to the same number of nuclei and UMI across cell types for each species.

**The authors also mention using Seurat to remove batch effects within species, but it is unclear how strong this batch**
 **effect was, and whether this again may affect interpretations. How correlated are technical/biological replicates on**
 **average?**

As detailed below, the differences between batches (i.e., replicate samples) are small and the data from Seurat's CCA
 batch corrections were only used to assign cell type labels across biological/technical replicates. Gene expression
 differences between replicates were fully retained and considered in all analyses of the paper. Hence, none of our
 analyses are biased by Seurat's batch correction.

In the correlation-based expression tree for pseudo-bulk transcriptomes for whole testes (Fig. 1c), biological replicates
 consistently cluster with each other in a phylogenetically sensible way (that is, intra-species differences are smaller than
 inter-species differences), with the exception of the samples for chimpanzee and bonobo, which are very closely related
 species. This observation illustrates that replicates have very similar expression signals and that the data are overall of
 high quality (P05, L168-170). To strengthen this point, we now also quantify the correlations between replicates for all
 species (new Extended Data Fig. 18a); the high correlation coefficients (median ρ : ~ 0.89), illustrated for humans in
 Extended Data Fig. 18b, confirm that replicates are overall very similar. Moreover, importantly, in the expression tree
 analyses for individual cell types and the PCA analysis (Fig. 1b, c; Extended Data Fig. 3), the replicates also consistently
 cluster with each other in a phylogenetically sensible manner, demonstrating that they have very similar expression
 signals also at the level of individual cell types.

To evaluate the effect of Seurat's CCA batch correction, we now show for humans the UMAPs of the individual samples,
 and the merged data before and after batch correction (Extended Data Fig. 18c, d). This shows that the replicates already

integrate well before the procedure and that the batch correction only slightly improves the integration. This is
 consistent with the high correlations found across replicates. We also assessed the expression of key cell type markers
 across the different replicates using the integrated object coordinates (Extended Data Fig. 18e), which shows that the
 marker genes are expressed in the same integrated areas across replicates and thus support the notion that the
 integration is correct.

 **Extended Data Fig. 18 | Batch comparisons.** **a**, Distribution of Spearman correlation coefficients (ρ) for pairwise comparisons
 between replicates for all species. **b**, Comparisons of gene expression levels (mean level for each gene across all cells of the sample)
 between human ind.1/rep.1 and the other replicates. **c**, UMAP of merged human samples before batch correction (left) and individual
 UMAPs of each independent human sample using the merged coordinates (right). **d**, UMAP of merged human samples after batch
 correction (left) and UMAPs of each independent human samples using the merged UMAP coordinates (right). **e**, Marker gene
 expression across human replicates using the merged UMAP coordinates after batch correction.

We cite and discuss this Figure in the Methods section “Integration of datasets”:

(P19, L724-728) “We note that – consistent with the high correlation between biological replicates (Extended Data Fig.
18a, b) – the data already integrate well before the batch correction (Extended Data Fig. 18c, d). We also note that key
marker genes are expressed in the same integrated areas across replicates when assessing their expression in the
different replicates using the integrated object coordinates (Extended Data Fig. 18e), which supports that the
integration is correct.”

Finally, we would like to reiterate that the merging was only done for cell type assignments and that gene expression
values remain unmodified in all of the analyses of the paper. Thus, expression variation between biological replicates,
which overall provide highly consistent results, are considered in the different analyses of our study (i.e., in all trajectory
analyses, gene expression trees, PCA and now also more clearly for the different metrics plotted in Fig. 2 and
corresponding Extended Data Fig. 4 – see response to comment 3).

**2. Due to the nature of this work, the authors focus on 1-to-1 ortholog genes for transcriptomic comparisons across**
**species. For all 11 vertebrate species, this effectively reduces transcriptomes to ~5,000 coding genes, less than 25%**
**of human genes, as described in Fig 1. It was not clear to me whether all analyses in the manuscript use this same set**
**of ~5,000 genes, or whether some parts (comparisons within primates, etc) use more extensive datasets. This is a**
**potential issue because deeply conserved 1-to-1 orthologs are functionally and evolutionarily biased, and conclusions**
**based on these genes may fail to generalize.**

This is a very important point and we should have been clearer about this in our original manuscript. Thank you for
pointing this out!

We used four sets of 1:1 orthologous genes, tailored for the different analyses, exactly for the reason advanced by the
reviewer:

- • Comparative analyses involving the 11 amniote species were performed using 4,498 1:1 orthologues that are
expressed across all species (among a total of 8,045 1:1 orthologues).
- • Comparative analyses involving only the 7 primates were performed using 8,451 1:1 orthologues expressed
across all primate species (among 11,948 1:1 orthologues).
- • Comparative analyses for the new Sertoli-germ cell communication analyses (please refer to our response to
comment 2 from referee #2 for details) was based on mapping 35,186 human testis-expressed genes to 1:1
orthologous genes in the other species (macaque: 13,090; mouse: 14,302; opossum: 10,865; and chicken:
10,515).
- • Species-specific analyses were performed using all genes expressed in a given species (~15k genes per cell
type).

We now clearly outline the different gene sets in the figure legends and in a new Methods section “Orthologous gene
sets”:

(P20, L776-785) “We used four different sets of orthologous genes in our study: 1) Comparative analyses involving all
11 amniote species were performed using 4,498 1:1 ortholog genes that are expressed (i.e., one UMI in at least three
cells of any cell type) across all species (among a total of 8,045 1:1 orthologues). 2) Comparative analyses involving the
7 primate species were performed using 8,451 1:1 ortholog genes expressed across all primate species (among 11,948
1:1 orthologues). 3) The comparative Sertoli-germ cell communication analysis was based on mapping 35,186 human
testis-expressed genes to 1:1 orthologous genes in the other species (macaque: 13,090; mouse: 14,302; opossum:
10,865; and chicken: 10,515). 4) Species-specific analyses were performed using all genes expressed in a given species
(~15,000 genes per cell type; Extended Data Fig. 1d). Orthologous gene sets were extracted from Ensembl²⁸ using the
biomaRt R package v2.40.5.”

**3. Fig 2 c-k: it is difficult to interpret these differences between cell types along spermatogenesis because the display**
**is confusing.**

(j) There are a mixture of boxplot types (e.g. panels f and g) which both represent distributions, presumably of
 different types, but I'm not always sure distributions of what (values by genes? averages per cell?). For example,
 panel d: is that the fraction of 4,742 tested lethal KO that are expressed per cell at each stage? Or is it the fraction of
 genes expressed at each stage that result in lethal KO (and if so – why are these distributions? is that per cell, despite
 low numbers of detected genes as noted in comment 1)? Similarly I don't understand how the authors obtain
 distributions of percentages of positively selected genes in panel f.

Thanks for pointing this out, which – together with a comment from reviewer 2 – allowed us to clarify these aspects by
 optimizing the plots and legends of Fig. 2 and the corresponding Extended Data Fig. 4, and by adding additional figures.

We now plot the different metrics per cell type, given that the previous binning added unnecessary complexity to Fig.
 2/Extended Data Fig. 4 (e.g., the presence of cells of different types within a bin) and explain precisely what is plotted.
 We now also provide the corresponding numbers of cells per cell type underlying the different plots in Fig. 2 in the new
 Extended Data Fig. 1a (see also response to comment 1).

 **Fig. 2 | Gene expression divergence and evolutionary forces.**

With respect to what exactly is plotted: in Fig. 2c-k (with the exception of Fig. 2 i) the values [mean (panels c, e, g, j) or
 percentage (d, f, h, k)] for the different measures are indeed plotted per cell for all cells of a given cell type (i.e., the box
 plots represent distributions of values across cells), considering all expressed genes in each cell. In Fig. 2i, however,
 translational efficiency (TE) values of all genes predominantly expressed in a given cell type are plotted (i.e., the box
 plots represent distributions of values across genes).

More specifically:

- • Fig. 2c: the mean pLI (mutational tolerance) score of all expressed genes per cell is plotted for all cells of a given
 cell type.

- • Fig. 2d: the percentage of expressed genes that lead to a lethal phenotype when knocked out in mouse (out of
4,742 knockouts) per cell is plotted for all cells of a given cell type.
- • Fig. 2e: the mean dN/dS ratio (normalized ratio of nonsynonymous over synonymous substitutions) of all
expressed genes per cell is plotted for all cells of a given cell type.
- • Fig. 2f: the percentage of expressed genes under positive selection per cell is plotted for all cells of a given cell
type.
- • Fig. 2g: the mean phylogenetic age of expressed genes per cell is plotted for all cells of a given cell type. Higher
values indicate larger contributions of lineage-specific genes (that is, younger transcriptomes).
- • Fig. 2h: the percentage of UMIs mapping to protein-coding genes (top) or intergenic elements (bottom) per
cell is plotted for all cells of a given cell type.
- • Fig. 2i: the TE values for all genes with predominant expression in a given cell type (i.e., genes whose expression
peak in that cell type, as determined in the trajectory analyses) are plotted. We note that, in this panel, we plot
values across genes for a given (pseudobulk) cell type, given that the TE values underlying this plot were derived
from bulk spermatogenic cell type data in our previous study (Wang et al. *Nature* 2020, PMID: 33177713).
- • Fig. 2j: the mean of tissue specificity values of all expressed genes per cell is plotted for all cells of a given cell
type.
- • Fig. 2k: the percentage of expressed genes associated with infertility per cell is plotted for all cells of a given
cell type.

To resolve the lack of clarity, we optimized the legend of Fig. 2 (and adapted that of corresponding Extended Data Fig.
4, which now refers to Fig. 2 for details of the measures):

(P06, L199-220) “**Fig. 2 | Gene expression divergence and evolutionary forces.** **a**, Comparisons of total branch lengths
of expression trees among the six main testicular cell types for amniotes and primates. Errors, 95% confidence intervals
based on bootstrapping analyses (1,000 replicates). **b**, Spearman’s correlations between humans and other species.
Dots correspond to values obtained in 100 bootstrap replicates. The lines correspond to linear regression trends (after
log transformation of the time axis) and were added for visualization purposes. Regression R^2 values range from 0.86 to
0.97. **c**, Boxplots showing the mean pLI value of all expressed genes per cell for all cells of a given cell type in human
testis (the pLI score³⁰ reflects the tolerance of a gene to a loss-of-function mutation; lower values mean less tolerance).
**d**, The percentage of expressed genes that lead to a lethal phenotype when knocked out in mouse (out of 4,742
knockouts³¹) per cell is plotted for all cells of a given cell type. **e**, The mean normalized ratio of nonsynonymous (dN)
over synonymous (dS) nucleotide substitutions of all expressed genes per cell is plotted for all cells of a given cell type
in macaque. **f**, The percentage of expressed genes under positive selection (out of 11,170 genes tested for positive
selection¹¹) per cell is plotted for all cells of a given cell type in chimpanzee. **g**, The mean phylogenetic age of expressed
genes per cell is plotted for all cells of a given cell type in mouse. Higher values indicate larger contributions of lineage-
specific genes (i.e., younger transcriptomes). **h**, The percentage of UMIs mapping to protein-coding genes (top) or
intergenic elements (bottom) per cell is plotted for all cells of a given cell type in gorilla. **i**, Translational efficiency (TE)
values (data from ref. ⁴) are plotted for all genes with predominant expression in a given cell type (i.e., expression peaks
in that cell type) in human. **j**, The mean of tissue specificity values (data from ref. ⁴) of all expressed genes per cell is
plotted for all cells of a given cell type in opossum. **k**, The percentage of expressed genes associated with infertility (out
of 3,552 knockouts) per cell is plotted for all cells of a given cell type in mouse. Superimposed thick black dots in **c-k**
indicate individual medians for cells from different biological replicates. Box plots depict the interquartile ranges, with
whiskers at 1.5 times the interquartile range. Red vertical lines separate somatic (OS: other somatic; ST: Sertoli cells)
and germ cells. Corresponding data for all other studied species are shown in Extended Data Fig. 4.”

We also refined the corresponding Methods section “Evolutionary forces” (P21, L805-853).

**(ii) As I don't fully understand how some of these distributions were obtained, it's unclear to me whether the**
**statistical differences in distributions are actually strong, or if they reflect some form of means estimation via data**
**bootstrapping or resampling (in which case, boxplot comparisons can be misleading).**

As now clarified (above), the plots are not based on bootstrapping or resampling.

The differences between cell types – especially between early and late spermatogenic cells (the main message in this
section) – are strong (non-overlap of notch areas and/or IQRs) and, importantly, highly robust and replicable. That is,
for all measures that can be assessed across species, all species show the same overall patterns (cross-species
replicability) (Fig. 2e-j and Extended Data Fig. 4b-h). In addition, as is visible in the revised Fig. 2 and associated legend
(see above), we now superimpose the median for all biological replicates (i.e., samples from different individuals) onto
the box plots (black dots); that is, the box plots represent distributions of cells (or specifically expressed genes in the
case of Fig. 2i) from all replicates, but we now also superimpose individual medians calculated for each replicate
separately. This shows that medians from different replicates are very similar and support very similar cross-
spermatogenic patterns for measures that could only be assessed in specific species (human or mouse) (i.e., pLI, lethals,
infertility; Fig. 2c,d,k).

**4. In Fig 3, the authors detect genes with either very conserved expression patterns along spermatogenesis during**
**evolution, or on the contrary, which have changed trajectories at different timepoints during primate evolution. This**
**analysis is a bit disappointing because the functional interpretation remains shallow, with just a few example genes**
**highlighted. What are those genes, are there any enriched functions and pathways, are they also expressed**
**elsewhere? The authors speculate in the discussion that these contributed to species-specific phenotypes (I.459) but**
**this is not actually investigated.**

We are grateful for the reviewer's comment – it motivated us to substantially expand on our previous functional
investigations of genes with conserved or changed trajectories, which provided interesting new insights, as detailed
below.

We originally supported our hypothesis that genes with highly conserved trajectories across mammals are crucial for
spermatogenesis by showing that genes leading to infertility when knocked out in the mouse are more conserved across
species than genes where fertility remains unaffected (Extended Data Fig. 8b). To strengthen this hypothesis and more
globally explore the functional roles of highly conserved spermatogenic genes, we screened for enriched processes
based on Gene Ontology (GO) annotations, categorizing genes according to the cell type in which their expression peaks.
This analysis implicates genes with conserved trajectories in fundamental spermatogenic processes that are typical of
the cell type in which they show peak expression (new Extended Data Fig. 9a, b). For example, conserved genes with
peak expression in spermatocytes are enriched for GO terms related to meiotic functions, including stage-specific ones
(e.g., in meiosis I, which starts in leptotene spermatocytes).

By contrast, a GO analysis for genes with trajectory changes in primates and another in amniotes, revealed enrichments
of broader molecular processes (i.e., non-spermatogenesis-specific), especially those related to metabolic processes
(Extended Data Fig. 9c, d).

To further explore the functional differences of genes with conserved and changed trajectories, we assessed their spatial
and developmental (i.e., temporal) expression patterns using our dataset of gene expression across mammalian
development for seven organs (Cardoso-Moreira et al. *Nature* 2019; PMID: 31243369). This analysis uncovered that
genes with changed expression are significantly more tissue-and time-specific than genes with conserved expression
(Extended Data Fig. 9e). This suggests that the reduced pleiotropy of changed genes – and hence the reduced purifying
selection acting on them – may have facilitated their expression change during evolution.

Extended Data Fig. 9 | Functions of conserved and changed trajectories. **a**, Top 20 enriched biological process GO terms of genes showing conserved expression trajectories across primates and peak expression in the different cell types, respectively. **b**, Top 20 enriched biological process GO terms of genes showing conserved trajectories between human and mouse and peak expression in the different cell types, respectively. **c**, The top 20 enriched biological process GO terms of genes with trajectory changes in primates. **d**, The top 20 enriched biological process GO terms in genes showing trajectory changes in amniotes. **e**, Human tissue (left) and time (right) specificity for changed and conserved trajectories across amniotes, respectively (two-sided Wilcoxon rank-sum tests were performed for statistical comparisons).

We now also provide a complete list of genes with changed trajectories in amniotes that are essential for spermatogenesis/fertility in mouse, as assessed by mouse KO studies (IMPC database), in the new **Supplementary Table 8**. This list includes striking mouse/rodent-specific trajectory changes that presumably led to new essential functions of these genes in mouse spermatogenesis. We now discuss one example, *SPIDR*, a gene whose expression switched from an ancestral early/spermatogonial to late spermatid expression during evolution (P10, L347-349; see also revised paragraph further below, **Supplementary Table 8**). Knockout of this gene leads to reduced testis size and a paucity of post-meiotic cells (Prakash et al. *Nat Commun.* 2021; PMID: 34253720).

Finally, we now also explore the contribution of new genes to the evolution of new spermatogenic functions (on the rodent lineage leading to mouse) by combining the fertility information from the IMPC database with gene age

information. This revealed the emergence of key spermatogenic genes at different time points during evolution
 (Extended Data Fig. 10a; Supplementary Table 8). The two identified rodent-specific genes – *D1Pas1* and *H2al2a* –
 represent interesting cases. They are both intronless retrogenes that originated via an RNA-based duplication from
 parental genes on the X chromosome, consistent with the view that the X has spawned a disproportionately large
 number of new retrogenes with important spermatogenesis functions during mammalian evolution (Necsulea and
 Kaessmann *Nat. Rev. Genet.* 2014; PMID: 25297727) (Extended Data Fig. 10b). *D1Pas1* encodes a DEAD-box RNA
 helicase essential for meiosis (Inoue et al. *Biochem. Biophys. Res. Commun.* 2016; PMID: 27473657), while *H2al2a*
 encodes a histone variant required for genome compaction around protamines (Barral et al. *Mol. Cell* 2017; PMID:
 28366643). Both genes show strongly increasing expression levels in late spermatogenesis (Extended Data Fig. 10c) and
 are thus in agreement with our notion that new duplicate genes contribute functional roles predominantly in later
 stages of spermatogenesis (Fig. 2g; Extended Data Fig. 4d; P10, L352-360). Our data also revealed various other
 interesting genes that evolved key spermatogenic functions during evolution; for example, the duplicate gene *Spink2*
 (encoding a serine protease inhibitor) emerged in the common eutherian ancestor and evolved an essential role in
 spermiogenesis (acrosome formation) (Kherraf et al. *EMBO Mol. Med.* 2017; PMID: 2854943).

 **Extended Data Fig. 10 | New genes with crucial spermatogenic functions.** **a**, percentages of genes among new genes of different
 ages (i.e., emergence in last common vertebrate, tetrapod, mammalian, eutherian, or rodent ancestors) that show infertility
 phenotypes when knocked out in the mouse (numbers of genes with infertility phenotypes and total number of genes considered
 for an age category are provided above each bar). **b**, Expression levels of the retrogenes *H2al2a* and *D1Pas1* in spermatogenic and
 somatic cell types. Error bars correspond to the range between minimum and maximum expression values across biological
 replicates. **c**, Trees of *H2al2a* and *D1Pas1* and paralogs from which they originated through the mechanism of RNA-based gene
 duplication (i.e., *D1Pas1* stems from *Ddx3y*, whereas the precise ancestral parental gene of *H2al2a* is not easily discernable due to a
 complex history of a number of RNA- and DNA-based duplication events).

 We now present all of the aforementioned new analyses and results in the section now termed “Gene expression
 conservation and innovation”, which now also includes an investigation of cell-cell communication across species (last
 pasted paragraph below), with a focus on Sertoli cells (please refer to our response to comment 2 from referee #2 for
 details):

 (P10, L330-376) “In agreement with these highly conserved sets of genes playing key roles in mammalian/amniote
 spermatogenesis, our analyses of fertility phenotypes³¹ unveiled that genes involved in fertility are significantly more
 conserved in their expression trajectories than genes not associated with fertility (Extended Data Fig. 8b). Consistently,
 a Gene Ontology⁴² enrichment analysis indicates an involvement of genes with conserved trajectories in fundamental
 spermatogenic processes that are typical of the cell type in which they show peak expression (Extended Data Fig. 9a,
 b). Thus, genes with conserved trajectories for which spermatogenesis functions remain uncharacterized represent a
 promising resource of candidates for the exploration of fertility phenotypes (Supplementary Tables 5, 7, 8). Notably,
 genes with lineage-specific trajectory changes are enriched with broader, typically metabolic, processes (Extended Data
 Fig. 9c, d). They are also significantly more tissue- and time-specific than genes with conserved expression (Extended
 Data Fig. 9e), which may have facilitated their expression change during evolution because of reduced pleiotropic

constraints. However, genes with changed trajectories nevertheless include many genes for which key fertility functions
 have been described (Supplementary Tables 7, 8). For example, *IP6K1*, which elicits infertility when knocked out in the
 mouse⁴³, shows strongly increasing expression towards the end of spermiogenesis in all amniotes except the primates,
 where expression is high in SG and then overall declines (Fig. 3b; Extended Data Fig. 8c). Thus, the primary function of
 *IP6K1* may have shifted from late to early spermatogenesis during primate evolution. An example of an essential
 spermatogenic gene with a rodent-specific change is *SPIDR*⁴⁴, whose expression switched from an ancestral
 early/spermatogonial to late spermatid expression during evolution (Supplementary Tables 7, 8).

 In conjunction with mouse fertility data, we used our data to explore the contribution of new genes (mostly arising from
 gene duplications) to the evolution of new spermatogenic functions, focusing on the rodent lineage leading to mouse.
 This analysis revealed the emergence of key spermatogenic genes at different time points during evolution (Extended
 Data Fig. 10a; Supplementary Table 8), such as two rodent-specific retrogenes – *D1Pas1* and *H2al2a* – that originated
 through RNA-based duplication from parental genes on the X chromosome (Extended Data Fig. 10c). *D1Pas1* is essential
 for meiosis⁴⁵, while *H2al2a* is essential for genome compaction in late spermatogenesis⁴⁶. Both genes show strongly
 increasing expression levels in late spermatogenesis (Extended Data Fig. 10b) and are thus in agreement with new genes
 contributing to functional roles predominantly during late spermatogenesis (see above; Fig. 2g; Extended Data Fig. 4d).
 Other examples include the paralog *Spink2* (Supplementary Table 8), which emerged in the common eutherian ancestor
 and also evolved an essential role in late spermatogenesis⁴⁷.

 Finally, we used our data to investigate ligand-receptor interactions underlying the communication between Sertoli
 cells, which have a central (paracrine) role in supporting and controlling spermatogenesis⁴⁸, and germ cells across
 species using CellPhoneDB⁴⁹. Our cross-species comparisons revealed various conserved known and novel ligand-
 receptor interactions (Extended Data Fig. 11a, b and Supplementary Table 9). For example, our data provide evidence
 that communication of Sertoli cells with SG, SC, and rSD occurs in all amniotes through interactions of the cell adhesion
 molecule *CADM1*⁵⁰ or between *CADM1* (Sertoli cells) and *NECTIN3* (SC/rSD) (Supplementary Table 9). We note that
 *CADM1* was previously thought to not to be expressed in Sertoli cells⁵¹, but our data reveals high expression in this cell
 type across amniotes, in agreement with human protein atlas data (proteinatlas.org/ENSG00000182985-
 *CADM1*/tissue/testis#). Our work also supports the notion⁵² that the *NECTIN2*-*NECTIN3* complex mediates the
 communication of Sertoli cells with spermatids not only in mouse⁵³ but also in humans (Supplementary Table 9).”

 We have also modified the relevant part in the discussion to reflect the new results. We now explicitly state that we did
 not directly (experimentally) demonstrate the contribution of individual trajectory changes to the emergence of species-
 491 /lineage-specific phenotypes. We share with the reviewer the desire for this functional work, but it is beyond the scope
 of this specific study.

 (P14, L497-502) “Our cross-species comparisons of individual genes revealed temporal expression differences across
 species, including human-specific changes, which were probably facilitated by reduced pleiotropic constraints. Our
 results thus provide an extensive list of candidates whose contributions to the evolution species-specific
 spermatogenesis phenotypes can be experimentally scrutinized. We also uncovered conserved expression programs
 underlying spermatogenic processes ancestral to individual mammalian lineages and mammals as a whole.”

 **5. In Fig 4b, it is unclear to me how genes "predominantly expressed" in one cell type were determined (I.365). This**
 **makes it difficult to appreciate the enrichments on the X chromosomes. Also, are the p-values corrected for multiple**
 **testing?**

 Thanks for pointing out that this was not clear.

 We first extracted genes that are specific to the testis (RPKM ≥ 1 in testis and RPKM < 1 in brain, cerebellum, heart,
 kidney, and liver, as assessed based on RNA-seq data from Cardoso-Moreira et al. *Nature* 2019; PMID: 31243369). For
 germ cells, predominant expression of a gene within the testis was then identified based on the trajectory analyses;
 that is, predominant expression was assigned based on the cell type in which the expression level of the gene peaks.
 For somatic cell types, we performed a differential gene expression analysis using a Wilcoxon Rank Sum test
 (FindAllMarkers function from the Seurat package) to identify the cell type in which the gene is predominantly
 expressed; that is, we compared expression of a gene in each somatic cell type, respectively, with its average expression
 in all other cell types.

We also thank the reviewer for reminding us of the need for multiple test correction in these analyses. We have now adjusted all *P*-values using the Benjamini-Hochberg method. The Sertoli/spermatogonial enrichment patterns of X-linked genes remains significant after this correction, with the exception of human Sertoli cells (Fig. 4b).

Figure 4b | Mammalian sex chromosome evolution.

We now clarified these aspects in the legend of Fig. 4b:

(P12, L406-411) “...b, Percentages of X-linked genes among testis-specific genes with predominant expression in a given cell type (i.e., expression peaks in that cell type) for human, macaque, mouse, opossum, and platypus (from left to right). The red horizontal dashed line represents the expected percentages of X-linked genes, if testis-specific genes with predominant expression in the different cell types were randomly distributed across the genome. Exact binomial tests were performed for statistical comparisons (Benjamini-Hochberg corrected $*P < 0.05$, $***P < 0.001$, $****P < 0.0001$).”

We also explained our definition of predominant expression in the Methods:

(P24, L935-942) “Testis-specific genes were obtained from previously generated RNA-seq data⁴ of adult organs (RPKM ≥ 1 in testis and RPKM < 1 in brain, cerebellum, heart, kidney, and liver). Among these, cell-type specific genes were studied for each chromosome. Genes with predominant expression in specific somatic cell types were identified using the *FindAllMarkers* function (parameter: only.pos = TRUE, min.pct = 0.05, logfc.threshold = 0.25, return.thresh = 0.05). Predominant expression of genes in specific germ cell types was assigned based on the trajectory analyses (see above); that is, predominant expression was assigned based on the cell type in which the expression level of the gene peaks in the trajectory analysis.”

6. Fig 4c-d is in my opinion the most interesting part of the paper, but the analysis of the results is again a bit frustrating because there is so little biological interpretation and the data feels underexploited. I assume that since X and Y-bearing cells can be separated based on transcriptional signatures, they differ on gene expression quite significantly: is this only due to genes on the X and Y chromosomes, or are there differently expressed genes on autosomes as well, which would reflect trans effects linked to the sex chromosome? What are these genes involved into? Are they the same across different species? And why do some X-bearing spermatids harbour Y genes transcripts in their nuclei?

We much appreciate (and share) the enthusiasm regarding this section. Therefore, we have expanded the analyses regarding MSCI and the transcriptional differences between X- and Y-bearing spermatids. We originally sought to exploit the separation of X- and Y-bearing spermatids to separately and thus cleanly assess the expression of X- and Y-linked genes across spermatogenesis, which allowed us to detect MSCI across mammals at optimal resolution. Please note in this context the new plots in Fig.4d (see below) and Extended Data Fig. 16a, which were made in response to a comment from referee #3 and now even better highlight patterns of MSCI, especially in platypus, by showing X/Y transcript abundances directly for all individual germ cells.

**Figure 4d | Mammalian sex chromosome evolution**

We also assessed the completeness of MSCI in platypus and the other species, motivated by a comment from referee
 #2. We did so based on a modification of a previous approach (Shami et al. *Dev. Cell* 2020, PMID: 32504559), in which
 transcript abundances of genes before and during MSCI are compared. Genes with an increased abundance in cells
 around the pachytene stage compared to the preceding spermatogonial stage are inferred to reflect active transcription
 during MSCI, as opposed to potential carry-over of stable transcripts, which may also occur. Specifically, we modified
 the previous approach by Shami et al. by assessing the statistical significance of gene expression increases at
 spermatocyte stages subject to MSCI (relative to the spermatogonia stage) using a differential expression analysis
 implemented in Seurat (Methods: P25, L983-994).

Our analyses based on this procedure revealed no human and only one potential mouse MSCI escapee (new **Extended**
 **Data Fig. 17**), in agreement with the notion that MSCI is complete in these species (Sin and Namekawa *Epigenetics* 2013;
 PMID: 23880818). For the opossum, we also only detected one potential candidate (**Extended Data Fig. 17**), which
 suggests that MSCI is as complete in marsupials as in eutherians, in agreement with previous work based on an
 experimental analysis of ten genes (Mahadevaiah et al. *Curr. Biol.* 2009; PMID: 19716301). Finally, and notably, for the
 platypus sexually differentiated regions (SDR), we also only identify one potential escapee (**Extended Data Fig. 17**). This
 observation nicely contrasts with that observed for the pseudoautosomal regions (PARs), where many genes show
 predominant expression in cells subject to MSCI (Fig. 4d) – a further illustration of the strong SDR-specific MSCI pattern
 in this species. Overall, our findings imply that MSCI is as complete in monotremes as in therian mammals and that MSCI
 (and hence MSUC) was already highly efficient in the common mammalian ancestor.

**Extended Data Fig. 17. Assessment of the completeness of MSCI across species.** To identify potential MSCI escapees, we screened
 for X-linked genes in SDRs with a significant increase in transcript abundance from SG to SC stages subject to MSCI, which ensures
 that potential escape genes are indeed actively transcribed in SC (i.e., they do not merely represent genes expressed in SG with stable
 transcripts still detectable in SC) (Methods). Expression differences (\log_2 -fold change, x-axis) of X and autosomal genes between SC

and SG were assessed using differential expression analysis (Methods; p -values on the Y-axis are Bonferroni-corrected). Vertical red
lines indicate a 0.25-fold expression change (log₂-scale); horizontal red lines indicate corrected $P < 0.05$. Potential escapees (i.e., X-
linked genes in SDRs with significantly higher expression in SC than SG) are indicated by red arrows.

We now describe these new results at the end of the section “Sex chromosomes”:

(P14, L469-480) “Indeed, while the joint analysis of all platypus X-linked genes only reveals a small expression dip around
the pachytene stage (Fig. 4d, upper platypus graph), an analysis only of SDR genes reveals a strong reduction of X
transcript levels. By contrast, PAR genes show stable expression levels across spermatogenesis (Fig. 4d, lower graph).
Moreover, the difference in transcript abundances between SDRs and PARs due to MSCI is visible for all five platypus X
chromosomes (Extended Data Fig. 16b). Notably, our assessment of the completeness of MSCI across species reveals
that platypus MSCI is as complete as that in other species for which there is little or no MSCI escape^{27,64,65} (i.e., we only
detected zero or one potential transcriptional escapee in each of the species, Extended Data Fig. 17). The presence of
MSCI at the SDRs in monotremes is consistent with the partial association of platypus sex chromosomes with
perinucleolar repressive histone modifications at the pachytene stage⁹. Altogether, our data reveal that efficient MSCI
is common to all mammalian sex chromosome systems, which implies that the general mechanism of MSUC is an
ancestral mammalian feature.”

We agree with the reviewer that assessing global transcriptional differences between the two types of spermatids is
also intriguing.

We thus proceeded with a differential expression analysis for X- vs. Y-bearing spermatids, which revealed, as expected,
a majority of sex-chromosome genes, including interesting gametologs (i.e., genes with homologous counterparts on X
and Y chromosomes), such as the translational regulators DDX3X/DDX3Y (new Extended Data Fig. 15; Supplementary
Table 11). Interestingly, however, we also find many autosomal genes to be differentially expressed, especially in human
and opossum Y spermatids (for mouse, we generally detected few expression-biased genes and in platypus all biased
genes but one are X-/Y-linked), which might reflect trans regulatory effects associated with the sex chromosomes. The
vast majority of cases are specific to the analyzed species, suggesting a rapid evolutionary turnover of spermatid
differential expression during mammalian evolution. Notably, many of the differentially expressed genes, including the
most significant cases, are putatively noncoding, which is noteworthy, because long noncoding RNAs (lncRNAs) are
usually localized in the nucleus (Derrien et al. *Genome Res.* 2012; PMID: 22955988) and hence their differential
expression levels are unlikely to be offset by transcript exchange between spermatid cells via cytoplasmic bridges. The
three most Y-spermatid-specific transcripts are lncRNAs emanating from homologous low copy repeats (LCRs; i.e.,
segmental duplications) on chromosomes 13 (FAM230C), 21 (XLOC-095504), and 22 (FAM230F) that cause genomic
disorders by triggering nonallelic homologous recombination (NAHR) events. They include the FAM230F lncRNA in the
q11.2 LCR region on chromosome 22 (22q11.2) that is particularly susceptible to NAHR-generated deletions that lead
to various congenital malformation disorders, including the DiGeorge syndrome, the most frequent microdeletion
disorder⁹⁷.

referee #1

**Extended Data Fig. 15 | Transcriptome differences between X and Y bearing spermatids.** The heatmaps indicate transcript
abundances of significantly enriched genes in X- and Y-bearing spermatids in the different species (a: human, b: mouse, c: opossum,
627 d: platypus). The corresponding bar plots (genes in the same order) show the genes with significantly enriched expression in X- and

628 Y-bearing cells, respectively (ordered according to increasing P -values) (see also lists in Supplementary Table 11). Genes marked with
 629 an asterisk are long non-coding RNAs. Genes in red are located on autosomes, those in black on sex chromosomes. “XLOC” denotes
 loci annotated in our previous work⁸; notably, only few XLOC loci (opossum: XLOC-076551, XLOC-076500, XLOC-076152, XLOC-
 076546) were found to have coding potential (as assessed by analyses based on ribosome profiling data generated in our previous
 work⁸), whereas all others likely represent lncRNAs.

We summarize the aforementioned new results in the section “Sex chromosomes” and – due to space constraints – in
 the Methods section (i.e., the observations regarding lncRNAs):

(P13, L448-453) “A differential expression analysis between X and Y spermatids identified, as expected, a majority of
 sex-chromosome genes, including gametologs (i.e., genes with homologous counterparts on X and Y chromosomes),
 such as the translational regulatory genes *DDX3X/DDX3Y* (Extended Data Fig. 15; Supplementary Table 11). However,
 we also found some autosomal genes, especially in some species (e.g., human), which might reflect trans-regulatory
 effects associated with the X and Y chromosomes (Extended Data Fig. 15; Supplementary Table 11; Methods).”

(P25, L969-980) “We note that a number of the differentially expressed genes, including the most significant cases in
 human (Extended Data Fig. 15a), are putatively noncoding, which is noteworthy, because long noncoding RNAs
 (lncRNAs) are typically nuclear⁹⁶ and hence their differential expression levels are unlikely to be offset by transcript
 exchange between spermatid cells via cytoplasmic bridges. The three most Y-spermatid-specific transcripts are lncRNAs
 emanating from homologous low copy repeats (LCRs; i.e., segmental duplications) on chromosomes 13 (*FAM230C*), 21
 (*XLOC-095504*), and 22 (*FAM230F*) that cause genomic disorders by triggering nonallelic homologous recombination
 (NAHR) events. They include the *FAM230F* lncRNA in the q11.2 LCR region on chromosome 22 (22q11.2) that is
 particularly susceptible to NAHR-generated deletions that lead to various congenital malformation disorders, including
 the DiGeorge syndrome, the most frequent microdeletion disorder⁹⁷.”

We would finally like to note that our data was not designed for such a differential X/Y spermatid analysis and is
 therefore underpowered (especially visible for the mouse, where the separation of X and Y spermatids was also more
 difficult). That is, while our work provides a proof of concept and initial interesting insights, future studies with more
 biological replicates for a given species are warranted to exhaustively trace transcriptomal differences between the two
 types of spermatids.

Next, we assessed the possibility of distinguishing X and Y bearing spermatids from single-cell data using our differential
 transcript procedure, also with the aim to illuminate the extent of transcript exchange between the two spermatid types
 via cytoplasmic bridges. We applied our approach to two testicular single-cell RNA-seq data sets for mouse (Ernst et al.
 *Nat. Commun.* 2019; PMID: 30890697) and human (Shami et al. *Dev. Cell* 2020; PMID: 32504559). Contrary to the two
 distinct spermatid cell populations detected in our human and mouse snRNA-seq data (i.e., the “L-shape” to the left in
 plots in the new Extended Data Fig. 14 a and b), cells did not separate into two populations in the single-cell data sets
 (new Extended Data Fig. 14 a/b, to the right). Thus, while transcript exchange through cytoplasmic bridges may not
 always be complete (Bhutani et al. *Science* 2021; PMID: 33446482), contrary to what was originally suggested (Braun et
 al. *Nature* 1989; PMID: 2911388), our observation is consistent with substantial transcript exchange across X and Y
 spermatids via cytoplasmic bridges (i.e., the transcript content between cells is overall rather similar). Our analysis thus
 also confirms that single-cell data is not suited to distinguish between X/Y spermatids.

**Extended Data Fig. 14 | Classification of X and Y bearing spermatids.** The scatter and bar plots show the percentages of X and Y
 transcripts, and the distribution of X transcripts (%), respectively, across nuclei or cells in our single-nucleus (left) and publicly
 available single-cell data in mouse⁶⁰ (a) and human²⁷ (b). The red and green lines depict the fitted curves for the bimodal distributions.

We summarize this new analysis in the **Sex chromosomes** section:

(P13, L437-446) “We next sought to separate X- and Y-bearing spermatids to investigate their distinct transcriptomal
 properties during spermiogenesis. Such an analysis is likely not possible using single-whole-cell transcriptomic data
 because X and Y spermatids remain connected by cytoplasmic bridges and hence are thought to contain similar
 cytoplasmic transcript pools^{59,60}. However, our single-nucleus data should afford the separation of X/Y spermatids.
 Indeed, based on differential X/Y transcript contents across spermatids (Methods), we were able to separate spermatids
 into distinct X and Y lineages across mammals (Fig. 4c; Supplementary Table 3). As expected, our approach failed to
 separate X/Y spermatids in available human²⁷ and mouse⁶¹ scRNA-seq data sets (Extended Data Fig. 14), supporting the
 notion of substantial transcript exchange across X and Y spermatids cells via cytoplasmic bridges⁶², although this
 equilibration may not always be complete⁵⁹.”

Regarding the reviewer’s final question: the fact that a small fraction of X-bearing spermatids seem to carry Y genes
 transcripts in their nuclei is likely explained by a carry-over of Y nuclear transcripts from preceding (meiotic) stages,
 and/or cytoplasmic carry-over from preceding stages or neighboring Y spermatids through cytoplasmic bridges that get
 stuck to nuclear membranes during the nucleus extraction/snRNA-seq procedure.

**Minor comments**

- **I. 122: Misleading phrasing: the authors refer to 34 libraries, but it actually accounts for 27 single-nuclei libraries
 and 7 bulk libraries only used for genome reannotation, which should be clarified.**

Thanks for spotting this imprecision – we now clarified this part so that it reads:

(P03, L121-125) “The dataset consists of 27 libraries, with one to three biological replicates per species and a median of
 ~275 million snRNA-seq reads per library (Supplementary Table 1). We refined and extended existing Ensembl²⁸ genome
 annotations across all species based on bulk-testis RNA-seq data (7 libraries) (Supplementary Tables 1, 2; Methods), to
 ensure optimal read-mapping and prevent biases in downstream cross-species analyses.”

- **I.179: which "other somatic" cell types are these? This should at least be clarified in the legend of Supp Fig 2.**

Thanks for pointing out that this was not clear. We had defined other somatic cell types based on marker genes, as
 detailed in the legend of Extended Fig. 2 (P31, L1227-1231) and Methods (P20, L746-751). We now clarify which cell
 types “Other somatic” refers to in the legend of Extended Data Fig. 3, as suggested by the reviewer:

(P33, L1234-1237) “**Extended Data Fig. 3: Mammalian testicular gene expression divergence. a,** Gene expression
 phylogenies. Neighbor-joining trees based on pairwise expression level distances (1–rho, Spearman’s correlation
 coefficient) for the main testicular cell types, respectively (“Other somatic” refers to peritubular, smooth muscle,
 endothelial, macrophage, and Leydig cells as detailed in the legend of Extended Data Fig. 2 and Methods).”

- **for simplicity, the orange box in the legend of Fig 1c could be removed since it is not used in the plot**

We have modified the Fig. 1c as suggested.

- **I.532: names and authorization numbers of human tissue biobanks and companies should probably be provided (to
 be confirmed by the editor)**

The human samples underlying the snRNA-seq data stem from the the official NIH tissue bank in Maryland
 (<http://medschool.umaryland.edu/btbank/>) and Tissue Solutions (<https://www.tissue-solutions.com/>), respectively,
 which both strictly adhere to all relevant ethical legislations. The samples used for the RNA *in situ* hybridization
 experiments were obtained from the tissue biobank at the Department of Growth and Reproduction (Rigshospitalet,

Copenhagen, Denmark) containing orchiectomy specimens from individuals with testicular cancer (The Danish Data
Protection Agency, permit number J.nr. 2001-54-0906). All patients have given informed consent for donating the
residual tissues for research. We now added this information to the ethics section statement of the methods section, in
which we had already previously noted all relevant authorization for using the human samples from these sources:

(P16, L575-589) “Our study complies with all relevant ethical regulations with respect to both human and other species’
samples. Human samples underlying the snRNA-seq data were obtained from scientific tissue banks
(<http://medschool.umaryland.edu/btbank/>) or dedicated companies (<https://www.tissue-solutions.com/>); informed
consent was obtained by these sources from donors before death or from next of kin. The samples used for the RNA *in*
*situ* hybridization experiments were obtained from the tissue biobank at the Department of Growth and Reproduction
(Rigshospitalet, Copenhagen, Denmark) containing orchiectomy specimens from individuals with testicular cancer (The
Danish Data Protection Agency, permit number J.nr. 2001-54-0906). All patients have given informed consent for
donating the residual tissues for research. The use of all human samples for the type of work described in this study was
approved by an ethics screening panel from the European Research Council (ERC) (associated with H.K.’s ERC
consolidator grant 615253, OntoTransEvol) and local ethics committees: from the Cantonal Ethics Commission in
Lausanne (authorization 504/12); from the Ethics Commission of the Medical Faculty of Heidelberg University
(authorization S-220/2017); and from the regional medical research ethics committee of the capital region of
Copenhagen (H-16019637).”

- for clarity, the reason for the difference in tissue processing protocols (fixed/unfixed) used for chicken and platypus
should be explicitly laid out (I suspect it is due to a change in chemistry in the 10x Genomics Chromium kits used?)

The reason is that the protocol without fixation resulted in data of high quality for platypus and chicken, whereas the
fixation protocol failed to yield data of acceptable quality these species.

We now added this explanation to the Methods:

(P17, L615-617) “For platypus and chicken, a similar preparation method was used, but the nuclei were not fixed, given
that this protocol gave optimal results for these species (the fixation protocol failed to yield data of adequate quality).”

The difference in protocols is unrelated to the change in 10x chemistry – the reason for the 10x chemistry change is that
the data for platypus and chicken were generated last in the project (i.e., when the new chemistry became available).

**Referee #2 (Remarks to the Author):**

**This study by Murat and colleagues analyzes 90K single nuclei isolated from the testis using the 10X genomics**
**platform. The 90K cells are comprised of eleven species from multiple mammalian lineages including monotremes,**
**eutherian and marsupial, with chicken being an outgroup. The data generated confirms many early observations**
**made in the field (e.g rapid evolution of spermiogenesis genes, enrichment of spermatogenesis genes on x**
**chromosome, enrichment of infertility genes in early stages, identification of conserved and diverged changes in**
**transcriptome). However, these datasets have enabled a finer assessment of conserved and diverged changes in the**
**gametogenesis program over a shorter evolutionary timescale. Moreover, these datasets revealed the existence of**
**meiotic sex chromosome silencing in monotremes, suggesting that MSCI program predates the split of monotremes**
**and therian mammals. Finally, this analysis allowed the distinct separation of X and Y bearing spermatids, but this**
**analysis fall short of revealing any interesting biological findings to cells with different chromosomal complement.**
**Taken together, the data is of high quality and would be great utility to the community, but the novelty of findings**
**may be rather limited in its current form.**

We thank the referee for the appreciation of our work and the very useful comments that led to new results and have
substantially improved the manuscript, as detailed below. We believe that our work, especially in conjunction with
insights obtained during the revision, substantially advance our knowledge of the evolution of spermatogenesis, as

summarized in the general section in the beginning of this response letter. We sought to better highlight the novelty of
our various results and conclusions.

**1) Given the large number of species analyzed, a greater dive in the data is needed to demonstrate functional**
**consequences of transcriptomic differences between species. Given the limitation with using primates– identifying a**
**novel rodent specific program and demonstrating functionality may be important to demonstrate impact or**
**importance of programmatic differences.**

We appreciate the reviewer’s comment, which is shared by referee #1. We have substantially expanded on our previous
functional investigations of genes with conserved or changed trajectories, which provided exciting new insights, as
detailed below.

We previously supported our original hypothesis that genes with highly conserved trajectories across mammals are
crucial for spermatogenesis by showing that genes leading to infertility when knocked out in the mouse are more
conserved across species than genes where fertility remains unaffected (Extended Data Fig. 8b). To strengthen this
hypothesis and globally explore functional roles of highly conserved spermatogenic genes, we screened for enriched
processes based on Gene Ontology (GO) annotations, categorizing genes according to the cell type in which their
expression peaks. This analysis revealed that enriched GO terms indicate an involvement of genes with conserved
trajectories in fundamental spermatogenic processes that are overall typical of the cell type in which they show peak
expression (new Extended Data Fig. 9a, b). For example, conserved genes with peak expression in spermatocytes are
enriched for GO terms related to meiotic functions, including stage-specific ones (e.g., in meiosis I, which starts in
leptotene spermatocytes).

By contrast, a GO analysis for genes with trajectory changes in primates and amniotes, respectively, revealed
enrichments of broader molecular (i.e., non-spermatogenesis-specific) processes, especially those related to metabolic
processes (Extended Data Fig. 9c, d).

To further explore the functional differences of genes with conserved and changed trajectories, we assessed their
overall spatial and developmental (i.e., temporal) expression patterns using our reference dataset for 7 organs (Cardoso-
Moreira et al. *Nature* 2019; PMID: 31243369). This analysis uncovered that genes with changed expression are
significantly more tissue- and time-specific than genes with conserved expression (Extended Data Fig. 9e). This suggests
that the reduced pleiotropy of the former genes – and hence the reduced purifying selection acting on them – facilitated
their expression change during evolution.

Extended Data Fig. 9 | Functions of conserved and changed trajectories. **a**, Top 20 enriched biological process GO terms of genes showing conserved expression trajectories across primates and peak expression in the different cell types, respectively. **b**, Top 20 enriched biological process GO terms of genes showing conserved trajectories between human and mouse and peak expression in the different cell types, respectively. **c**, The top 20 enriched biological process GO terms of genes with trajectory changes in primates. **d**, The top 20 enriched biological process GO terms in genes showing trajectory changes in amniotes. **e**, Human tissue (left) and time (right) specificity for changed and conserved trajectories across amniotes, respectively (two-sided Wilcoxon rank-sum tests were performed for statistical comparisons).

We now also provide a complete list of genes with changed trajectories in amniotes that are essential for spermatogenesis/fertility in mouse, as assessed by mouse KO studies (IMPC database), in the new **Supplementary Table 8**. This list includes striking mouse/rodent-specific trajectory changes that presumably led to new essential functions of these genes in mouse spermatogenesis. We now discuss one such case (P10, L347-349, see also revised paragraph further below), that of *SPIDR*, whose expression switched from an ancestral early/spermatogonial to late spermatid expression during evolution (**Supplementary Table 8**). Knockout of this gene leads to reduced testis size and a paucity of post-meiotic cells (Prakash et al. *Nat Commun.* 2021; PMID: 34253720).

Finally, we now also explore more directly the contribution of new genes to the evolution of new spermatogenic functions (on the rodent lineage leading to mouse) by combining the fertility information from the IMPC database with

829 gene age information. This revealed the emergence of key spermatogenic genes at different time points during
 evolution (Extended Data Fig. 10a; Supplementary Table 8). The two identified rodent-specific genes – *D1Pas1* and
 *H2al2a* – represent interesting cases. They are both intronless retrogenes that originated via an RNA-based duplication
 mechanisms from parental genes on chromosome X, consistent with the view that the X has spawned a
 disproportionately large number of new retrogenes with important spermatogenesis functions during mammalian
 evolution (Necsulea and Kaessmann *Nat. Rev. Genet.* 2014; PMID: 25297727) (Extended Data Fig. 10c). *D1Pas1* encodes
 a DEAD-box RNA helicase essential for meiosis (Inoue et al. *Biochem. Biophys. Res. Commun.* 2026; PMID: 27473657),
 while *H2al2a* encodes a histone variant required for genome compaction around protamines (Barral et al. *Mol. Cell*
 2017; PMID: 28366643). Both genes show strongly increasing expression levels in late spermatogenesis (Extended Data
 Fig. 10b) and are thus in agreement with our notion that new duplicate genes contribute functional roles predominantly
 in later stages of spermatogenesis (Fig. 2g; Extended Data Fig. 3d; P10, L351-360). Our data also revealed various other
 interesting genes that evolved key spermatogenic functions during evolution; for example, the duplicate gene *Spink2*
 (encoding a serine protease inhibitor) emerged in the common eutherian ancestor and evolved an essential role in
 spermiogenesis (acrosome formation) (Kherraf et al. *EMBO Mol. Med.* 2017; PMID: 2854943).

 **Extended Data Fig. 10 | New genes with crucial spermatogenic functions.** **a**, percentages of genes among new genes of different
 ages (i.e., emergence in last common vertebrate, tetrapod, mammalian, eutherian, or rodent ancestors) that show infertility
 phenotypes when knocked out in the mouse (numbers of genes with infertility phenotypes and total number of genes considered
 for an age category are provided above each bar). **b**, Expression levels of the retrogenes *H2al2a* and *D1Pas1* in spermatogenic and
 somatic cell types. Error bars correspond to the range between minimum and maximum expression values across biological
 replicates. **c**, Trees of *H2al2a* and *D1Pas1* and paralogs from which they originated through the mechanism of RNA-based gene
 duplication (i.e., *D1Pas1* stems from *Ddx3y*, whereas the precise ancestral parental gene of *H2al2a* is not easily discernable due to a
 complex history of a number of RNA- and DNA-based duplication events).

 We now present all of the aforementioned new analyses and results in the section now termed “Gene expression
 conservation and innovation” to accommodate all previous and new results (also the new cell-cell communication
 analysis – see response to next comment below):

(P10, L330-376) “In agreement with these highly conserved sets of genes playing key roles in mammalian/amniote
 spermatogenesis, our analyses of fertility phenotypes³¹ unveiled that genes involved in fertility are significantly more
 conserved in their expression trajectories than genes not associated with fertility (Extended Data Fig. 8b). Consistently,
 a Gene Ontology⁴² enrichment analysis indicates an involvement of genes with conserved trajectories in fundamental
 spermatogenic processes that are typical of the cell type in which they show peak expression (Extended Data Fig. 9a,
 b). Thus, genes with conserved trajectories for which spermatogenesis functions remain uncharacterized represent a
 promising resource of candidates for the exploration of fertility phenotypes (Supplementary Tables 5, 7, 8). Notably,
 genes with lineage-specific trajectory changes are enriched with broader, typically metabolic, processes (Extended Data
 Fig. 9c, d). They are also significantly more tissue- and time-specific than genes with conserved expression (Extended
 Data Fig. 9e), which may have facilitated their expression change during evolution because of reduced pleiotropic
 constraints. However, genes with changed trajectories nevertheless include many genes for which key fertility functions
 have been described (Supplementary Tables 7, 8). For example, *IP6K1*, which elicits infertility when knocked out in the
 mouse⁴³, shows strongly increasing expression towards the end of spermiogenesis in all amniotes except the primates,

where expression is high in SG and then overall declines (Fig. 3b; Extended Data Fig. 8c). Thus, the primary function of
 *IP6K1* may have shifted from late to early spermatogenesis during primate evolution. An example of an essential
 spermatogenic gene with a rodent-specific change is *SPIDR*⁴⁴, whose expression switched from an ancestral
 early/spermatogonial to late spermatid expression during evolution (Supplementary Tables 7, 8).

 In conjunction with mouse fertility data, we used our data to explore the contribution of new genes (mostly arising from
 gene duplications) to the evolution of new spermatogenic functions, focusing on the rodent lineage leading to mouse.
 This analysis revealed the emergence of key spermatogenic genes at different time points during evolution (Extended
 Data Fig. 10a; Supplementary Table 8), such as two rodent-specific retrogenes – *D1Pas1* and *H2al2a* – that originated
 through RNA-based duplication from parental genes on the X chromosome (Extended Data Fig. 10c). *D1Pas1* is essential
 for meiosis⁴⁵, while *H2al2a* is essential for genome compaction in late spermatogenesis⁴⁶. Both genes show strongly
 increasing expression levels in late spermatogenesis (Extended Data Fig. 10b) and are thus in agreement with new genes
 contributing to functional roles predominantly during late spermatogenesis (see above; Fig. 2g; Extended Data Fig. 4d).
 Other examples include the paralog *Spink2* (Supplementary Table 8), which emerged in the common eutherian ancestor
 and also evolved an essential role in late spermatogenesis⁴⁷.”

 Overall, we have substantially expanded on the functional characterization of genes with conserved and changed
 trajectories and also added insights pertaining to the contribution of new genes to the functional evolution of
 spermatogenesis. Together, our data and results provide a major resource for the further exploration and experimental
 scrutiny of transcriptome innovation in the testis.

 We have also modified the relevant part in the discussion to reflect the new results. We now explicitly state that we did
 not directly (experimentally) demonstrate the contribution of individual trajectory changes to the emergence of species-
 895 /lineage-specific phenotypes. We share with the reviewer the desire for this functional work, but it is beyond the scope
 of this specific study.

 (P14, L497-502) “Our cross-species comparisons of individual genes revealed temporal expression differences across
 species, including human-specific changes, which were probably facilitated by reduced pleiotropic constraints. Our
 results thus provide an extensive list of candidates whose contributions to the evolution of species-specific
 spermatogenesis phenotypes can be experimentally scrutinized. We also uncovered conserved expression programs
 underlying spermatogenic processes ancestral to individual mammalian lineages and mammals as a whole.”

 **2) The snRNAseq allows the identification of Sertoli cells which have been missed by earlier scRNAseq datasets of
 human and macaque because of the difficulty of sorting intact cells. Here, the authors have an opportunity to glean
 insights about transcriptomic and signaling differences of Sertoli cells, which may reveal novel insights in the
 regulation of the gametogenesis program.**

This is a very good suggestion. We decided to expand our work on Sertoli cells by investigating ligand-receptor
 interactions underlying the communication between Sertoli cells and germ cells across species using the *CellPhoneDB*
 framework (Efremova et al. *Nat Protoc.* 2020; PMID: 32103204) (Methods: P23, L918-928). Our cross-species
 comparisons revealed various conserved known and novel ligand-receptor interactions (Extended Data Fig. 11a, b and
 Supplementary Table 9). Given potential false positive (and negative) predictions of CellPhoneDB and similar
 approaches (Dimitrov et al. *Nat Commun.* 2022; PMID: 35680885), we consider interactions that are predicted for
 multiple species and likely reflect evolutionary conservation (Extended Data Fig. 11; Supplementary Table 9) to be more
 reliable than species-specific predictions, as also noted in the Method section (P24, L928-932). We therefore focus our
 presentation on these cases.

 Our analyses suggest, for example, that Sertoli cells communicate with spermatogonia, spermatocytes and round
 spermatids in all amniotes through interactions of *CADM1* between the cell types pairs, or between *CADM1* (Sertoli
 cells) and *NECTIN3* (spermatocytes/round spermatids). These interactions thus likely emerged as a common mechanism
 already in the amniote ancestor. It is noteworthy that while it was known that *CADM1* is crucial for spermatogenesis
 (e.g., Yamada et al. *Mol. Cell Bio.* 2006; PMID: 16612000), previous work suggested that Sertoli cells do not express this
 gene (Wakayama and Iseki *Anat. Sci. Int.* 2009; PMID: 19337787), whereas we find it to be highly expressed in Sertoli
 cells across species, in agreement with human protein atlas data ([https://www.proteinatlas.org/ENSG00000182985-
 CADM1/tissue/testis#img](https://www.proteinatlas.org/ENSG00000182985-CADM1/tissue/testis#img)); thus, *CADM1* is likely a key mediator of the aforementioned cell-cell communications.
 Another interesting example is the inferred *NECTIN2*-*NECTIN3*-mediated communication of Sertoli cells with
 round/elongated spermatids in both human and mouse. While this interaction was known in mouse (Ozaki-Kuroda et

al. *Curr. Biol.* 2002; PMID: 12121624), previous human data was not conclusive (Bronson et al. *J. Assist. Reprod. Genet.*
 2017; PMID: 28689229). Overall, our data and results represent a major resource for exploring cellular communications
 in the testis and its molecular basis across amniotes.

 We now describe these new analyses and results in the section “Gene expression conservation and innovation”.

 (P10, L364-376) “Finally, we used our data to investigate ligand-receptor interactions underlying the communication
 between Sertoli cells, which have a central (paracrine) role in supporting and controlling spermatogenesis⁴⁸, and germ
 cells across species using CellPhoneDB⁴⁹. Our cross-species comparisons revealed various conserved known and novel
 ligand-receptor interactions (Extended Data Fig. 11a, b and Supplementary Table 9). For example, our data provide
 evidence that communication of Sertoli cells with SG, SC, and rSD occurs in all amniotes through interactions of the cell
 adhesion molecule CADM1⁵⁰ or between CADM1 (Sertoli cells) and NECTIN3 (SC/rSD) (Supplementary Table 9). We note
 that CADM1 was previously thought to not to be expressed in Sertoli cells⁵¹, but our data reveals high expression in this
 cell type across amniotes, in agreement with human protein atlas data (proteinatlas.org/ENSG00000182985-
 CADM1/tissue/testis#). Our work also supports the notion⁵² that the NECTIN2-NECTIN3 complex mediates the
 communication of Sertoli cells with spermatids not only in mouse⁵³ but also in humans (Supplementary Table 9).”

 **Extended Data Fig. 11 | Sertoli-germ cell communications mediated by ligand-receptor interactions in testis across mammals. a,**
 **Significant ligand-receptor interactions (as assessed by CellPhoneDB⁴⁹; see Methods) across species for Sertoli-spermatogonia,**
 **Sertoli-spermatocytes, Sertoli-round spermatids and Sertoli-elongated Spermatids communications (details in Supplementary Table**
 **9). b, Overview of all distinct significant ligand-receptor interactions between Sertoli and the four principal germ cell types across**
 **species. The number of detected significant interactions for each species in both panels is indicated to the bottom left of each plot**
 **(Set size).**

**3) The ability to distinguish X and Y bearing spermatids is exciting. Are there greater programmatic differences**
 **between these spermatids other than the X and y transcripts. Comparing pseudo bulk transcriptomes of these**
 **separate cells will be interesting since many chromatin modifiers are detected on either X or Y – which can lead to**
 **more global differences in transcriptome.**

 We are grateful for the enthusiasm regarding this section and we have extended the X/Y spermatid analyses, as detailed
 below. We originally sought to exploit the separation of X-and Y-bearing spermatids to separately and thus cleanly
 assess expression of X- and Y-linked genes across spermatogenesis, respectively, which allowed us to detect MSCI across
 mammals at optimal resolution – please note in this context the new plots in Fig.4d, which were made in response to a

comment from referee #3 and now even better highlights patterns of MSCI, especially the novel one detected in
 platypus, by showing directly all individual cells using an even better measure (X/A or Y/A transcript abundance ratio
 instead of percentage of transcripts from X or Y).

 **Figure 4d | Mammalian sex chromosome evolution.**

We agree with the reviewer that assessing global transcriptional differences between the two types of spermatids is
 also intriguing.

 We thus proceeded with a differential expression analysis for X- vs. Y-bearing spermatids, which revealed, as expected,
 a majority of sex-chromosome genes, including interesting gametologs (i.e., genes with homologous counterparts on X
 and Y chromosomes), such as the translational regulators DDX3X/DDX3Y (new Extended Data Fig. 15; Supplementary
 Table 11). Interestingly, however, we also find many autosomal genes to be differentially expressed, especially in human
 and opossum Y spermatids (for mouse, we generally detected few expression-biased genes and in platypus all biased
 genes but one are X-/Y-linked), which might reflect trans regulatory effects associated with the sex chromosomes. The
 vast majority of cases are specific to the analyzed species, suggesting a rapid evolutionary turnover of spermatid
 differential expression during mammalian evolution. Notably, many of the differentially expressed genes, including the
 most significant cases, are putatively noncoding, which is noteworthy, because long noncoding RNAs (lncRNAs) are
 usually localized in the nucleus (Derrien et al. *Genome Res.* 2012; PMID: 22955988) and hence their differential
 expression levels are unlikely to be offset by transcript exchange between spermatid cells via cytoplasmic bridges. The
 three most Y-spermatid-specific transcripts are lncRNAs emanating from homologous low copy repeats (LCRs; i.e.,
 segmental duplications) on chromosomes 13 (FAM230C), 21 (XLOC-095504), and 22 (FAM230F) that cause genomic
 disorders by triggering nonallelic homologous recombination (NAHR) events. They include the FAM230F lncRNA in the
 q11.2 LCR region on chromosome 22 (22q11.2) that is particularly susceptible to NAHR-generated deletions that lead
 to various congenital malformation disorders, including the DiGeorge syndrome, the most frequent microdeletion
 disorder⁹⁷.

referee #2

**Extended Data Fig. 15 | Transcriptome differences between X and Y bearing spermatids.** The heatmaps indicate transcript
abundances of significantly enriched genes in X- and Y-bearing spermatids in the different species (a: human, b: mouse, c: opossum,
995 d: platypus). The corresponding bar plots (genes in the same order) show the genes with significantly enriched expression in X- and

996 Y-bearing cells, respectively (ordered according to increasing P -values) (see also lists in Supplementary Table 11). Genes marked with
 997 an asterisk are long non-coding RNAs. Genes in red are located on autosomes, those in black on sex chromosomes. “XLOC” denotes
 loci annotated in our previous work⁸; notably, only few XLOC loci (opossum: XLOC-076551, XLOC-076500, XLOC-076152, XLOC-
 076546) were found to have coding potential (as assessed by analyses based on ribosome profiling data generated in our previous
 work⁸), whereas all others likely represent lncRNAs.

 We summarize the aforementioned new results in the section “Sex chromosomes” and – due to space constraints – in
 the Methods section (i.e., the observations regarding lncRNAs):

 (P13, L448-453) “A differential expression analysis between X and Y spermatids identified, as expected, a majority of
 sex-chromosome genes, including gametologs (i.e., genes with homologous counterparts on X and Y chromosomes),
 such as the translational regulatory genes *DDX3X/DDX3Y* (Extended Data Fig. 15; Supplementary Table 11). However,
 we also found some autosomal genes, especially in some species (e.g., human), which might reflect trans-regulatory
 effects associated with the X and Y chromosomes (Extended Data Fig. 15; Supplementary Table 11; Methods).”

 (P25, L969-980) “We note that a number of the differentially expressed genes, including the most significant cases in
 human (Extended Data Fig. 15a), are putatively noncoding, which is noteworthy, because long noncoding RNAs
 (lncRNAs) are typically nuclear⁹⁶ and hence their differential expression levels are unlikely to be offset by transcript
 exchange between spermatid cells via cytoplasmic bridges. The three most Y-spermatid-specific transcripts are lncRNAs
 emanating from homologous low copy repeats (LCRs; i.e., segmental duplications) on chromosomes 13 (*FAM230C*), 21
 (*XLOC-095504*), and 22 (*FAM230F*) that cause genomic disorders by triggering nonallelic homologous recombination
 (NAHR) events. They include the *FAM230F* lncRNA in the q11.2 LCR region on chromosome 22 (22q11.2) that is
 particularly susceptible to NAHR-generated deletions that lead to various congenital malformation disorders, including
 the DiGeorge syndrome, the most frequent microdeletion disorder⁹⁷.”

 We would finally like to note that our data was not designed for such a differential X/Y spermatid analysis and is
 therefore underpowered (especially visible for the mouse, where the separation of X and Y spermatids was also more
 difficult). That is, while our work provides a proof of concept and initial interesting insights, future studies with more
 biological replicates for a given species are warranted to exhaustively trace transcriptomal differences between the two
 types of spermatids.

 Next, we assessed the possibility of distinguishing X and Y bearing spermatids from single-cell data using our differential
 transcript procedure, also with the aim to illuminate the extent of transcript exchange between the two spermatid types
 via cytoplasmic bridges. We applied our approach to two testicular single-cell RNA-seq data sets for mouse (Ernst et al.
 *Nat. Commun.* 2019; PMID: 30890697) and human (Shami et al. *Dev. Cell* 2020; PMID: 32504559). Contrary to the two
 distinct spermatid cell populations detected in our human and mouse snRNA-seq data (i.e., the “L-shape” to the left in
 plots in the new Extended Data Fig. 14 a and b), cells did not separate into two populations in the single-cell data sets
 (new Extended Data Fig. 14 a/b, to the right). Thus, while transcript exchange through cytoplasmic bridges may not
 always be complete (Bhutani et al. *Science* 2021; PMID: 33446482), contrary to what was originally suggested (Braun et
 al. *Nature* 1989; PMID: 2911388), our observation is consistent with substantial transcript exchange across X and Y
 spermatids via cytoplasmic bridges (i.e., the transcript content between cells is overall rather similar). Our analysis thus
 also confirms that single-cell data is not suited to distinguish between X/Y spermatids.

**Extended Data Fig. 14 | Classification of X and Y bearing spermatids.** The scatter and bar plots show the percentages of X and Y
transcripts, and the distribution of X transcripts (%), respectively, across nuclei or cells in our single-nucleus (left) and publicly
available single-cell data in mouse⁶⁰ (a) and human²⁷ (b). The red and green lines depict the fitted curves for the bimodal distributions.

We summarize this new analysis in the Sex chromosomes section:

(P13, L437-446) “We next sought to separate X- and Y-bearing spermatids to investigate their distinct transcriptomal
properties during spermiogenesis. Such an analysis is likely not possible using single-whole-cell transcriptomic data
because X and Y spermatids remain connected by cytoplasmic bridges and hence are thought to contain similar
cytoplasmic transcript pools^{59,60}. However, our single-nucleus data should afford the separation of X/Y spermatids.
Indeed, based on differential X/Y transcript contents across spermatids (Methods), we were able to separate spermatids
into distinct X and Y lineages across mammals (Fig. 4c; Supplementary Table 3). As expected, our approach failed to
separate X/Y spermatids in available human²⁷ and mouse⁶¹ scRNA-seq data sets (Extended Data Fig. 14), supporting the
notion of substantial transcript exchange across X and Y spermatids cells via cytoplasmic bridges⁶², although this
equilibration may not always be complete⁵⁹.”

**4) In extended figure 9 – it's curious that Gibbons are the only species where spermatogonia UMI counts are**
**comparable to pachytene cells, is there anything special about Gibbon's? Furthermore, in opossum and chicken**
**zygotene stages appear to be expanded. Is this a color scheme or cell classification issue?**

Gibbon (and also platypus) might indeed have an overall elevated transcript diversity in undifferentiated
spermatogonia, although we note that the larger apparent similarity between undifferentiated spermatogonia and
pachytene spermatocytes (i.e., overlapping distributions) in this species is also explained by the high variation in
transcript diversity across individual cells in these cell types and that the transcript diversity distributions along
spermatogenesis overall vary to different extents between species due to some biological and/or technical variation
(the data are now presented in the new Extended Data Fig. 1b).

We apologize that our color scheme was confusing. The color of the box plots of zygotene spermatocytes in primates
(where we were able to identify different spermatocyte subtypes thanks to the cross-species data integration)
corresponds to spermatocytes in general in the non-primate species. To resolve this issue and for other reasons
(comments from referee #3), we now do not bin anymore (i.e., we display single boxes per cell type) and also better
place/separate the legends for the two groups of species. See the new Extended Data Fig. 1b below:

**Extended Data Fig. 1 | Data overview and quality controls across cell types and species. a**, Numbers of nuclei per cell type. **b**,
 Numbers of UMI per nucleus. **c**, Numbers of genes per nucleus. **d**, Numbers of protein-coding genes per cell type. **e**, Percentages of
 mitochondrial UMIs per nucleus. **f**, Cumulative sum of reads per droplet (top) and fraction of intronic reads per droplet (bottom) for
 one human individual (human.2.2); droplets are ranked by the number of reads (decreasing order).

**5) MSCI appears to be conserved feature of all mammalian species. Is MSCI as tightly regulated in platypus as it is in**
 **mammals? How many genes escape silencing and how does this compare to other species.**

Many thanks for raising the interesting idea of assessing the completeness of MSCI in platypus and compare it to that
 of the other mammals, including opossum, for which the completeness of MSCI has not been systematically explored.

We set out to assess the completeness of MSCI across species by modifying a previously developed approach (Shami et
 al. *Dev. Cell* 2020; PMID: 32504559), in which transcript abundances of genes before and during MSCI are compared.
 Genes with an increased abundance in cells around the pachytene stage compared to the preceding spermatogonial
 stage are inferred to reflect active transcription during MSCI, as opposed to potential carry-over of stable transcripts,
 which may also occur. Specifically, we modified the previous approach by Shami et al. by assessing the statistical
 significance of gene expression increases at spermatocyte stages subject to MSCI (relative to the spermatogonia stage)
 using a differential expression analysis implemented in Seurat (Methods: P25, L983-994).

Our analyses based on this procedure revealed no human and only one potential mouse MSCI escapee (new **Extended**
 **Data Fig. 17**), in agreement with the notion that MSCI is complete in these species (Sin and Namekawa *Epigenetics* 2013;
 PMID: 23880818). For the opossum, we also only detected one potential candidate (**Extended Data Fig. 17**), which
 suggests that MSCI is as complete in marsupials as in eutherians, in agreement with previous work based on an
 experimental analysis of ten genes (Mahadevaiah et al. *Curr. Biol.* 2009; PMID: 19716301). Finally, and notably, for the
 platypus sexually differentiated regions (SDR), we also only identify one potential escapee (**Extended Data Fig. 17**). This
 observation nicely contrasts with that observed for the pseudoautosomal regions (PARs), where many genes show
 predominant expression in cells subject to MSCI (Fig. 4d) – a further illustration of the strong SDR-specific MSCI pattern
 in this species. Overall, our findings imply that MSCI is as complete in monotremes as in therian mammals and that MSCI
 (and hence MSUC) was already highly efficient in the common mammalian ancestor.

**Extended Data Fig. 17. Assessment of the completeness of MSCI across species.** To identify potential MSCI escapees, we screened
 for X-linked genes in SDRs with a significant increase in transcript abundance from SG to SC stages subject to MSCI, which ensures
 that potential escape genes are indeed actively transcribed in SC (i.e., they do not merely represent genes expressed in SG with stable
 transcripts still detectable in SC) (Methods). Expression differences (\log_2 -fold change, x-axis) of X and autosomal genes between SC
 and SG were assessed using differential expression analysis (Methods; p -values on the Y-axis are Bonferroni-corrected). Vertical red
 lines indicate a 0.25-fold expression change (\log_2 -scale); horizontal red lines indicate corrected $P < 0.05$. Potential escapees (i.e., X-
 linked genes in SDRs with significantly higher expression in SC than SG) are indicated by red arrows.

We now describe these new results at the end of the section “Sex chromosomes”:

(P14, L469-480) “Indeed, while the joint analysis of all platypus X-linked genes only reveals a small expression dip around
 the pachytene stage (Fig. 4d, upper platypus graph), an analysis only of SDR genes reveals a strong reduction of X
 transcript levels. By contrast, PAR genes show stable expression levels across spermatogenesis (Fig. 4d, lower graph).
 Moreover, the difference in transcript abundances between SDRs and PARs due to MSCI is visible for all five platypus X
 chromosomes (Extended Data Fig. 16b). Notably, our assessment of the completeness of MSCI across species reveals
 that platypus MSCI is as complete as that in other species for which there is little or no MSCI escape^{27,64,65} (i.e., we only
 detected zero or one potential transcriptional escapee in each of the species, Extended Data Fig. 17). The presence of
 MSCI at the SDRs in monotremes is consistent with the partial association of platypus sex chromosomes with
 perinucleolar repressive histone modifications at the pachytene stage⁹. Altogether, our data reveal that efficient MSCI
 is common to all mammalian sex chromosome systems, which implies that the general mechanism of MSUC is an
 ancestral mammalian feature.”

Finally, we note that the few previously reported escape genes in human and mouse reported by Shami et al. are not
 detected as escapees in our analysis and/or are too lowly expressed to be considered (**Reviewer Fig. 1**). The difference
 between Shami et al. and our study may be due to the different types of data (scRNA-seq vs. snRNA-seq) between the
 studies, the fact that Shami et al. did not assess statistical significance, and/or differences in the individual samples
 between the studies. In any event, escape from MSCI is a rare event.

 **Reviewer Fig. 1.** Assessment of the human escapees suggested by Shami et al. 2020 (16 were suggested in that study; 11 are
 sufficiently highly expressed in our data to be assessed).

 **6) Large variations in Extended Figure 4 are noted for each of the clustered patterns expression patterns especially in**
 **primates as compared to non-primates. Can this be a result of Seurat integration? Has the integration of cell across**
 **species checked in an independent manner?**

The variation of the gene expression trajectory clusters displayed in this figure (Extended Data Fig. 5) is typical for this
 type of analysis (see e.g. Extended Data Fig. 7 from Sarropoulos et al. *Nature* 2019; PMID: 31243368). It appears more
 pronounced for the primates than for non-primates because of the larger number of species, orthologs, and (sub)cell
 types in primates. That is, for the primates the analysis is based on trajectories for 7 species, 8,451 expressed 1:1
 orthologs, and 8 (sub)cell types, whereas for the amniote analysis, the analysis is based on 5 species, 4,498 expressed
 1:1 orthologs, and 4 cell types. Importantly, all trajectory clusters are robust and all genes of the same clusters share
 the same predominant expression peak in the same cell type – the essence of this analysis.

 We re-checked the integration of the primate data (done with LIGER) using marker gene expression. In Reviewer Fig. 2,
 we show the expression of key marker genes in the different primates across the cell types annotated based on the
 merged data set. The marker genes show very similar expression trajectories (i.e., predominant expression in the same
 cell types) across the assigned cell types, thus validating the cell type calls made from the merged data.

**Reviewer Figure 2 | Marker gene expression across primates.** The gene expression level is shown for each species across cell types
 that were annotated based on the LIGER-merged dataset.

 **Referee #3 (Remarks to the Author):**

 **Dear authors,**

**I read the manuscript "The molecular evolution of spermatogenesis across mammals" with great interest. The**
 **manuscript is an interesting resource for the community, even though several of the datasets (at least human, mouse,**
 **macaque) have been previously compared and recently published (Niederriter Shami 2020). In that sense, I am not**
 **sure if the current manuscript provides sufficient novelty and advance in knowledge. However, I am aware of the**
 **uniqueness and the novelty of some of the datasets and the computational analysis, although in many aspects**
 **relatively standard, is well performed/presented and some of the results are nicely validated, making the results**
 **robust.**

Thank you for the appreciation of our work and the very useful comments that helped us to substantially improve the
 manuscript. We believe that our work, especially in conjunction with the new results obtained during the revision,
 substantially improves our understanding of the evolution of spermatogenesis. We overall sought to optimize the
 presentation of our various results and conclusions, to better highlight the novelty and impact of our work.

**I have some points that would need clarification:**
 **Figure 2: on the binning, what is the logic to choosing 20 bins? Could you mention the number of cells per bin**
 **per species in a Suppl panel? How did you chose a color of the bin if it included cells from several clusters?**
 **Could you provide the results regarding the pseudotime per species, colored by cluster and provide there**
 **information on the bins there as well. This is an important aspect used through the manuscript and needs to**
 **be clarified/presented.**

We are grateful to the reviewer for making us realize that binning the data adds unnecessary complexity and
 complicates inferences. We originally used bins to obtain a higher temporal resolution. However, we realized

that removing the binning allows for a clearer data representation and avoids the issues highlighted by the reviewer
 (e.g., bins with cells from different clusters), while still providing adequate resolution. In the revised manuscript, instead
 of using bins, we now separate the data by (sub)cell type in the different plots of our paper, including those in Fig. 2.

 **Fig. 2 | Gene expression divergence and evolutionary forces.**

 Moreover, we now also provide the number of cells per cell type in Extended Data Fig. 1a (the new Extended Data Fig.
 1 provides various general measures and quality controls).

**Extended Data Fig. 1 | Data overview and quality controls across cell types and species.** **a**, Numbers of nuclei per cell type. **b**,
 Numbers of UMI per nucleus. **c**, Numbers of genes per nucleus. **d**, Numbers of protein-coding genes per cell type. **e**, Percentages of
 mitochondrial UMIs per nucleus. **f**, Cumulative sum of reads per droplet (top) and fraction of intronic reads per droplet (bottom) for
 one human individual (human.2.2); droplets are ranked by the number of reads (decreasing order).

Finally, we modified **Extended Data Fig. 4** to provide the data presented in **Fig. 2** for all species in the same way as **Fig.**
 **2** (i.e., plotting the data per cell type).

**Extended Data. Fig. 4.**

**Figure 3: how many bins were used to generate these plots? Should you not use the previous binning for consistency**
 **(20 bins)? Perhaps you can consider using the same number of dashed lines across all the plots in all graphs depicting**
 **pseudotime-spermatogenesis for consistency?**

As detailed above, in our revised manuscript we do not bin the data anymore.

For the plots in **Figure 3**, we actually never binned the data – the gene expression trajectories in these plots are based
 on expression values for the different cell types. The high resolution afforded by data integration across primates
 allowed the detection of eight (sub)cell types for the species of this evolutionary lineage (**Fig. 3a**), whereas the trajectory
 comparisons across representative amniotes is based on aligning trajectories for the four principal germ cell types
 identified across all species (**Fig. 3b**); all figures in the paper (e.g., also **Fig. 2**) are based on these cell type classifications.
 Given that the trajectory comparisons are based on these cell types, we prefer to depict the trajectories accordingly in
 this figure.

Thank you for pointing out that we had not consistently put dashed lines (marking the four principal cell
 type/spermatogenic phases) in all plots. We have now added these lines also in the new Fig. 3b and, moreover, added
 labels marking the four principal cell types/spermatogenic phases in the trajectory plots of Fig. 3a-c.

 **Fig. 3 | Evolution of gene expression trajectories along spermatogenesis.**

 **Also for consistency: I would suggest to use one cartoon for human throughout.**

To avoid copyright issues and to facilitate the design, we based the design of the human icons on the appearance of the
 two shared first authors (Florent Murat and Noe Mbengue). However, if this remains a sticking point, we would
 obviously be happy to converge on one common icon across figures in the final paper version.

 **Figure 4: It is not usual to present % X-linked genes, but it would be more informative to have a ratio X/A when
 discussing MSCI. Could the authors also provide the plots for Figure 4 with the X/A ratio in the main figure and place
 the % X-linked in the supplementary data?**

Thank you for this excellent suggestion. We now plot the X/A transcript ratio for each cell ordered across spermatogenic
 pseudotime (Fig. 4d), which (as the reviewer predicted) illustrates even better the MSCI patterns, especially the newly
 detected tight SDR-specific MSCI in platypus. As mentioned above, we now do not use any binning anymore in the
 manuscript, stimulated by the reviewer's earlier comment.

**Fig. 4 | Mammalian sex chromosome evolution.**

**Figure 4: the use of red arrow to mark (X-bearing and Y-bearing) spermatids and in different graphs to mark**
**spermatocytes in pachytene is confusing. I would suggest to use arrows with different colors to mark the different**
**populations of interest.**

This is a good suggestion. We now use black arrows in Fig. 4c and red arrows in Fig. 4d (see pasted Fig. 4 above).

**Line 696-697: TAGLN and ACTA2 peritubular cells and smooth muscle cells**

Indeed, many thanks for spotting this. We changed the text accordingly in the method section and, similarly, in the
legend of Extended Data Fig. 2:

(P20, L746-747) and (P31, L1227) "...TAGLN and ACTA2 peritubular and smooth muscle cells..."

**Ext Data Fig 7c: please use the 20 bins and dashed lines, instead of 4 bins? What do you mean with 'replicates' in the**
**Fig legends?**

Stimulated by the reviewer's earlier comment, we now do not use bins throughout the manuscript (and the trajectory
analyses are anyway based on cell types). However, we have modified this figure (Extended Data Fig. 8), so that it is
now based on the eight (sub)cell types identified in primates instead of the four principal cell types. We had previous
put plots for the four cell types, to make it more comparable to the amniote analysis of these genes (Fig. 3, where we
can only compare the four principal cell types – see above). However, now we think it is better to exploit the higher
resolution afforded by our primate analyses, and the conservation of the trajectories in both the primate and amniote

analysis remain nicely visible. As requested by the reviewer and consistent with main Fig. 3a-c, we have now also added
 dashed lines to indicate the principal spermatogenic phases in Extended Data Fig. 8c.
 By “replicates”, we mean biological replicates (i.e., samples from different individuals). We now clarify this in the legend
 (see legend of the pasted Extended Data Fig. 7 below).

 **Extended Data Fig. 8 | Conservation of gene expression trajectories along spermatogenesis. a**, Global trajectory conservation score
 across amniotes for changed and conserved trajectories ($P < 2.2 \times 10^{-16}$, two-sided Wilcoxon rank-sum test). **b**, Trajectory conservation score
 between human and mouse of genes that lead to either a fertile or an infertile phenotype when knocked out in mouse ($P =$
 0.007 , two-sided Wilcoxon rank-sum test). **c**, Gene expression trajectories (*DMRT1*, top; *IP6K1*, bottom) along primate
 spermatogenesis (spermatogenesis, from left to right). Dots depict the median gene expression across biological replicates (i.e.,
 samples from different individuals) for each cell type and the whiskers show minimum and maximum gene expression values across
 replicates. The dashed vertical lines separate spermatogonia (SG), spermatocytes (SC), round (rSD) and elongated spermatids (eSD).

 **Ext Data Fig 8: could you explain more extensively in the MM how the 'testis-specific' genes were identified? Was**
 **the only threshold RPKM>=1? What is meant by 'genome wide' gene expression? How was that calculated: is this**
 **RPKM>=1 is each of the organ (brain/heart/kidney/liver) but RPKM<1 in testis?**

To identify testis-specific genes for the analyses in Extended Data Fig. 12 and also in Fig. 4, we used our previous
 extensive bulk tissue data set (Cardoso-Moreira et al. *Nature* 2019; PMID: 31243369); that is, we required RPKM to be
 greater or equal to 1 in the testis but lower than 1 in the other four organs (i.e., brain, heart, kidney, and liver).

“% genes genome-wide” in the labeling of the X-axes of the plots in Extended Data Fig. 12 refers to the percentage of
 all genes in the genome located on a given chromosome displayed in the plots (i.e., the number of genes on a given
 chromosome divided by the total number of genes in the genome).

Thus, for example, in Extended Data Fig. 12a, in the human plot (top left), 25% of the testis-specific genes with
 predominant expression in spermatogonia are located on the X chromosome (referred as “% genes testis-specific”
 on the Y axis). This represents a significant enrichment of such genes on the X, given that only 4.2% of genes in the genome
 are located on this chromosome (i.e., if such genes would be randomly located in the genome, we would expect
 approximately 4.2% of them to be located on chromosome X).

We have now refined the paragraph in the corresponding Methods section (“Cell type and testis-specific genes per
chromosome”) describing these aspects as well as further details of the analyses shown in these figures (e.g., the
identification of spermatogenesis cell type specific genes among the testis-specific genes), which now reads:

(P24, L935-948) “Testis-specific genes were obtained from previously generated RNA-seq data⁴ of adult organs (RPKM
≥ 1 in testis and RPKM < 1 in brain, cerebellum, heart, kidney, and liver). Among these, cell-type specific genes were
studied for each chromosome. Genes with predominant expression in specific somatic cell types were identified using
the *FindAllMarkers* function (parameter: only.pos = TRUE, min.pct = 0.05, logfc.threshold = 0.25, return.thresh = 0.05).
Predominant expression of genes in specific germ cell types was assigned based on the trajectory analyses (see above);
that is, predominant expression was assigned based on the cell type in which the expression level of the gene peaks in
the trajectory analysis. We then first calculated the percentage of genes located on a given chromosome among all
genes in the genome (x-axis of plots in Extended Data Fig. 12a, c; red horizontal line for X-linked genes in Fig. 4b). We
then contrasted this with the percentage of testis-specific genes with predominant expression in a given cell type (y-
axis of plots in Extended Data Fig. 12a, c; y-axis of plots in Fig. 4b for X-linked genes). Finally, the percentage of testis-
specific genes per cell type and chromosome was statistically compared to the percentage of genes per chromosome in
the genome using exact binomial tests.”

We now also clearly define the meaning of “% genes genome-wide” in the legend of Extended Data Fig. 12:

(P42, L1316-1322) “Extended Data Fig. 12 | Per chromosome testis-specific genes enriched in spermatogonia and
Sertoli cells. **a, c**, Per chromosome percentage of expressed testis-specific genes (y-axis) with predominant expression
in differentiated spermatogonia (human and macaque) or spermatogonia (mouse, opossum and platypus) (**a**) and
Sertoli cells (**c**) versus the percentage of all genes in the genome located on a given chromosome (x-axis: “% genes
genome-wide”; i.e., the number of genes on a given chromosome divided by the total number of genes in the genome).
X chromosomes are colored in red. The diagonal shows the numbers expected if testis-specific genes were randomly
distributed across the genome. **b**, Location of platypus testis-specific genes enriched in spermatogonia on chromosomes
X1, X3 and X5.”

**Ext Data Fig 10: either use the bins or use the clusters...using them interchangeably throughout the manuscript is**
**confusing.**

Agreed. We now only plot the data for the different cell types and do not bin the data anymore.

**Some Supp Tables are not well formatted (need to have commas instead of dots): ST1, ST2**

Thank you for alerting us to this issue. The formatting depends on the Excel version/language (i.e., it looks fine with our
settings). We now reformatted to display numbers only so the user can choose a way to display them.

**I would suggest the authors mention the limitations of the study: including number of individuals; limitations of**
**technique and limitations of the computational analysis that the authors may be aware of.**

We believe that the main limitations of study are related to our snRNA-seq data. While our data has key advantages
(e.g., snRNA-seq allowed the generation of data from frozen samples from otherwise inaccessible species; testicular cell
type diversity is well covered), it also has limitations. Mainly, it does not cover the cytoplasmic transcriptome, isoform
variation, and post-transcriptional information. We summarize these limitations at the end of the Discussion section:

(P15, L521-528) “Our data and results, together with the accompanying online resource we have developed
(<https://apps.kaessmannlab.org/SpermEvol/>) to facilitate the exploration of gene expression profiles across species,
provide an extensive resource for investigating the biology of the testis and associated fertility disorders across
mammals. Future studies should seek to complement our snRNA-seq data to overcome its limitations. Single-cell RNA-
seq data will be valuable for inferring transcriptome patterns unique to the cytoplasm, single-cell full-length transcript

data is needed to assess the pronounced isoform diversity of the testis¹⁴, and single-cell translome data⁷⁰ is required
to understand the contribution of post-transcriptional changes⁸ to the evolution of spermatogenesis.”

Reviewer Reports on the First Revision:

Referees' comments:

Referee #1 (Remarks to the Author):

I thank the authors for their careful revisions following my and the other reviewers' comments. The authors have appropriately addressed the issues I raised. This is an exciting, well presented manuscript with beautiful data, and I have no further requests for modifications.

Of note, I respectfully disagree with some of my colleagues' comments that the recent publication of similar datasets in human, macaque and mouse severely undermines the novelty of this work - I believe that the data in varied model and non-model non-human primates offers exciting perspectives for the field of evolutionary genomics.

With my best wishes,

Camille Berthelot

Referee #2 (Remarks to the Author):

The authors have addressed all concerns raised by reviewers, the revisions have significantly improved the manuscript and the manuscript is ok from my end.

Referee #3 (Remarks to the Author):

Dear Authors,

You have considerably increased the quality and clarity of the manuscript. You went the extra mile to include all our suggestions and provided a detailed point-by-point answer.
Congratulations!

The only think I missed was the inclusion of a marked manuscript to be able to see the changes in context in the manuscript. I am not sure why this was not provided, but it is pity it was not in place.